# ENTROPY-GUIDED DYNAMIC TOKENS FOR GRAPH–LLM ALIGNMENT IN MOLECULAR UNDERSTANDING

**Zihao Jing**[1], **Qiuhao Zeng**[1], **Ruiyi Fang**[1], **Yan Sun**[1], **Boyu Wang**[1], **Pingzhao Hu**[1,2] *

[1]Department of Computer Science, Western University, London, ON, Canada
[2]Department of Biochemistry, Western University, London, ON, Canada
`zjing29@uwo.ca, phu49@uwo.ca`

## ABSTRACT

Molecular understanding is central to advancing areas such as scientific discovery, yet Large Language Models (LLMs) struggle to understand molecular graphs effectively. Existing graph–LLM bridges often adapt the Q-Former-style connector with fixed-length static tokens, which is originally designed for vision tasks. These designs overlook stereochemistry and substructural context and typically require costly LLM-backbone fine-tuning, limiting efficiency and generalization. We introduce **EDT-Former**, an **E**ntropy-guided **D**ynamic **T**oken Trans**former** that generates tokens aligned with informative molecular patches, thereby preserving both local and global structural features for molecular graph understanding. Beyond prior approaches, EDT-Former enables alignment between frozen graph encoders and LLMs without tuning the LLM backbone (excluding the embedding layer), resulting in computationally efficient finetuning, and achieves state-of-the-art results on MoleculeQA, Molecule-oriented Mol-Instructions, and property prediction benchmarks (TDC, MoleculeNet), underscoring its effectiveness for scalable and generalizable multimodal molecular understanding.

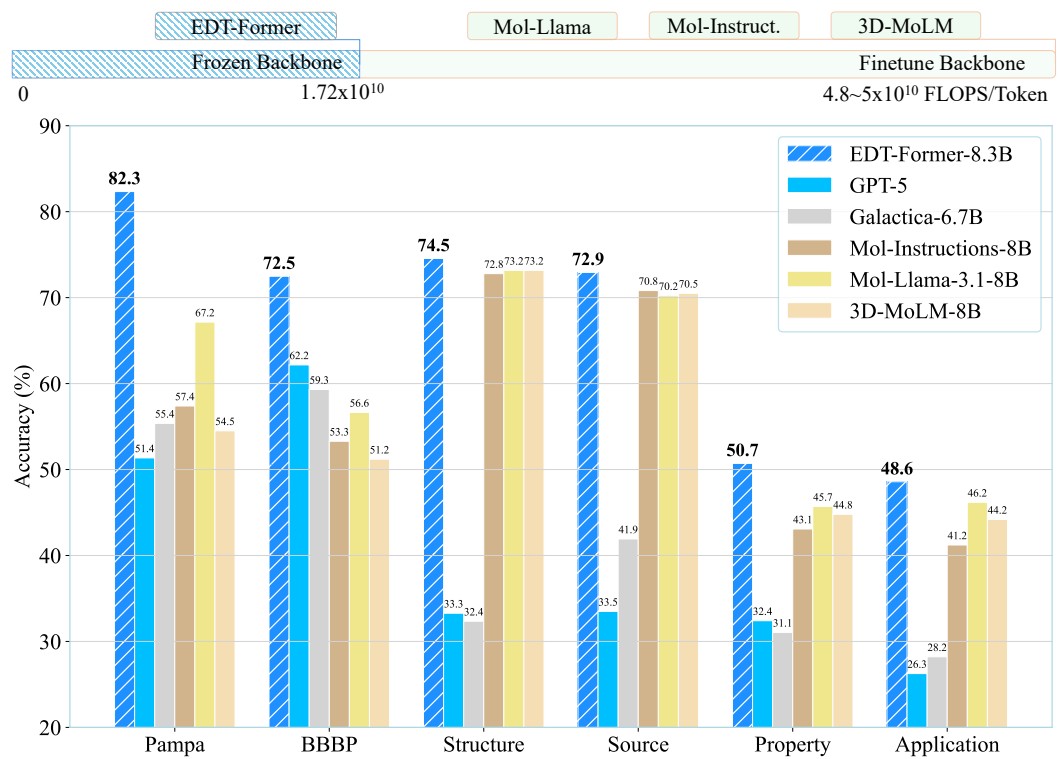

Figure 1: LLM joint fine-tuning efficiency and benchmark performance preview of EDT-Former.

---

*Corresponding author. Departments of Computer Science and Biochemistry, Western University, 1400 Western Road, London, Ontario N6G 2V4, Canada. E-mail: phu49@uwo.ca (P.H.)

# 1  INTRODUCTION

**Background.**  Large language models (LLMs) such as Llama (Touvron et al., 2023a; MetaAI, 2024) and GPT (OpenAI, 2023; 2025) series demonstrate powerful multimodal reasoning across domains. This rapid progress highlights the potential adaptation to specialized scientific domains, and NatureLM (Xia et al., 2025) exemplifies this direction in scientific knowledge. Within molecular science, recent foundation models have extended large-scale pretraining to molecular graphs and geometry structures. Uni-Mol-v2 (Ji et al., 2024) leverages extensive conformer datasets to jointly model atomic, graph, and geometric representations, while UniMolT (Zhang et al., 2024) adapts instruction-tuned LLMs to molecular reasoning and understanding tasks.

To bridge molecular graphs with natural language, multimodal connectors have been introduced. Most approaches adopt the Q-Former mechanism (Li et al., 2023), where a fixed number of modality-anchor tokens are learned to query structural encoders. Representative examples include 3D-MolT5 (Pei et al., 2024b), Mol-LLM (Lee et al., 2025), and Mol-LLaMA (Kim et al., 2025), which demonstrate the feasibility of aligning molecular structures (graph and geometry) with language models and enabling cross-modal molecular understanding.

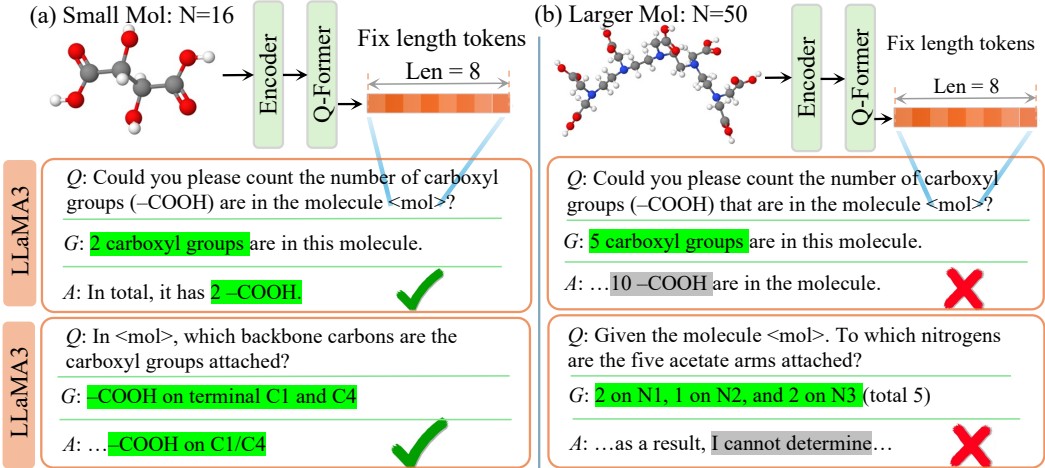

Figure 2: Illustration of motivation–Loss of structure. Comparison of molecules of different sizes (atom counts $N = 16$ and $N = 50$) encoded by the same fixed query length Q-Former bridge (8 query tokens) to Llama-3.1-8B backbone, with example prompts and generated responses.

**Challenges.**  **(1) Loss of structure.** Current graph–LLM bridges typically fuse modalities by introducing a fixed number of learnable query tokens, which interact with molecular encoder outputs and are then fed into the language model. (Lee et al., 2025; Kim et al., 2025). However, compressing length-heterogeneous molecules into the fixed set collapses critical features such as stereochemistry and functional groups. This loss carries over to reasoning, where even chain-of-thought prompting provides little gain and can reduce accuracy on property prediction (Kim et al., 2025). As shown in Fig. 2, fixed-length fusion captures key groups in small molecules (N=16 atoms) and supports structure-aware tasks; however, in larger molecules (N=50 atoms), it results in incomplete substructure coverage, information loss, and consequently brittle, chemically unfaithful predictions.

**(2) Heavy fine-tuning.** Most prior systems train the bridge jointly with the LLM, requiring gradient updates to the backbone itself. Such updates hinder generalization: models overfit to narrow datasets, lose robustness across modalities and structural variations, and fail to transfer alignment when scaled to larger frozen backbones.

Table 1: Computing costs comparison between frozen/unfrozen backbone settings of Mol-Llama.

| Connector | LLM | Trainable | FLOPs/Token | Time/Step |
|---|---|---|---|---|
| Finetune | Finetune | 8.1B | 4.9e10 | 0.93 |
| Finetune | Frozen | 84M | 1.7e10 | 0.23 |

Clearly, they are also computationally inefficient: as shown in Tab. 1, jointly tuning the LLM and connector requires ×96 more trainable parameters than tuning the connector alone (details are in App. D.3). Consequently, a connector-only bridge with frozen encoders and a frozen LLM is urgently needed to achieve scalable, low-cost deployment.

**Our approach.** We propose EDT-Former, an Entropy-guided Dynamic Token Transformer that aligns chemical graphs with frozen LLMs while preserving structural fidelity. EDT-Former introduces an **Entropy-Guided Patching** strategy that segments molecules into informative sub-groups, generating dynamic query tokens that retain stereochemistry. It further incorporates the **Dynamic Query Transformer** module that integrates these dynamic tokens with learned modality anchors to form a stable cross-modal interface before mapping into the LLM embedding space. Together, these designs enable frozen-backbone alignment, yielding efficient training, robust generalization, and chemically faithful understanding (Fig. 1). Our contributions are summarized as follows:

- This work presents **EDT-Former**, the first connector-only method that aligns chemical graphs with frozen LLMs via dynamic, substructure-aware query tokens.
- We present **Entropy-Guided Patching** and **Dynamic Query Transformer**, which together enable efficient cross-modal alignment without updating backbone parameters.
- EDT-Former achieves state-of-the-art results on molecular understanding and property prediction benchmarks, demonstrating both scalability and robust generalization.

## 2 RELATED WORK

**Multimodal Fusion for Molecular Generative Modeling.** Beyond SMILES-based models (e.g., KV-PLM (Zeng et al., 2022) and Mol-Instructions (Fang et al., 2023)), recent work has attempted to fuse SMILES, topological graphs, and 3D conformers with large language models. GIMLET (Zhao et al., 2023) showed that instruction tuning over paired graph-text corpora can provide LLMs with basic molecular understanding. Larger frameworks extended modality coverage: HIGHT (Chen et al., 2024) uses a hierarchical BRICS-based graph tokenization approach, and MoleculeSTM (Liu et al., 2022), MolFM (Luo et al., 2023a), BioT5+ (Pei et al., 2024a), and ProLLaMA (Lv et al., 2024) combine discrete structure tokens, biomedical knowledge graphs, and protein sequences, while 3D-MolT5 (Pei et al., 2024b) incorporates coarse 3D geometry for conditional generation. Most recently, UniMolT (Zhang et al., 2024), Mol-LLM (Lee et al., 2025), and Mol-Llama (Kim et al., 2025) adopt the Q-Former mechanism from BLIP-2 (Li et al., 2023) to bridge molecular graphs and LLMs, relying on a fixed-length learnable modality anchor tokens. Such fixed-length fusion compresses stereochemistry and graph substructures, leading to structural information loss. More flexible alignment strategies are needed to retain sub-structural fidelity as molecular complexity and conformer diversity increase.

**Molecular Understanding with Multimodal LLMs.** Molecular understanding has been explored by combining structural encoders with instruction-tuned LLMs. MolReasoner Zhao et al. (2025), StructCoT (Jang et al., 2024) showed that coupling molecular tokens with prompts enables step-wise reaction explanation and property inference. Subsequent efforts introduced structural bias, such as Mol-LLM (Lee et al., 2025) with topology-conditioned attention and ProLLaMA (Lv et al., 2024) with protein–ligand co-representations. However, most approaches still depend on fixed-length anchor token representations or extensive fine-tuning of the LLM, which weakens structural fidelity and makes training inefficient. Recent chemical agents, such as ChemCrow (Bran et al., 2023), attempt to add planner feedback but remain constrained by conformer sensitivity and limited scalability. Progress, therefore, requires a multimodal alignment method that preserves modality-specific cues without relying on heavy LLM tuning and adaptation. More related works and relationships to current works are discussed in App. B.

## 3 EDT-FORMER: ENTROPY-GUIDED DYNAMIC GRAPH–LLM ALIGNMENT

EDT-Former provides efficient and substructure-aware alignment between molecule and LLM. Sec. 3.1 outlines the overall architecture, after which Sec. 3.2 introduces the Entropy-Guided Patching strategy for dynamic token generation from graph node-embeddings, while Sec. 3.3 presents the Dynamic Query Transformer module. Finally, Sec. 3.4 describes the training process under the frozen-backbone regime. The implementation details are discussed in App. C.

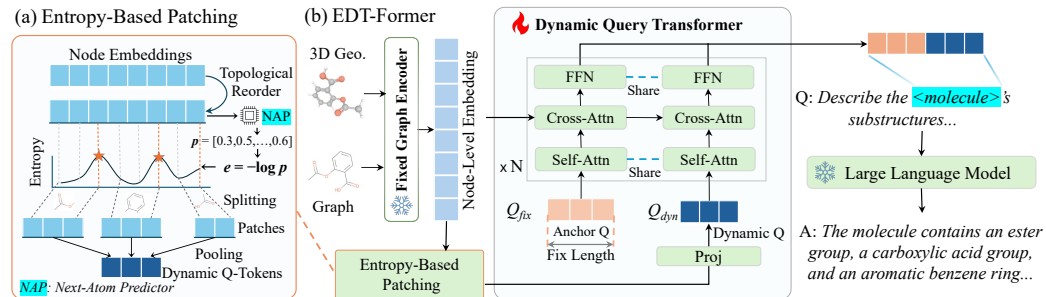

Figure 3: The architecture of EDT-Former. (a) Entropy-based Patching segments node embeddings into patches to produce dynamic query tokens. (b) EDT-Former integrates anchors and dynamic queries through Dynamic Query Transformer to align the molecular graph with the LLM.

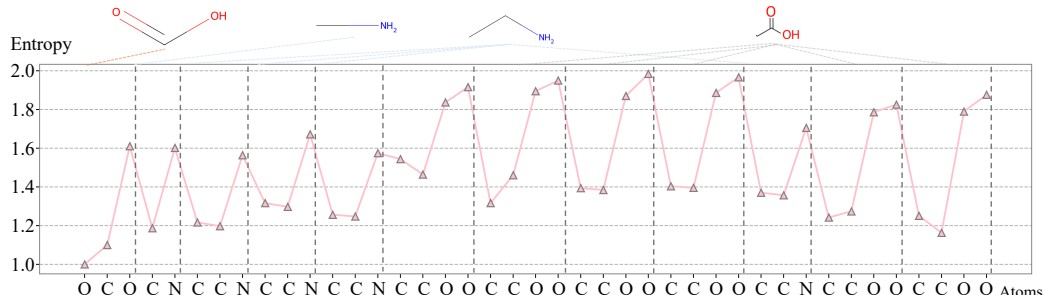

Figure 4: Illustration of entropy-guided patching on an example molecule. Atom-level entropy is plotted along the molecular sequence, and a new patch is initiated after each local maximum.

## 3.1 STRUCTURE OVERVIEW

EDT-Former establishes alignment between molecular encoders and frozen LLMs through two complementary components: Entropy-Guided Patching and Dynamic Query Transformer. As shown in Fig. 3, node embeddings from frozen graph encoders are processed by entropy-guided patching, which measures node-level uncertainty for language models, segments molecules into informative sub-groups, and pools them into dynamic tokens that preserve local graph features. These tokens are combined with static modality anchor tokens in a shared query bank, where self-attention propagates context and cross-attention retrieves structural evidence from molecular embeddings. The resulting representations are projected into the LLM embedding space, forming a cross-modal alignment interface that balances global consistency from modality anchors with local fidelity from dynamic query tokens, enabling modality alignment with high efficiency and structural fidelity.

## 3.2 ENTROPY-GUIDED SUB-GRAPH PATCHING

**Setup.** As shown in Fig. 3(a), let a molecule be represented by a SMILES-ordered atom sequence $(a_1, \ldots, a_T)$. We pre-train a lightweight Next-Atom Predictor $f_\theta$ (NAP, a small Transformer) on large canonical SMILES corpora to model $p(a_{t+1} \mid a_{1:t})$. At inference time, given logits $L_t \in \mathbb{R}^V$ over the atom vocabulary $\mathcal{V}$, the probability assigned to the ground-truth next atom is

$$p_t = \mathrm{softmax}(L_t)[a_{t+1}], \qquad t = 1, \ldots, T-1. \tag{1}$$

The information content is defined (negative log-likelihood) as

$$e_t = -\log p_t \quad \text{(nats; use } \log_2 \text{ for bits if desired)}. \tag{2}$$

**Peak-based segmentation.** Instead of thresholding, we identify local maxima of the entropy signal $\{e_t\}$ and place split points at peaks (shown in Fig. 4):

$$\mathrm{peak}(t) = \Big[ (e_{t-1} < e_t) \wedge (e_t > e_{t+1}) \Big], \tag{3}$$

optionally enforcing a minimal separation $\Delta$ (non-maximum suppression) and a small prominence $\gamma$ to remove spurious bumps:

$$\mathrm{NMS}(t) = \Big[ \min_{|s-t| \leq \Delta} e_t \geq e_s \Big], \qquad \mathrm{prom}(t) = \Big[ e_t - \tfrac{1}{2}(e_{t-1} + e_{t+1}) \geq \gamma \Big]. \tag{4}$$

---

**Algorithm 1:** Dynamic Query Transformer (anchors + dynamic tokens)

---

**Input** : Graph $G$; frozen graph encoder $\mathcal{E}$; frozen LLM $\mathcal{L}$; dynamic tokens $Z = [z_1, \ldots, z_M]$
      from Sec. 3.2; #anchors $k$; layers $L$; projector $W_{\text{proj}}$.
**Output:** $U \in \mathbb{R}^{(k+M) \times d_{\text{LLM}}}$ (LLM-conditioning sequence).
**Node embeddings:** $X \leftarrow \mathcal{E}(G)$ ;                    `// frozen molecular encoder`
**Anchors:** $Q_{\text{fix}} \in \mathbb{R}^{k \times d}$ ;                    `// learnable, modality-stable`
**Query bank:** $Q_{bank} \leftarrow [Q_{\text{fix}}; Z] \in \mathbb{R}^{(k+M) \times d}$ ;        `// construct query bank`

**Multimodal Alignment (L layers):** ;         `// Stack of Transformer Blocks`
**for** $\ell \leftarrow 1$ **to** $L$ **do**
  $\quad Q_{bank} \leftarrow Q_{bank} + \text{Self-Attn}(Q_{bank})$ ;         `// mix anchors ↔ dynamics`
  $\quad Q_{bank} \leftarrow Q_{bank} + \text{Cross-Attn}(Q_{bank}, K = X, V = X)$ ;  `// retrieve node-level`
   `  evidence`
  $\quad Q_{bank} \leftarrow Q_{bank} + \text{FFN}(Q_{bank})$ ;     `// shared FFN for all query tokens`

**Projection to LLM space:** $U \leftarrow W_{\text{proj}} Q_{bank}$ ;                    `// (k+M)×d`$_{\text{LLM}}$
**return** $U$ ;          `// LLM consumes U; L frozen; train bridge only`

---

Then, the valid split indices are collected as $\mathcal{T}_\star = \{ t \in \{2, \ldots, T-2\} \mid \text{peak}(t) \wedge \text{NMS}(t) \wedge \text{prom}(t) \}$ and form dynamic segments by cutting after each peak:

$$1 = \tau_0 < \tau_1 < \cdots < \tau_M = T, \quad \{\tau_1, \ldots, \tau_{M-1}\} = \mathcal{T}_\star, \quad \mathcal{S}_k = \{ t \mid \tau_{k-1} \le t < \tau_k \}. \quad (5)$$

**Graph mapping and pooling.** Let $\pi : \{1, \ldots, T\} \to \{1, \ldots, N\}$ map SMILES positions to graph node indices. Each segment $\mathcal{S}_k$ induces a substructure (node set)

$$\widehat{\mathcal{S}}_k = \{ \pi(t) : t \in \mathcal{S}_k \} \subseteq \{1, \ldots, N\}. \quad (6)$$

Given frozen graph-encoder node embeddings $X \in \mathbb{R}^{N \times d}$, we construct one query token for each substructure via average pooling:

$$z_k = \text{AvgPool}\big(\{ X_i : i \in \widehat{\mathcal{S}}_k \}\big) \in \mathbb{R}^d, \qquad k = 1, \ldots, M. \quad (7)$$

The collection $\{z_k\}_{k=1}^M$ constitutes dynamic query tokens for Dynamic Query Transformer.

**Rationale.** The entropy $e_t$ quantifies the predictive uncertainty of extending the current SMILES atom with $a_{t+1}$ under a lightweight Next-Atom Predictor (NAP). Local peaks indicate positions that are hard for a SMILES-based sequence model to predict, and grouping tokens between peaks yields data-driven patches that mark information-dense segments in the 1D sequence, rather than exact chemical subgraphs. Since SMILES is generated via a DFS traversal of the molecular graph, adjacent tokens typically reflect nearby atoms, so these entropy patterns still inherit meaningful local structure while remaining strictly sequence-based and aligned with the LLM's view. This entropy-driven segmentation is simple, reproducible, and automatically matches dynamic-token count to molecular complexity. The theoretical analysis of the rationale of Entropy-Guided Patching is discussed in App. A.1, and the detailed analysis is in App. D.9.

## 3.3 DYNAMIC QUERY TRANSFORMER

Dynamic Query Transformer integrates fixed modality anchors with dynamic substructure tokens to form a compact, structure-aware interface to a frozen LLM, as shown in Fig. 3(b). Let $X \in \mathbb{R}^{N \times d}$ be node embeddings from the frozen graph encoder and let $Z = [z_1, \ldots, z_M] \in \mathbb{R}^{M \times d}$ be dynamic tokens from Sec. 3.2. As shown in Algorithm 1, we initialize $k$ learnable anchors $Q_{\text{fix}} \in \mathbb{R}^{k \times d}$ and build a query bank $Q_{bank} = [Q_{\text{fix}}; Z] \in \mathbb{R}^{(k+M) \times d}$. A lightweight transformer with $L$ layers refines $Q_{bank}$ by (i) self-attention over queries to mix global and local context, (ii) cross-attention from queries to node embeddings $X$ to retrieve substructure evidence, and (iii) a shared feed-forward network. The enriched queries are projected to the LLM embedding space via $U = W_{\text{proj}} Q_{bank} \in \mathbb{R}^{(k+M) \times d_{\text{LLM}}}$ and then consumed by the frozen LLM as conditioning. During training, only the bridge parameters (anchors, attention/FFN, and $W_{\text{proj}}$) are updated; both the molecular encoder and the LLM remain frozen, yielding an efficient graph–LLM alignment.

Table 2: Zero-shot accuracy (%) on 10 molecular property prediction tasks from MoleculeNet and TDC benchmarks, where each value is averaged over 13 shared prompts per task for fairness. Prompt settings details are in App. D.5. The best (pink) and second-best (lightpink) results are highlighted.

| Model | Size | PAMPA | BBBP | BACE | CLINTOX | DILI | HERG | HIA | HIV | AMES | PGP |
|---|---|---|---|---|---|---|---|---|---|---|---|
| *General LLMs* | | | | | | | | | | | |
| GPT-4o | - | 50.52 | 63.52 | 39.48 | 26.34 | 53.47 | 64.04 | 76.00 | 17.02 | 53.59 | 50.21 |
| Llama2 | 7B | 69.27 | 49.06 | 38.96 | 26.05 | 51.26 | 65.02 | 76.66 | 15.20 | 53.08 | 51.09 |
| Llama3.1 | 8B | 58.00 | 55.64 | 40.44 | 23.93 | 52.31 | 64.28 | 78.07 | 16.34 | 53.46 | 50.99 |
| *Molecular LLMs* | | | | | | | | | | | |
| 3D-MoLM | 7B | 54.50 | 51.20 | 41.43 | 25.92 | 48.78 | 62.67 | 72.93 | 26.16 | 53.01 | 51.55 |
| LLaMo | 7B | 53.09 | 57.18 | 41.91 | 25.55 | 49.08 | 62.77 | 72.75 | 24.39 | 51.67 | 51.22 |
| Mol-Ins.-Llama2 | 8B | 40.78 | 51.22 | 42.61 | 25.99 | 48.91 | 62.79 | 72.78 | 22.82 | 53.72 | 50.72 |
| Mol-Ins.-Llama3.1 | 8B | 57.39 | 53.29 | 40.12 | 26.79 | 52.39 | 47.23 | 50.95 | 16.70 | 48.98 | 52.70 |
| Mol-LLaMA-2 | 7.2B | 74.23 | 53.99 | 41.54 | 27.96 | 51.92 | 65.60 | 67.88 | 22.51 | 52.97 | 51.14 |
| Mol-LLaMA-3.1 | 8.3B | 67.15 | 56.64 | 38.68 | 23.07 | 51.76 | 70.61 | 78.25 | 21.89 | 54.50 | 50.73 |
| *Ours* | | | | | | | | | | | |
| EDT-Former | 8.3B | 82.34 | 72.48 | 49.12 | 56.55 | 50.27 | 73.46 | 82.15 | 46.56 | 55.20 | 54.64 |

## 3.4 FROZEN-BACKBONE ALIGNMENT TRAINING

EDT-Former is trained via a two-stage regimen (pretraining, then alignment tuning) on the Mol-LLaMA-Instruct dataset (Kim et al., 2025). In pretraining, only the Dynamic Query Transformer is optimized together with frozen graph encoders, without attaching the LLM. Objectives follow standard adapter-style practice (cf. Q-Former): a cross-modal contrastive loss for global cohesion, an anchor–modality matching loss to stabilize the fixed interface, and a masked substructure reconstruction loss to inject local chemical semantics. Details are discussed in App. C.2.

In the alignment tuning stage, the frozen LLM is attached while the graph encoders and LLM remain frozen; only the parameters of Dynamic Query Transformer are updated to align structure-aware queries with instruction prompts. This protocol preserves backbone stability, enables efficient adaptation, and yields a chemically faithful graph–LLM interface. Detail are in App. C.3

## 4 EXPERIMENTS

In this section, EDT-Former is evaluated on benchmarks—MoleculeQA (Lu et al., 2024), Molecule-oriented Mol-Instructions (Fang et al., 2023), Pampa, and BBBP from TDC Huang et al. (2021), addressing the following questions: **Q1)** Does EDT-Former outperform existing multimodal molecular LLM baselines? **Q2)** Can Entropy-Guided Patching capture subgraph features? **Q3)** Does the Dynamic Query Transformer improve LLM understanding compared to fixed-length query connectors? **Q4)** Does freezing the LLM reduce compute while retaining strong accuracy? See App. D for baseline and dataset descriptions, comprehensive experiment implementation, benchmark and evaluation details.

### 4.1 PROPERTY PREDICTION TASKS

As shown in Tab. 2, we evaluate on ten molecular property prediction tasks from TDC (Huang et al., 2021) and MoleculeNet (Wu et al., 2018) benchmarks using official splits and zero-shot inference. Comparisons cover general LLMs and molecular LLMs (including multimodal variants). To reduce prompt sensitivity, the average accuracy over 13 prompt settings is reported, with all the prompt settings per-prompt accuracies in App. D.5). Under identical conditions, EDT-Former attains the best results on both tasks, with >20% relative gains over the strongest baseline and pushing average accuracy above 70% on BBBP, HIA, and PAMPA datasets, indicating practical viability for LLM-based property prediction (**Q1**). Potential dataset imbalance is discussed in App. D.11.

### 4.2 MOLECULEQA (REASONING AND UNDERSTANDING)

Evaluation is conducted on the large-scale MoleculeQA benchmark (Lu et al., 2024) (Tab. 3), a multiple-choice suite covering four tasks—structure, source, property, and application. Large GPT

Table 3: Performance on the MoleculeQA benchmark. Models are compared across four tasks (Structure, Source, Property, Application) with accuracy (%) reported. The best (pink) and second-best (lightpink) results are highlighted. Modality T/G/3D represents text, graph, and 3D geometry.

| Model | Modality | Imp. | Size | Strct. | Src. | Prop. | App. | Avg. | Total |
|---|---|---|---|---|---|---|---|---|---|
| *Random* | | | | | | | | | |
| Random | - | - | - | 24.41 | 22.30 | 23.04 | 24.57 | 23.58 | 24.03 |
| *Molecular LLMs* | | | | | | | | | |
| BioMedGPT-LM | Text | SFT | 7B | 54.19 | 60.01 | 38.85 | 40.90 | 48.49 | 52.23 |
| Mol-Inst.-Llama3.1 | T/G | SFT | 8B | 72.79 | 70.82 | 43.08 | 41.22 | 56.98 | 65.31 |
| 3D-MoLM | T/G/3D | SFT | 8B | 73.17 | 70.50 | 44.79 | 44.19 | 58.16 | 65.96 |
| LLaMo | Text | SFT | 8B | 70.56 | 66.63 | 44.60 | 45.18 | 56.74 | 63.74 |
| Mol-Llama3.1 | Text/Graph/3D | SFT | 8.2B | 73.16 | 70.22 | 45.70 | 46.18 | 58.82 | 66.21 |
| *General LLMs* | | | | | | | | | |
| Galactica | Text | SFT | 6.7B | 32.35 | 41.92 | 31.05 | 28.21 | 33.38 | 33.96 |
| BLOOM | Text | SFT | 7.1B | 35.01 | 47.51 | 31.46 | 33.56 | 36.89 | 37.31 |
| Pythia | Text | SFT | 6.9B | 42.79 | 58.90 | 38.58 | 39.07 | 44.84 | 45.61 |
| Llama-2-chat | Text | SFT | 7B | 28.75 | 39.84 | 31.33 | 27.71 | 31.91 | 31.54 |
| Vicuna-v1.5 | Text | SFT | 13B | 37.01 | 43.19 | 30.64 | 31.55 | 35.60 | 37.07 |
| *Large Scale General LLMs* | | | | | | | | | |
| GPT-3.5 | Text | 10-Shot | - | 25.60 | 37.60 | 28.04 | 32.22 | 30.87 | 29.29 |
| GPT-4 | Text | 10-Shot | - | 60.94 | 50.19 | 35.57 | 43.91 | 47.65 | 53.47 |
| GPT-5 | Text | 10-Shot | - | 62.78 | 53.22 | 36.42 | 46.91 | 49.83 | 56.12 |
| *Ours* | | | | | | | | | |
| EDT-Former | T/G/3D | 10-Shot | 8.3B | 66.46 | 60.98 | 40.35 | 36.40 | 51.05 | 58.78 |
| **EDT-Former** | T/G/3D | SFT | 8.3B | 74.55 | 72.39 | 50.71 | 48.58 | 61.56 | 68.34 |

Table 4: Results on the Mol-Instructions dataset. Models are finetuned and evaluated on two tasks: molecular description generation (BLEU, ROUGE, and METEOR scores) and molecular property prediction (MAE). The best (pink) and second-best (lightpink) results are highlighted.

| Models | Molecular Description | | | | | | Property (MAE) |
|---|---|---|---|---|---|---|---|
| | BLUE-2 | BLUE-4 | ROUGE-1 | ROUGE-2 | ROUGE-L | METEOR | |
| Alpaca-7B | 0.068 | 0.014 | 0.178 | 0.041 | 0.136 | 0.107 | 322.109 |
| Baize-7B | 0.064 | 0.015 | 0.189 | 0.053 | 0.148 | 0.106 | 261.343 |
| LLaMA-2-7B | 0.059 | 0.014 | 0.164 | 0.066 | 0.148 | 0.184 | 5.553 |
| Vicuna-v1.5-13B | 0.052 | 0.011 | 0.151 | 0.055 | 0.130 | 0.168 | 860.051 |
| Galatica-6.7B | 0.024 | 0.008 | 0.074 | 0.015 | 0.063 | 0.065 | 0.568 |
| Qwen3-8B | 0.098 | 0.029 | 0.207 | 0.050 | 0.157 | 0.173 | 4.737 |
| Mol-Inst.-Llama2-7B | 0.217 | 0.143 | 0.337 | 0.196 | 0.291 | 0.254 | 0.013 |
| Mol-Inst.-Llama3.1-8B | 0.419 | 0.361 | 0.719 | 0.646 | 0.709 | 0.637 | 15.059 |
| Mol-LLaMA2-7.2B | 0.433 | 0.385 | 0.711 | 0.649 | 0.601 | 0.601 | 0.0087 |
| Mol-LLaMA3.1-8.2B | 0.445 | 0.398 | 0.717 | 0.656 | 0.709 | 0.617 | 0.0079 |
| MolReasoner-7B | 0.438 | 0.322 | 0.553 | 0.366 | 0.482 | 0.475 | 10.323 |
| **EDT-Former-8.3B** | 0.424 | 0.402 | 0.726 | 0.652 | 0.717 | 0.631 | 0.0062 |

models are assessed in a 10-shot setting, while all other models are fine-tuned on the official splits for the same number of epochs under matched hyperparameters. Under these conditions, our SFT model consistently achieves the best performance across all four tasks, demonstrating strong molecular reasoning. Notably, the 10-shot variant of EDT-Former outperforms the newest GPT-5 model, indicating an efficient trade-off between scale and domain alignment. These results answer **Q1**.

## 4.3 MOL-INSTRUCTIONS (MOLECULE-ORIENTED)

For benchmark Mol-Instructions (Fang et al., 2023), we evaluate all 6 molecule-oriented and 1 open-question tasks. For fair comparison, all baselines and EDT-Former are fine-tuned under identical settings on the official splits. Tab. 4, 5, 6, 7 show the performance on Molecule description generation, Property prediction, Retrosynthesis, Forward reaction prediction, and Reagent prediction tasks (with Description-guided molecule design task reported in App. D.6).EDT-Former attains stronger chemical ability on most metrics of all the tasks, indicating improved performance relative to both general and molecular LLM baselines. The detailed experiment settings and metrics are discussed in App D.3. We further investigate the influence on natural language ability of EDT-Former and Mol-Llama (Kim et al., 2025) when performing fine-tuning in App D.7. These address **Q1**.

Table 5: Results on the Retrosynthesis task from the Mol-Instructions benchmark. Models are finetuned and evaluated on the official splits, reporting 7 metrics as defined by the benchmark. The best (pink) and second-best (light pink) results are highlighted.

| Model | Exact↑ | BLEU↑ | Levenshtein↓ | RDK FTS↑ | MACC FTS↑ | Morgan FTS↑ | Validity↑ |
|---|---|---|---|---|---|---|---|
| Alpaca | 0.000 | 0.063 | 46.915 | 0.005 | 0.023 | 0.007 | 0.160 |
| Baize | 0.000 | 0.095 | 44.714 | 0.025 | 0.050 | 0.023 | 0.112 |
| ChatGLM | 0.000 | 0.117 | 48.365 | 0.056 | 0.075 | 0.043 | 0.046 |
| LLaMa | 0.000 | 0.036 | 46.844 | 0.018 | 0.029 | 0.017 | 0.010 |
| Vicuna | 0.000 | 0.057 | 46.877 | 0.025 | 0.030 | 0.021 | 0.017 |
| Galactica | 0.000 | 0.452 | 34.940 | 0.167 | 0.274 | 0.134 | 0.984 |
| Text+Chem T5 | 0.141 | 0.765 | 24.043 | 0.685 | 0.765 | 0.585 | 0.698 |
| Mol-Ins.-Llama2 | 0.009 | 0.705 | 31.227 | 0.283 | 0.487 | 0.230 | 1.000 |
| Mol-Ins.-Llama3.1 | 0.333 | 0.842 | 17.642 | 0.704 | 0.815 | 0.646 | 1.000 |
| Mol-LLama-3.1 | 0.340 | 0.877 | 38.324 | 0.708 | 0.822 | 0.601 | 1.000 |
| HIGHT | 0.008 | 0.863 | 28.912 | 0.564 | 0.340 | 0.309 | 1.000 |
| EDT-Former | 0.387 | 0.930 | 34.202 | 0.721 | 0.836 | 0.670 | 1.000 |

Table 6: Results on the Forward Reaction Prediction task from Mol-Instructions benchmark. Models are fine-tuned and evaluated on the official splits, reporting 7 metrics as defined by the benchmark. The best (pink) and second-best (light pink) results are highlighted.

| Model | Exact↑ | BLEU↑ | Levenshtein↓ | RDK FTS↑ | MACC FTS↑ | Morgan FTS↑ | Validity↑ |
|---|---|---|---|---|---|---|---|
| Alpaca | 0.000 | 0.065 | 41.989 | 0.004 | 0.024 | 0.008 | 0.138 |
| Baize | 0.000 | 0.044 | 41.500 | 0.004 | 0.025 | 0.009 | 0.097 |
| ChatGLM | 0.000 | 0.183 | 40.008 | 0.050 | 0.100 | 0.044 | 0.108 |
| LLaMa | 0.000 | 0.020 | 42.002 | 0.001 | 0.002 | 0.001 | 0.039 |
| Vicuna | 0.000 | 0.057 | 41.690 | 0.007 | 0.016 | 0.006 | 0.059 |
| Galactica | 0.000 | 0.468 | 35.021 | 0.156 | 0.257 | 0.097 | 0.946 |
| Text+Chem T5 | 0.239 | 0.782 | 20.413 | 0.705 | 0.789 | 0.652 | 0.762 |
| Mol-Ins.-Llama2 | 0.045 | 0.654 | 27.262 | 0.313 | 0.509 | 0.262 | 1.000 |
| Mol-Ins.-Llama3.1 | 0.503 | 0.883 | 13.410 | 0.756 | 0.863 | 0.708 | 1.000 |
| Mol-LLama | 0.440 | 0.912 | 17.120 | 0.724 | 0.859 | 0.665 | 1.000 |
| HIGHT | 0.037 | 0.869 | 23.759 | 0.59 | 0.394 | 0.34 | 0.993 |
| EDT-Former | 0.471 | 0.964 | 18.130 | 0.776 | 0.871 | 0.712 | 1.000 |

Table 7: Results on the reagent prediction task from the Mol-Instructions benchmark. Models are fine-tuned and evaluated on the official splits, reporting 7 metrics as defined by the benchmark. The best (pink) and second-best (light pink) results are highlighted.

| Model | Exact↑ | BLEU↑ | Levenshtein↓ | RDK FTS↑ | MACC FTS↑ | Morgan FTS↑ | Validity↑ |
|---|---|---|---|---|---|---|---|
| Alpaca | 0.000 | 0.026 | 29.037 | 0.029 | 0.016 | 0.001 | 0.186 |
| Baize | 0.000 | 0.051 | 30.628 | 0.022 | 0.018 | 0.004 | 0.099 |
| ChatGLM | 0.000 | 0.019 | 29.169 | 0.017 | 0.006 | 0.002 | 0.074 |
| LLaMa | 0.000 | 0.003 | 28.040 | 0.037 | 0.001 | 0.001 | 0.001 |
| Vicuna | 0.000 | 0.010 | 27.948 | 0.038 | 0.002 | 0.001 | 0.007 |
| Galactica | 0.000 | 0.141 | 30.760 | 0.036 | 0.127 | 0.051 | 0.995 |
| Text+Chem T5 | 0.000 | 0.225 | 49.323 | 0.039 | 0.186 | 0.052 | 0.313 |
| Mol-Ins.-Llama2 | 0.044 | 0.224 | 23.167 | 0.237 | 0.364 | 0.213 | 1.000 |
| Mol-Ins.-Llama3.1 | 0.101 | 0.648 | 18.326 | 0.412 | 0.521 | 0.375 | 1.000 |
| Mol-LLama | 0.132 | 0.495 | 49.230 | 0.411 | 0.521 | 0.361 | 1.000 |
| HIGHT | 0.050 | 0.462 | 28.970 | 0.441 | 0.314 | 0.275 | 1.000 |
| EDT-Former | 0.145 | 0.650 | 46.950 | 0.464 | 0.531 | 0.431 | 1.000 |

## 4.4 ABLATION STUDIES

Comprehensive ablation experiments are conducted for question **Q2–Q4**, ranging from multimodal fusion to graph patching strategies for Entropy-Guided Patching and connector design for Dynamic Query Transformer. All experiments use official splits with matched token budgets and hyperparameter settings, see App. D. Additional ablation studies are in App. E.

**Component Effects.** Multimodal fusion and each core component are ablated on the four MoleculeQA tasks (Fig. 5). The full EDT-Former performs best across Structure, Source, Prop-

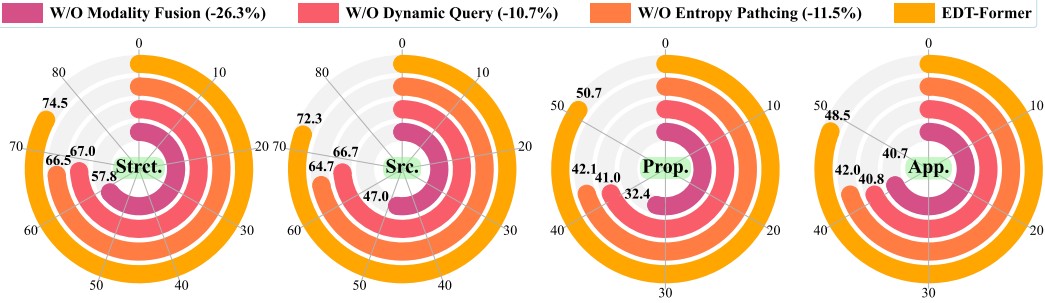

Figure 5: Ablation study of components on the MoleculeQA dataset. Accuracy is reported across four task types (Structure, Source, Property, and Application) when removing each component (modality fusion, Entropy-Guided Patching, or Dynamic Query Transformer).

Table 8: Ablations on patching strategies. Four methods are tested on the BBBP and Pampa. Accuracy (%) and F1 (subscript) are reported, with the top and second best highlighted.

| Methods | BBBP | Pampa | Avg. Drop |
|---|---|---|---|
| Entropy | $75.06_{75.06}$ | $84.52_{91.61}$ | 0% |
| BRICS | $73.59_{84.74}$ | $71.90_{83.09}$ | 3.96% |
| Random | $68.90_{80.13}$ | $66.67_{79.32}$ | 8.81% |
| None | $39.67_{49.58}$ | $78.62_{87.90}$ | 21.60% |

Table 9: Effect of EDT-Former on different LLM backbones. Four novel LLMs are tested on the MoleculeQA benchmark. Total accuracy and average gain for each model are reported.

| LLMs | LLM Only | +EDT-Former | Avg. Gain |
|---|---|---|---|
| Mistral-8B | 28.00 | 55.63 | +98.71% |
| Qwen3-8B | 27.18 | 54.79 | +101.58% |
| Llama2-7B | 38.37 | 65.89 | +71.72% |
| Llama3.1-8B | 49.31 | 68.34 | +38.59% |

erty, and Application. In *w/o Modality Fusion*, the Dynamic Query Transformer is replaced by standard transformer blocks without graph–text–3D alignment; in *w/o Dynamic Query*, we keep the 2D/3D encoders and anchors but substitute the dynamic tokens with a fixed-length projection of graph embeddings; in *w/o Entropy Patching*, entropy-guided segmentation is replaced by uniform 3-token SMILES patches. Removing multimodal fusion produces the largest degradation (26% avg.), confirming that combining text, graph, and 3D is critical for LLM comprehension. Disabling either Dynamic Query Transformer or Entropy-Guided Patching yields more than 10% average drops (10.7% and 11.5%), indicating that dynamic queries are necessary to handle variable-sized graphs and capture more evidence than fixed-length fusion, while entropy-driven patches expose LLM-salient subgraphs for stronger structure-aware understanding (**Q2, Q3**).

**Patching Strategies.** We compare four segmentation methods on PAMPA/BBBP with matched splits, prompts, and query budgets: Entropy-Guided Patching (entropy peaks), BRICS-based fragments (Jinsong et al., 2024), random patches (shuffle from BRICS), and no patching. As shown in Tab. 8, entropy-guided patching yields the strongest results, BRICS trails slightly (chemically sound segmentation) while random patching degrades more (misses the true subgraph boundaries), and removing patching produces the largest drop, confirming that learned, data-driven boundaries are essential. Entropy peaks from a small next-atom transformer mark LLM-salient transitions, exposing true substructure boundaries to the query interface; this data-driven segmentation likely generalizes beyond molecules and is necessary to realize multimodal fusion gains (**Q2**).

**Backbone Sensitivity.** On MoleculeQA with official splits, attaching our connector to Mistral, Qwen3, and Llama series models yields consistent improvements over the text-only backbones (Tab. 9). Gains are larger for weaker backbones but remain substantial across all four.

**Efficiency Analysis.** As shown in Tab. 10, relative to LoRA finetuning, our strategy halves GPU memory and achieves ~3.5× faster training per step under identical settings; full backbone tuning is practically infeasible in our setting. These support that efficient alignment is achievable without updating the LLM (**Q4**).

Table 10: Estimated memory usage and training time per step for EDT-Former with Llama3.1-8B.

| EDT-Former | Llama3.1-8B | Mem (GB) | Time/Step (s) |
|---|---|---|---|
| Train | Frozen | 37 | 0.26 |
| Train | LoRA | 77 | 0.93 |
| Train | Train | >200 | – |

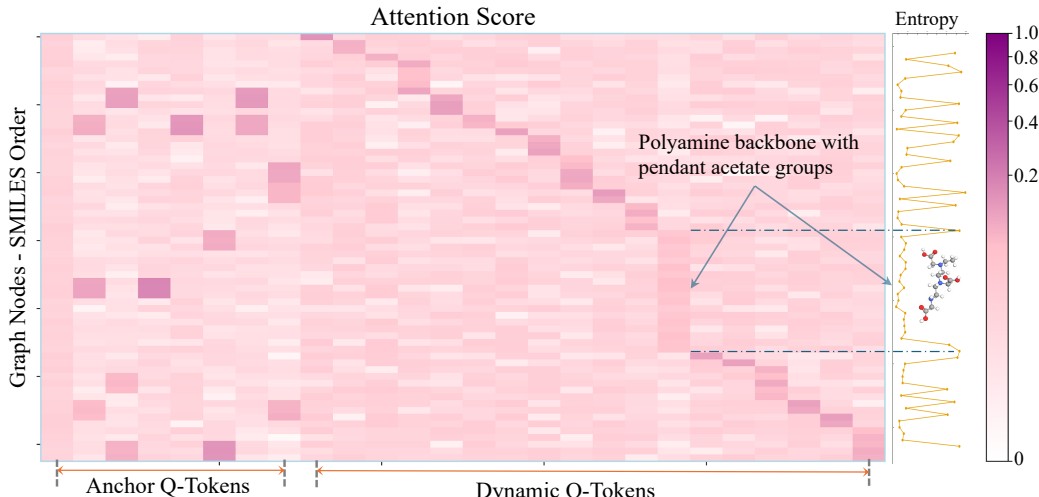

Figure 6: Attention visualization for EDT-Former. Last-layer attention map for a single molecule ($N$=63 atoms) with atoms in SMILES order along the vertical axis and query tokens (fixed-length anchors followed by dynamic tokens) along the horizontal axis. The right panel shows the corresponding entropy signal for the SMILES sequence. Values indicate normalized attention scores.

Table 11: Modality ablation on MoleculeQA. Four modality combinations are tested on four tasks, with the accuracies (%) reported. The top (pink) and second-best (lightpink) results are highlighted.

| Modality | Structure | Source | Property | Application | Average | Total | Avg. Drop |
|---|---|---|---|---|---|---|---|
| Text/Graph/3D | 74.55 | 72.39 | 50.71 | 48.58 | 61.56 | 68.34 | 0.00% |
| Text/Graph | 70.84 | 68.72 | 49.22 | 46.51 | 58.82 | 65.11 | 4.73% |
| Text/3D | 66.50 | 64.77 | 42.17 | 42.02 | 53.87 | 60.48 | 11.50% |
| Text Only | 66.46 | 60.98 | 40.35 | 36.40 | 51.05 | 58.77 | 14.00% |

**Fusion Benefits.** In Tab. 11, four modality settings are evaluated on MoleculeQA. Text-only (SMILES) performs worst, underscoring the need for explicit structural features. Removing 3D causes a modest decline relative to full fusion, whereas removing 2D (graph) degrades substantially, indicating that graph information is essential while 3D provides complementary gains.

**Attention Insights.** The attention visualization in Fig. 6 reveals the complementary roles for the two query types. Fixed-length tokens behave as modality anchors: their attention is diffuse and often spans multiple, unrelated patches, which is useful for global alignment but unreliable for isolating substructures. In contrast, dynamic tokens align with the SMILES-ordered entropy segments and attend sharply to contiguous node ranges that correspond to chemically meaningful subgraphs (e.g., the polyamine backbone with pendant acetate groups highlighted). These patterns substantiate **Q2** and **Q3**: multimodal fusion benefits when local structure is exposed, and the proposed novelties (Entropy-Guided Patching plus the Dynamic Query Transformer) provide a substructure-aware interface that preserves local fidelity while anchors maintain global consistency.

## 5 CONCLUSION

EDT-Former is introduced as a method for aligning molecular graphs with large language models. Entropy-Guided Patching segments molecules at entropy peaks to yield substructure-aware dynamic tokens, and Dynamic Query Transformer integrates these tokens with modality anchors to project into the LLM space, providing a stable interface without updating backbone parameters. Across standard benchmarks, EDT-Former delivers consistent gains over text-only and prior multimodal baselines while markedly reducing training cost relative to LLM tuning. Overall, EDT-Former offers chemically faithful, efficient graph–LLM alignment and a general recipe for multimodal fusion that can extend to broader graph domains; code and processed data are released for reproducibility, and for current constraints and future directions, see Apps. F.1 and F.2.

## 6 ETHICS STATEMENT

This work does not involve human subjects, personally identifiable information, or non-public/proprietary datasets. All data are publicly available under their respective licenses, and no wet-lab or animal experiments were conducted. Although molecular AI can present dual-use risks (e.g., aiding the design of harmful compounds), our contribution focuses on multimodal representation learning and evaluation on standard benchmarks; we neither train nor release models intended to generate or optimize novel toxic molecules. We will release code and models with terms of use that prohibit harmful or unlawful applications. All authors complied with institutional policies and responsible research practices.

## 7 REPRODUCIBILITY STATEMENT

This work is fully open-source under the MIT license. We release end-to-end resources to reproduce every result in the paper, including: (i) complete training and evaluation code (with all ablation scripts), (ii) baseline and benchmark runners with fixed seeds, (iii) processed pretraining and fine-tuning datasets with scripts to rebuild them from the original sources, (iv) model checkpoints, and (v) Dockerfile for bitwise-reproducible environments. A single-entry script reproduces all main tables and even the figures; expected variances from stochastic training are reported in the repository. No external services are required at evaluation time (offline scoring only). The repository is actively maintained and will serve as the foundation for subsequent extensions. All materials are available via the anonymous GitHub link: `https://anonymous.4open.science/r/EDT-Former-844D`.

As part of reproducibility, App. C.1 and App. C.2 detail the implementation of our two novelties and the pre-training settings. The fine-tuning settings and parameters are provided in App. C.3. As well, App. D.2 documents the data processing for each dataset. Finally, App. D.3 reports the hyperparameter choices and model configurations for each benchmark.

## 8 ACKNOWLEDGEMENTS

This work was supported in part by the Canada Research Chairs Tier II Program (CRC-2021-00482), the Canadian Institutes of Health Research (PLL 185683, PJT 190272, PJT204042, CFA - 205059), the Natural Sciences, Engineering Research Council of Canada (RGPIN-2021-04072, ALLRP 602759-24) and The Canada Foundation for Innovation (CFI) John R. Evans Leaders Fund (JELF) program (#43481).

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

APPENDIX TABLE OF CONTENTS

# A   THEORETICAL ANALYSIS AND PROOFS

**Scope and assumptions.**    The analyses in App. A.1 and App. A.2 are intended as proof sketches under standard modeling assumptions (calibrated next-token predictors, piecewise-stationary generation along SMILES order, and locally Lipschitz Transformer blocks on bounded inputs). We work with token-level surprisal $e_t = -\log p(a_{t+1} \mid a_{1:t})$ (and note that entropy refers to its expectation). The results formalize why (i) segmentation at surprisal peaks aligns with change points and minimizes a budgeted upper bound on pooling/modeling loss, and (ii) anchor-augmented projection yields a stable, low-rank–expressive interface to a frozen LLM. Constants and high-probability terms are suppressed for clarity.

A.1 ENTROPY PATCHING ANALYSIS

**Setup.** Let a molecule be represented by a SMILES-ordered atom sequence $(a_1, \ldots, a_T)$ generated by a piecewise-stationary conditional distribution $\mathcal{P}^\star(a_{t+1} \mid a_{1:t})$, with unknown change points $1 = \kappa_0 < \kappa_1 < \cdots < \kappa_M = T$ such that for $t \in [\kappa_{m-1}, \kappa_m - 1]$ the law of $a_{t+1} \mid a_{1:t}$ belongs to a stationary family $\mathcal{P}_m$. A trained next-atom predictor $f_\theta$ outputs a calibrated estimate $p_\theta(a_{t+1} \mid a_{1:t})$.[1] Define the surprisal (negative log-likelihood)

$$e_t = -\log p_\theta(a_{t+1} \mid a_{1:t}), \qquad t = 1, \ldots, T - 1.$$

Entropy-guided patching places a cut after indices in $T^\star = \{t : t \text{ is a local maximum of } e_t\}$, refined by non-maximum suppression (window $\Delta$) and a prominence threshold $\gamma$, yielding segments $S_k = \{t : \tau_{k-1} \leq t < \tau_k\}$ where $1 = \tau_0 < \cdots < \tau_M = T$ are the retained peaks.

**Objective.** Given a token budget (implicit in $\Delta, \gamma$) we seek a segmentation that (i) maximizes the information that segment tokens carry about the underlying structure, and (ii) minimizes within-segment heterogeneity that would be averaged out by pooling into a single token.

Let $X \in \mathbb{R}^{N \times d}$ be frozen node embeddings and let $\pi$ map SMILES indices to graph nodes. Each segment induces a node set $\widehat{S}_k = \{\pi(t) : t \in S_k\}$ and a token $z_k = \text{AvgPool}(\{X_i : i \in \widehat{S}_k\})$.

**Assumption 1 (Piecewise stationarity).** Within each true region $[\kappa_{m-1}, \kappa_m)$ the conditional $a_{t+1} \mid a_{1:t}$ has constant entropy $H_m = \mathbb{E}[e_t \mid t \in [\kappa_{m-1}, \kappa_m)]$ and adjacent regions satisfy a separation in total variation: $\text{TV}(\mathcal{P}_m, \mathcal{P}_{m+1}) \geq \delta > 0$.

**Assumption 2 (Calibration / bounded error).** $\big| \mathbb{E}[e_t] - H^\star(a_{t+1} \mid a_{1:t}) \big| \leq \varepsilon$ uniformly in $t$.

**Key quantity.** For a candidate segmentation $\mathcal{S} = \{S_k\}_{k=1}^M$, define the within-segment entropy

$$\mathcal{E}(\mathcal{S}) = \sum_{k=1}^M \frac{1}{|S_k|} \sum_{t \in S_k} \mathbb{E}[e_t].$$

This upper-bounds both (a) the Bayes error of predicting local structural events within segments via Fano-type inequalities, and (b) the information lost by averaging embeddings inside a segment (see Lemma 2).

**Lemma 1** (Surprisal peaks mark change points). *Under Assumptions 1–2, for any interior index $t$ that lies $\eta$ away from the nearest true change point, the expected discrete second difference satisfies $\mathbb{E}[e_{t+1} - 2e_t + e_{t-1}] \leq c_1 \varepsilon - c_2 \delta$ near a change point and $\geq -c_1 \varepsilon$ away from it, for constants $c_1, c_2 > 0$. Hence the probability that a local maximum of $e_t$ occurs within $O(\eta)$ of each change point tends to 1 as $\varepsilon \to 0$.*

*Sketch.* At a change point, the one-step law jumps from $\mathcal{P}_m$ to $\mathcal{P}_{m+1}$, inducing a kink in the log-likelihood path with magnitude controlled by $\text{TV}(\mathcal{P}_m, \mathcal{P}_{m+1})$. Calibrated predictors track this kink up to $\varepsilon$, yielding an excess in $e_t$ locally. Away from changes, stationarity implies $e_t$ is a martingale difference with bounded variance, suppressing spurious peaks in expectation. The discrete curvature argument yields the stated bounds. $\square$

**Lemma 2** (Pooling loss is entropy-controlled). *Let $g$ be any $L$-Lipschitz readout on sets of node embeddings (the self/cross-attention layer consuming $z_k$ is $L$-Lipschitz under standard assumptions). Then for each segment $S_k$,*

$$\mathbb{E}\Big[\big\| g(\{X_i : i \in \widehat{S}_k\}) - g(\{\bar{X}_k\}^{|S_k|}) \big\| \Big] \leq C \sqrt{\text{Var}(X \mid S_k)} \leq C' \sqrt{\mathcal{E}(S_k)},$$

*where $\bar{X}_k$ is the mean in the segment and the final inequality follows because variation of embeddings within a segment is controlled by variation of the local next-atom law, which is upper-bounded (up to constants) by the average surprisal in that segment.*

---

[1] Calibrated means $\mathbb{E}[\mathbf{1}\{A = a\} \mid p_\theta(A = a \mid \cdot) = u] = u$; consistency is not required, but strengthens the results.

**Proposition 1** (Peak cutting minimizes an upper bound on loss). *Among all segmentations with a given minimum spacing $\Delta$ (token budget), selecting cuts at the most prominent local maxima of $e_t$ (with NMS window $\Delta$) minimizes $\mathcal{E}(\mathcal{S})$ up to $O(\varepsilon)$, hence minimizes the stated upper bounds on both Bayes error within segments and pooling-induced representation loss.*

*Sketch.* Because $H_m$ is (approximately) constant inside each stationary region, any cut placed within a region increases $\mathcal{E}(\mathcal{S})$ only by splitting a constant-entropy block, while a cut placed at a change point prevents averaging across different entropies $H_m \neq H_{m+1}$ and thus strictly decreases $\mathcal{E}(\mathcal{S})$ by an amount $\Omega(\delta)$ (Assumption 1). Lemma 1 guarantees that the largest local maxima of $e_t$ concentrate at change points, so NMS over a spacing $\Delta$ selects (approximately) one cut per change region, which is optimal under a fixed budget. Lemma 2 transfers the bound from $\mathcal{E}$ to the representation loss after pooling. $\square$

**Corollary 1** (Adaptive token count matches structural complexity). *Let $K_\gamma$ be the number of retained peaks after prominence $\gamma$ and window $\Delta$. Then $\mathbb{E}[K_\gamma]$ is non-decreasing in the total variation budget $\sum_m \text{TV}(\mathcal{P}_m, \mathcal{P}_{m+1})$ and non-increasing in $(\Delta, \gamma)$, yielding a token count that adapts to molecular complexity while remaining budget-controlled.*

**Robustness and determinism.** The algorithm is deterministic given $(\Delta, \gamma)$, linear in $T$ for entropy and segmentation, and stable to predictor noise: if $\varepsilon < c\,\delta$ (Assumption 2), the selected peaks coincide with true change neighborhoods with high probability; otherwise NMS and prominence thresholding suppress spurious bumps so that $\mathcal{E}(\mathcal{S})$ remains within $O(\varepsilon)$ of the optimal budgeted value.

**Takeaway.** Entropy-guided patching is a principled change-point segmentation that (i) places boundaries where the conditional law changes (surprisal peaks), (ii) provably minimizes an entropy-based upper bound on within-segment error and pooling loss under a token budget, and (iii) adapts the number of dynamic tokens to structural complexity while remaining compute-efficient.

## A.2 DYNAMIC QUERY ANALYSIS

**Setup.** Let $Z = \{z_k\}_{k=1}^K$ be the variable-length substructure tokens from Entropy-Guided Patching, $A = \{a_j\}_{j=1}^Q$ be $Q$ modality anchors (fixed, learnable, or lightly tuned), and let $P \in \mathbb{R}^{d_{\text{in}} \times d_{\text{LLM}}}$ be a connector that linearly projects tokens into the frozen LLM embedding space. The input to the LLM is the concatenated sequence $\widetilde{X} = [\,PA\,;\,PZ\,] \in \mathbb{R}^{(Q+K) \times d_{\text{LLM}}}$. Denote the first Transformer block (self-attention + MLP) of the frozen LLM by $f : \mathbb{R}^{(Q+K) \times d_{\text{LLM}}} \to \mathbb{R}^{(Q+K) \times d_{\text{LLM}}}$, with attention maps parameterized by frozen matrices $(W_Q, W_K, W_V)$ and output projection $W_O$. Subsequent blocks are identical copies with frozen parameters.

**Assumption A1 (Lipschitz block).** Each LLM block is $L$-Lipschitz on a compact input set $\mathcal{X}$: $\|f(X) - f(Y)\| \leq L\|X - Y\|$ (this holds under standard bounded-activation/softmax assumptions).

**Assumption A2 (Anchor coverage).** Let $\mathcal{S}_A = \text{span}\{W_K PA\} \subset \mathbb{R}^{d_k}$ be the key subspace induced by anchors. There exists $\sigma_{\min} > 0$ such that the minimum singular value of $[W_K PA]$ is $\geq \sigma_{\min}$, and $\text{span}\{W_Q PA\} = \mathcal{S}_A$ as well (anchors expose a stable key/query frame).

**Assumption A3 (Bounded projection error).** Let $Z^\star \in \mathbb{R}^{K \times d_{\text{LLM}}}$ be an (unknown) ideal embedding that would produce the desired frozen-LLM behavior. The learned connector satisfies $\|PZ - Z^\star\| \leq \varepsilon$ on the data manifold.

**Stability via anchors.** Consider attention logits in the first block: $S = \widetilde{X} W_Q (W_K^\top \widetilde{X}^\top) / \sqrt{d_k}$. Anchors contribute structured rows/columns $S_{AA}$ and $S_{AZ}$ that do not depend on $Z$'s length or order. Because A2 ensures a well-conditioned anchor subspace $\mathcal{S}_A$, the Jacobian of attention w.r.t. $PZ$ is bounded, which yields an input-output Lipschitz bound uniform in $K$.

**Lemma 3** (Anchor-conditioned Lipschitz stability). *Under A1–A2, there exists $C = C(L, \sigma_{\min}, \|W_Q\|, \|W_K\|, \|W_V\|)$ such that for any two token sets $Z_1, Z_2$ (possibly different lengths) and the same anchors $A$,*

$$\left\| f([PA; PZ_1]) - f([PA; PZ_2]) \right\| \leq C \left\| PZ_1 - PZ_2 \right\|.$$

*Hence the per-block perturbation is controlled by the connector-space distance, independent of $K$.*

*Sketch.* Write the attention output as a composition of (i) bilinear logits $(\cdot)W_Q(W_K^\top \cdot^\top)$ and (ii) softmax-weighted value mixing. The anchor block $PA$ produces a key/query frame whose spectrum is bounded below by $\sigma_{\min}$ (A2), preventing ill-conditioning in the softmax sensitivity. Combine with A1 and standard Jacobian bounds for softmax to obtain the stated Lipschitz constant $C$. $\qquad\square$

**Expressivity vs. LoRA at the input.** Linearizing the first block at an operating point $\widetilde{X}_0 = [PA; PZ_0]$ gives $f(\widetilde{X}) \approx f(\widetilde{X}_0) + J_0(\widetilde{X} - \widetilde{X}_0)$, where $J_0$ is the Jacobian. A rank-$r$ LoRA update at the input embedding ($E \mapsto E + \Delta E$, $\text{rank}(\Delta E) \leq r$) induces an output shift $J_0 \Delta E$ of rank at most $r$. If $P$ is factored as $P = UV^\top$ with $\text{rank}(P) \leq r$, then $PZ$ can realize an equivalent family of rank-$r$ shifts in the first-block input, matching what a LoRA adapter would add at that boundary.

**Proposition 2** (Connector-only can emulate low-rank input adaptation). *Fix anchors $A$ and linearization point $\widetilde{X}_0$. For any rank-$r$ update $\Delta E$ to the input embeddings, there exists a rank-$r$ connector $P = UV^\top$ and a perturbation $\Delta Z$ such that $J_0([0; \Delta E]) = J_0([0; P\Delta Z])$. Hence, in the first-order regime, connector-only training with rank-$r$ $P$ is as expressive as a rank-$r$ LoRA at the input boundary.*

*Sketch.* Let $J_0$ define a linear map on inputs; match images by choosing $P\Delta Z = \Delta E$ in the column space that $J_0$ uses. Factor $P$ with rank $\leq r$ and select $\Delta Z$ in the corresponding $r$-dimensional coordinate system to reproduce $J_0 \Delta E$. $\qquad\square$

**Information preservation.** Let $\phi$ be the (frozen) LLM map from the first block to the layer used by the cross-attention interface (or to logits for analysis). Define the task-relevant mutual information $I(Z; \phi([PA; PZ]))$. Because $f$ is $L$-Lipschitz and the connector error is $\varepsilon$ (A3), data processing and stability imply only small information loss.

**Lemma 4** (Lower bound on retained information). *Under A1–A3 and mild regularity (bounded support and densities), there exists $\alpha > 0$ such that*

$$I(Z; \phi([PA; PZ])) \ \geq \ I(Z; \phi([PA; Z^\star])) - \alpha\,\varepsilon.$$

*Thus, if $P$ approximates the ideal embedding within $\varepsilon$, downstream information loss is $O(\varepsilon)$.*

*Sketch.* Combine Lemma 3 with standard continuity bounds for mutual information under Lipschitz perturbations (via Wasserstein stability or Pinsker-type inequalities). The $\alpha$ constant absorbs Lipschitz and measure-regularity terms. $\qquad\square$

**Budget awareness and interference control.** Let $\Pi_A$ be the orthogonal projector onto $\text{span}\{PA\}$ in the LLM embedding space. Adding an orthogonality penalty $\|\Pi_A(PZ)\|^2$ ensures that content tokens $PZ$ occupy a subspace complementary to anchors, reducing cross-attention interference and concentrating gradients on task-relevant directions.

**Proposition 3** (Conditioning and interference). *If training enforces (approximately) $\Pi_A(PZ) = 0$ and normalizes $PA$, then the attention Gram matrix on $\widetilde{X}$ has condition number bounded by a constant depending only on anchors. Consequently, gradient norms for $P$ are well-conditioned and do not deteriorate with token count $K$.*

*Sketch.* Under the orthogonality constraint, the mixed Gram blocks between anchors and content vanish at first order, yielding a block-structured Gram with bounded spectrum on the anchor block (A2) and controlled spectrum on the content block via normalization and Lipschitzness, hence a uniform bound on the condition number. $\qquad\square$

**Takeaways.** (i) Stability: anchors provide a well-conditioned key/query frame that makes the frozen LLM locally Lipschitz in the connector space, uniformly in token count (Lemma 3). (ii) Expressivity: a low-rank connector can emulate first-order effects of a low-rank (LoRA) input adapter (Proposition 2), explaining strong performance without tuning the backbone. (iii) Information retention: if $P$ approximates an ideal embedding, the downstream information loss is $O(\varepsilon)$ (Lemma 4). (iv) Good conditioning: orthogonalizing content against anchors controls attention conditioning and prevents interference (Proposition 3).

Overall, Dynamic Query Transformer yields a stable, expressive, and budget-aware interface to frozen LLMs, complementing Entropy-Guided Patching's adaptive tokenization.

## B  EXTENDED RELATED WORK

In addition to the discussion in Sec. 2, this section provides further analysis of how our approach relates to prior efforts. Specifically, we highlight the differences between our Dynamic Query Transformer and fixed-length Q-Former-style bridges, as well as connections to entropy-related works and recent multimodal molecular language models. This extended review situates our model within the broader landscape and clarifies its novel contributions.

### B.1  FROM REPRESENTATION TO MULTIMODAL UNDERSTANDING

**Representation models vs. language models.** Early molecular representation learning focused on embedding models that map graphs or sequences to vector spaces for downstream prediction, e.g., MolFormer (Wu et al., 2023), Uni-Mol/Uni-Mol-v2 (Zhou et al., 2023; Ji et al., 2024), and ChemBERTa-2 (Ahmad et al., 2022). These encoders excel at property classification and regression as they optimize contrastive and masked-token training over large chemical corpora, yielding stable, geometry-aware embeddings for downstream heads. However, they do not natively support natural-language generation (e.g. UniMoT (Zhang et al., 2024)) or multi-hop reasoning (Jang et al., 2024); in these settings, evidence remains implicit in continuous vectors, and exact numeric targets are better served by calibrated regressors than autoregressive tokens.

**From generation to reasoning with LLMs.** LLM-based molecular systems provide a natural-language interface for explanation, instruction following, and tool use, enabling generative descriptions and stepwise reasoning (Fang et al., 2023; Zhao et al., 2023; Zhang et al., 2024). Yet vanilla LLMs struggle with exact property prediction due to the mismatch between discrete token likelihoods and calibrated real-valued targets, unit/scale sensitivity, and exposure bias during numeric generation; recent work addresses these issues with instruction data, tool-augmented agents, and chain-of-thought methods tailored to molecular structure (Kim et al., 2025; Bran et al., 2023; Jang et al., 2024). Overall effectiveness hinges on how molecular structure is exposed to the language model–whether via tokenized geometry, graph-aware connectors, or substructure tokens.

**Multimodal fusion for molecular generative modeling.** Beyond SMILES-only models (e.g., KV-PLM (Zeng et al., 2022)), recent work fuses SMILES, molecular graphs, and 3D conformers with LLMs. Instruction-tuned systems such as GIMLET (Zhao et al., 2023) show that paired graph-text supervision imparts basic molecular understanding; larger frameworks (e.g., MoleculeSTM (Liu et al., 2023), MolFM (Luo et al., 2023a), BioT5+ (Pei et al., 2024a), ProLLaMA (Lv et al., 2025)) broaden modality and biomedical knowledge coverage; and 3D-aware models (e.g., 3D-MolT5 (Pei et al., 2024b)) introduce coarse geometry for conditional generation. More recent graph–LLM connectors (e.g., Mol-LLM (Lee et al., 2025), Mol-LLaMA (Kim et al., 2025)), and UniMolT (Zhang et al., 2024) adapt the BLIP-2/Q-Former (Li et al., 2023) paradigm with fixed-length, learnable query tokens. Although effective for alignment, fixed-length fusion compresses stereochemistry and substructures, resulting in structural information loss that worsens with increasing molecular size and conformer diversity; this motivates the development of dynamic and substructure-aware interfaces that present structures more faithfully to the LLM.

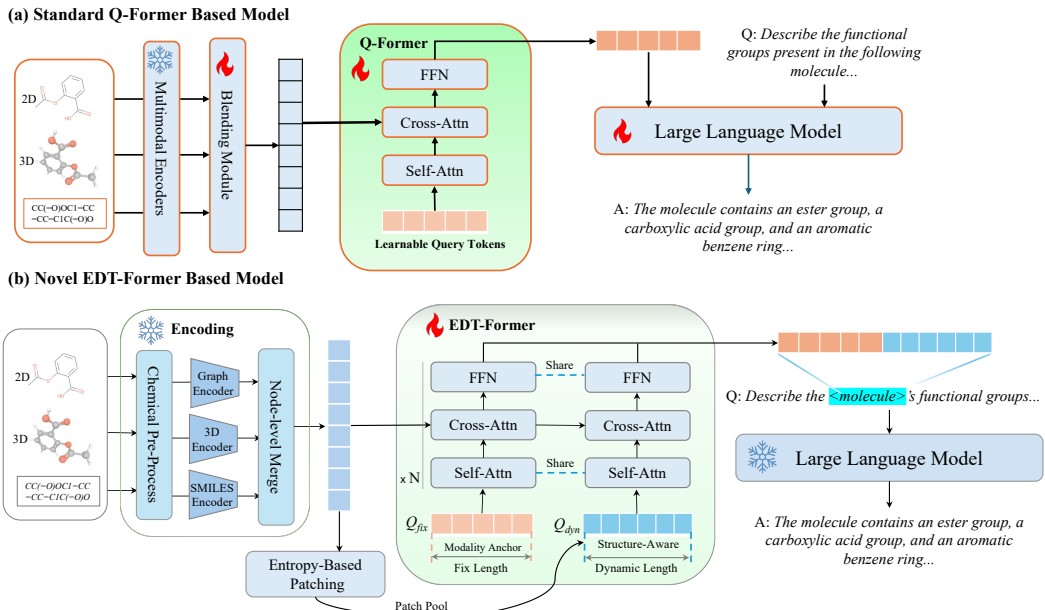

Figure 7: Architecture comparison between Q-Former-based model and EDT-Former. (a) Standard Q-Former connector that uses fixed-length learnable query tokens as modality anchors. (b) Novel EDT-Former connector that combines fixed-length tokens and entropy-guided structure-aware dynamic query tokens.

## B.2 RELATIONSHIP TO RELATED WORKS

**Relationship to Q-Former.** BLIP-2 introduces a lightweight Querying Transformer (Q-Former) that uses a fixed set of learnable query tokens to extract visual evidence from a frozen image encoder via cross-attention, then projects the resulting tokens to a frozen LLM—enabling efficient vision–language alignment without updating the backbones (Li et al., 2023). The Q-Former is pretrained with vision–language objectives (e.g., contrastive/alignment and generative stages) to ensure its queries attend to informative regions and produce LLM-consumable embeddings. Subsequent systems (e.g., InstructBLIP (Dai et al., 2023)) keep the same mechanism but condition the Q-Former on textual instructions, improving instruction following while retaining the fixed query budget. Related adapters such as Flamingo's (Alayrac et al., 2022) Perceiver-Resampler also distill high-dimensional visual features to a small, fixed-length token set for a frozen LM.

As shown in Fig. 7, Dynamic Query Transformer differs in two aspects. 1) Modality. Molecules are variable-sized graphs (and conformers), where structural fidelity hinges on preserving substructures (stereochemistry, functional groups), not just salient image regions. 2) Interface. Instead of a fixed query budget, EDT-Former upgrades the Q-Former's fixed queries with **entropy-guided dynamic queries**. Concretely, Entropy-Guided Patching discovers entropy-peak–based segments and pools node embeddings into dynamic tokens, and Dynamic Query Transformer refines these tokens (with a small set of anchors) via self-/cross-attention before projection to the LLM space. This design preserves Q-Former–style efficiency (frozen backbones, a lightweight adapter) while improving graph-specific fidelity by matching the number and placement of queries to molecular structure, thereby enabling multimodal molecular models without fine-tuning the LLM backbone (excluding the embedding layer). In short, we adopt the Q-Former rationale—frozen encoders bridged by a compact adapter—but move beyond fixed-length queries to a dynamic structure-aligned interface tailored to graphs.

**Relationship to BLT (Byte Latent Transformer).** BLT groups raw bytes into dynamic variable-length patches using the entropy of the next byte, so that higher-entropy regions receive more compute; a latent transformer then operates over these patches and projects to a language model, yielding tokenizer-free efficiency and robustness (Pagnoni et al., 2025). Our approach draws inspiration from the entropy-driven segmentation principle, but targets a different modality and objective: molecules

are variable-sized graphs where fidelity depends on preserving substructures and stereochemistry. Concretely, we estimate next-atom uncertainty over SMILES to place peak-based cut points, map segments to graph nodes, and pool node embeddings into dynamic substructure tokens. These tokens are integrated (with a small set of anchors) via self-/cross-attention and projected to a frozen LLM, aligning query number and placement to molecular structure. Thus, while BLT's entropy patching motivates adaptive granularity, our entropy definition, boundary rule, and graph-aware interface are tailored to structural chemistry rather than byte-level text.

**Comparison to other molecular LLMs.** (1) Mol-Instructions (Fang et al., 2023). Mol-Instructions establishes large-scale instruction data for molecules and fine-tunes general LLMs (e.g., Llama-2/3 (Touvron et al., 2023b; MetaAI, 2024)) from SMILES inputs to demonstrate the value of instruction supervision for captioning and property-related responses. We adopt its benchmarks and reported baselines for our evaluations on molecular description and property prediction. While effective as a single-modality (text/SMILES) approach, it does not expose explicit structural signals (graphs/3D), which limits structure-aware reasoning beyond what can be inferred from tokenized strings. (2) Mol-LLaMA (Kim et al., 2025). Mol-LLaMA bridges molecular structure and language using a Q-Former–style connector: a fixed set of learnable query tokens cross-attends to strong encoders and conditions an LLM. This design proves the feasibility of graph–LLM alignment but inherits the fixed-length bottleneck, which compresses substructures and stereochemistry as molecular size and conformer diversity grow. In contrast, our method retains the lightweight adapter philosophy while replacing fixed queries with dynamic entropy-guided substructure tokens refined by a dynamic query transformer. For completeness, we employ recognized 2D/3D encoders (e.g., MoleculeSTM (Liu et al., 2023), Uni-Mol (Zhou et al., 2023)) and comparable processed training data when reproducing baselines; these choices standardize comparisons but are orthogonal to the core novelty.

**Other graph-text alignment works.** Recent frameworks such as MoleculeSTM (Liu et al., 2023) and FineMolTex (Li et al., 2025) learn new multimodal encoders from scratch via large-scale structure–text contrastive training and motif-level masking on curated molecule–caption pairs, targeting strong standalone representations for retrieval and editing without relying on a frozen LLM. In contrast, EDT-Former adopts a connector-only paradigm: we freeze both the molecular encoder and the LLM backbone (excluding the embedding layer), and learn a lightweight bridge that uses entropy-guided dynamic SMILES patches plus a few anchors to adapt the token budget to molecular complexity. These approaches are complementary—such pretrained encoders could serve as stronger molecular backbones, while EDT-Former focuses on efficiently aligning them with instruction-tuned LLMs for QA, property prediction, and reaction-centric generation.

**Inspiration from graph and fusion embedding models.** Our previous work MuMo (Jing et al., 2026) is a multimodal embedding model for sequences, graphs, and 3D structures, evaluated across a comprehensive suite of molecular tasks. We also drew broader insights on graph modeling and transfer techniques from graph-related works: (Fang et al., 2022), (Fang et al., 2025a), (Fang et al., 2025b), (Fang et al., 2026a), (Fang et al., 2026b).

## C    IMPLEMENTATION DETAILS

This section details the implementation for reproducibility. It begins with the entropy-guided patching algorithm, including entropy computation, peak detection, the next-token predictor, and a brief rationale recap. Attention computations for integrating dynamic tokens with anchors are then formalized. Configurations of EDT-Former, covering both graph encoders and LLM backbones, are described next. The section concludes with pretraining settings (objectives, prompts, and hyperparameters) and finetuning protocols (setups, prompts, and representative inference examples).

### C.1    ENTROPY-GUIDED PATCHING

**Algorithm of Entropy-Guided Patching.** Algorithm 2 segments a SMILES sequence into dynamic substructures using an entropy signal from a lightweight next-atom predictor (NAP). For

Table 12: Configurations and Training settings for the Next-Atom Prediction (NAP) model, architecture, hyperparameters, together with the resulting compute footprint and parameter count.

| Configurations | | Pre-Training | |
|---|---|---|---|
| Item | Value | Item | Value |
| Vocabulary size | 39 | Learning rate | $1 \times 10^{-4}$ |
| Model architecture | GPT-2LMHead | Batch size | 64 |
| Maximum position embeddings | 1024 | Training epochs | 1 |
| Context window | 512 | Training steps | 4,929 |
| Transformer layers | 2 | Weight decay | 0.01 |
| Attention heads per layer | 2 | Total FLOPs | $7.5395 \times 10^{13}$ |
| Hidden size (embedding dimension) | 128 | Model size | 0.538M |
| Activation function | GELU | Dataset | Pubchem |
| Dropout - residual connections | 0.10 | Max Length | 128 |
| Dropout - token/position embeddings | 0.10 | Concat Data | True |
| Dropout - attention | 0.10 | Precision | FP16 |
| LayerNorm epsilon | $1 \times 10^{-5}$ | Objective | NAP |

Table 13: Molecular atom vocabulary from SMILES grouped by category for the NAP (Next-Atom Prediction) model, with per-category counts (total tokens = 39, including 3 special tokens).

| Category | Symbols | Count |
|---|---|---|
| Special tokens | `<pad>`, `<bos>`, `<eos>` | 3 |
| Halogens | F, Cl, Br, I | 4 |
| Noble gases | He, Ne, Ar, Kr, Xe | 5 |
| Alkali metals | Li, Na | 2 |
| Alkaline earth metals | Mg, Ca | 2 |
| Transition metals | Fe, Co, Ni, Cu, Zn, Ag, Au, Pd, Pt, Mn, Hg | 11 |
| Post-transition metals | Al, Sn | 2 |
| Metalloids | B, Si, Sb, Te | 4 |
| Other nonmetals | C, N, O, P, S, Se | 6 |
| Total | | 39 |

each position, the predictor provides the probability of the ground-truth next atom; its negative log-probability defines the entropy trace. Local maxima are detected with a prominence threshold and non-maximum suppression, and cuts are placed after the retained peaks. Segments are mapped from SMILES indices to graph nodes and average-pooled over frozen node embeddings to form dynamic substructure tokens. The procedure is simple, deterministic given $(\Delta, \gamma)$, linear in sequence length for entropy and segmentation, and produces structure-aware tokens aligned with molecular complexity.

**The NAP model configuration.** As summarized in Tab. 12, the NAP model adopts a standard transformer architecture with a maximum input context of 512 tokens. This context window bounds the effective sequence length used during training and inference. The SMILES atom vocabulary shows in Tab. 13) comprises 39 tokens and spans major chemical groups–halogens, noble gases, alkali/alkaline earth metals, transition/post-transition metals, metalloids, and other nonmetals–providing broad elemental coverage for next-atom prediction.

**Pre-training of NAP.** We train the NAP model with standard autoregressive next-token prediction over SMILES. Given a tokenized sequence $x_{1:T}$ (prepended with $\langle bos \rangle$ and optionally terminated by $\langle eos \rangle$), a causal Transformer parameterized by $\theta$ defines the left-to-right factorization

$$p_\theta(x_{1:T}) = \prod_{t=1}^{T} p_\theta(x_t \,|\, x_{<t}).$$ (8)

---

**Algorithm 2:** Entropy-Guided Patching

---

**Input** : SMILES atom sequence $(a_1, \ldots, a_T)$; trained next-atom predictor $f_\theta$; node embeddings $X \in \mathbb{R}^{N \times d}$ (frozen graph encoder); map $\pi$ (SMILES$\rightarrow$graph); NMS window $\Delta$; prominence $\gamma$.
**Output:** Dynamic tokens $\{z_k\}_{k=1}^M$; SMILES segments $\{\mathcal{S}_k\}_{k=1}^M$.

**Entropy computation:**;
**for** $t \leftarrow 1$ **to** $T-1$ **do**
$\quad L_t \leftarrow f_\theta(a_{1:t})$ ;                    `// logits over atom vocab` $\mathcal{V}$
$\quad p_t \leftarrow \text{softmax}(L_t)[a_{t+1}]$ ;        `// prob. of ground-truth next atom`
$\quad e_t \leftarrow -\log p_t$ ;              `// information content (nats);` $O(T)$ `total`

**Peak detection (local maxima + prominence):**;
$\mathcal{T}_\star \leftarrow \emptyset$ ;                        `// candidate split indices`
**for** $t \leftarrow 2$ **to** $T-2$ **do**
$\quad \text{isLocalMax} \leftarrow (e_{t-1} < e_t) \wedge (e_t > e_{t+1})$ ;              `// strict peak`
$\quad \text{prominent} \leftarrow \left(e_t - \frac{1}{2}(e_{t-1} + e_{t+1}) \geq \gamma\right)$ ;      `// remove shallow bumps`
$\quad$ **if** isLocalMax $\wedge$ prominent **then**
$\quad\quad \lfloor \mathcal{T}_\star \leftarrow \mathcal{T}_\star \cup \{t\}$ ;                        `// keep peak` $t$

**Non-maximum suppression (window** $\Delta$**):**;
Sort $\mathcal{T}_\star$ by descending $e_t$ to get $(t^{(1)}, t^{(2)}, \ldots)$ ;          `// highest entropy first;`
$O(|\mathcal{T}_\star| \log |\mathcal{T}_\star|)$
$\mathcal{T} \leftarrow \emptyset$ ;                                    `// final peak set`
**foreach** $u \in (t^{(1)}, t^{(2)}, \ldots)$ **do**
$\quad$ **if** $\forall v \in \mathcal{T}, |u-v| > \Delta$ **then**
$\quad\quad \lfloor \mathcal{T} \leftarrow \mathcal{T} \cup \{u\}$ ;                        `// suppress neighbors within` $\Delta$

**Segmentation (cut after peaks):**;
Set $1 = \tau_0 < \tau_1 < \cdots < \tau_{M-1} < \tau_M = T$ with $\{\tau_1, \ldots, \tau_{M-1}\} = \mathcal{T}$ ;
`// deterministic cuts`
**for** $k \leftarrow 1$ **to** $M$ **do**
$\quad \mathcal{S}_k \leftarrow \{t \mid \tau_{k-1} \leq t < \tau_k\}$ ;                    `// SMILES index segment`
$\quad \widehat{\mathcal{S}}_k \leftarrow \{\pi(t) : t \in \mathcal{S}_k\}$ ;                        `// map to graph nodes`

**Pooling to dynamic tokens (avg-pool):**;
**for** $k \leftarrow 1$ **to** $M$ **do**
$\quad z_k \leftarrow \text{AvgPool}\left(\{X_i : i \in \widehat{\mathcal{S}}_k\}\right) \in \mathbb{R}^d$ ;  `// substructure token; linear in`
$\quad |\widehat{\mathcal{S}}_k|$
**return** $\{z_k\}_{k=1}^M$, $\{\mathcal{S}_k\}_{k=1}^M$ ;                `// dynamic structure-aware tokens`

---

We optimize the negative log-likelihood (cross-entropy) under teacher forcing, averaged over non-padding tokens:

$$\mathcal{L}(\theta) = -\frac{1}{\sum_{t=1}^T m_t} \sum_{t=1}^T m_t \log p_\theta(x_t \mid x_{<t}), \tag{9}$$

where $m_t \in \{0, 1\}$ masks out $\langle \texttt{pad} \rangle$ (and any positions beyond the context window). In practice, logits are computed over the 39-token vocabulary; optional label smoothing can be applied by replacing the one-hot targets with a smoothed distribution. During inference, we sample (or decode greedily) from $p_\theta(\cdot \mid x_{<t})$ until $\langle \texttt{eos} \rangle$ or the 512-token context limit is reached.

**Computing costs analysis of NAP model.** Using the settings in Tab. 12 on a single RTX 3090, end-to-end training lands in the "single-digit minutes" range (well below 0.1 GPU-hours for a full pass), so the cost is effectively negligible. The model is tiny ($\approx 0.54$ M parameters) and can also be trained comfortably on a modern multi-core CPU within an hour, avoiding the need for a GPU altogether. Compared with a typical 8B-parameter model, this NAP setup is $\approx 150\times$ smaller and

Table 14: Configurations and pre-training settings of Dynamic Query Transformer. The model configuration is based on SciBERT and trained from scratch.

| Item | Value | Item | Value |
|---|---|---|---|
| Base config | SciBERT | Embedding dimension | 512 |
| Max dynamic queries | 64 | Anchor queries | 16 |
| Max input tokens | 594 | Batch size | 48 |
| Attention heads | 8 | Transformer layers | 8 |
| Training epochs | 10 | Total steps | 15,599 |
| Precision | bf16-mixed | Scheduler | cosine_lr |
| Initial learning rate | $1.0 \times 10^{-4}$ | Minimum learning rate | $1.0 \times 10^{-5}$ |
| Warmup learning rate | $1.0 \times 10^{-6}$ | Warmup steps | 500 |
| Weight decay | 0.05 | Temperature | 0.10 |
| Top-$k$ | 50 | Top-$p$ | 1.0 |
| Repetition penalty | 1.0 | Accumulate grad batches | 1 |

$\approx 1000\times$ faster for both training and inference, so it does not increase the training or inference budget.

Entropy peaks from the NAP predictor identify structural transition points in SMILES; cutting at these peaks yields data-driven substructures and dynamic tokens that preserve locality and align cleanly with the language stream–simple, rule-free, and robust under a frozen backbone. The pipeline adds negligible overhead and does not materially affect training or inference costs.

## C.2 DYNAMIC QUERY TRANSFORMER

**Modeling Details with Anchors and Dynamic Tokens.** We keep standard Transformer machinery and only specify the routed, two-stream updates. Let $Q_{\text{fix}}^{(\ell)} \in \mathbb{R}^{k \times d}$ (anchors) and $Z^{(\ell)} \in \mathbb{R}^{M \times d}$ (dynamic tokens).

$$Q_{\text{fix,sa}}^{(\ell)} = Q_{\text{fix}}^{(\ell)} + \text{SelfAttn}_{\text{fix}}\big(Q_{\text{fix}}^{(\ell)}, Z^{(\ell)}\big), \tag{10}$$

$$Z_{\text{sa}}^{(\ell)} = Z^{(\ell)} + \text{SelfAttn}_Z\big(Q_{\text{fix}}^{(\ell)}, Z^{(\ell)}\big). \tag{11}$$

With graph node embeddings $X \in \mathbb{R}^{N \times d}$, each stream retrieves evidence independently:

$$\widehat{Q}_{\text{fix}}^{(\ell)} = Q_{\text{fix,sa}}^{(\ell)} + \text{CrossAttn}_{\text{fix}}\big(Q_{\text{fix,sa}}^{(\ell)}, X\big), \tag{12}$$

$$\widehat{Z}^{(\ell)} = Z_{\text{sa}}^{(\ell)} + \text{CrossAttn}_Z\big(Z_{\text{sa}}^{(\ell)}, X\big). \tag{13}$$

A shared FFN is applied to each stream separately:

$$Q_{\text{fix}}^{(\ell+1)} = \widehat{Q}_{\text{fix}}^{(\ell)} + \text{FFN}\big(\widehat{Q}_{\text{fix}}^{(\ell)}\big), \qquad Z^{(\ell+1)} = \widehat{Z}^{(\ell)} + \text{FFN}\big(\widehat{Z}^{(\ell)}\big). \tag{14}$$

After $L$ layers, project to the LLM space and concatenate to form the conditioning sequence:

$$U = \big[ Q_{\text{fix}}^{(L)} W_{\text{proj}} ; Z^{(L)} W_{\text{proj}} \big] \in \mathbb{R}^{(k+M) \times d_{\text{LLM}}}. \tag{15}$$

**Configurations and pre-training settings**. We pretrain the Dynamic Query Transformer bridge with a fixed anchor/dynamic query budget, moderate-depth Transformer, and a standard mixed-precision schedule with warmup, cosine decay, and weight decay; full hyperparameters and decoding settings are reported in Tab. 14.

**Pre-training objectives.** The Dynamic Query Transformer is pre-trained with three complementary objectives on Mol-Llama-Instruct dataset (Kim et al., 2025): (1) cross-modal contrastive alignment to pull together representations of the same molecule across modalities and push apart different molecules; (2) modality–matching to make fixed anchors decode the modality identity, enforcing a

shared semantic interface; and (3) masked substructure reconstruction to teach dynamic tokens to preserve fine-grained structural content.

*(1) Cross-modal contrastive loss.* For a mini-batch $\mathcal{B}$, and modalities $m$ and $m'$, we use an InfoNCE objective

$$\mathcal{L}_{\text{contrast}} = -\frac{1}{|\mathcal{B}|} \sum_{(i,m) \in \mathcal{B}} \log \frac{\exp\big(\text{sim}(z^{(i,m)}, z^{(i,m')})/\tau\big)}{\sum_{j \in \mathcal{B}} \exp\big(\text{sim}(z^{(i,m)}, z^{(j,m')})/\tau\big)}, \tag{16}$$

where $z^{(i,m)} \in \mathbb{R}^d$ is the pooled query embedding of molecule $i$ in modality $m$, sim is cosine similarity, and $\tau$ is a temperature.

*(2) Modality-matching loss.* Fixed-length anchors $Q_{\text{fix}}$ must predict the modality identity via an $M$-way classification:

$$\mathcal{L}_{\text{match}} = \frac{1}{|\mathcal{B}|} \sum_{(i,m) \in \mathcal{B}} \text{CE}\big(\mathbf{W}_m^\top \bar{z}^{(i)}, m\big), \tag{17}$$

where $\bar{z}^{(i)}$ is the average of the $k$ anchor outputs for molecule $i$, $\mathbf{W}_m$ are classifier parameters, and CE is cross-entropy.

*(3) Masked substructure reconstruction loss.* Dynamic queries $Q_{\text{dyn}}$ are trained to recover masked fragments within each modality:

$$\mathcal{L}_{\text{recon}} = \frac{1}{|\mathcal{B}|\,M} \sum_{(i,m) \in \mathcal{B}} \text{CE}\big(f_{\text{dec}}^{(m)}(z^{(i,m)}), \text{ masked\_tokens}^{(i,m)}\big), \tag{18}$$

where $f_{\text{dec}}^{(m)}$ is a modality-specific decoder and $\text{masked\_tokens}^{(i,m)}$ are the targets.

The total objective is a weighted sum:

$$\mathcal{L}_{\text{total}} = \lambda_1\, \mathcal{L}_{\text{contrast}} + \lambda_2\, \mathcal{L}_{\text{match}} + \lambda_3\, \mathcal{L}_{\text{recon}}, \qquad \lambda_1, \lambda_2, \lambda_3 > 0. \tag{19}$$

## C.3 EDT-FORMER SETTINGS AND FINETUNING

After pre-training the Dynamic Query Transformer, the next stage is finetuning this connector with frozen LLMs.

**Selection of LLMs.** 9 After pre-training the Dynamic Query Transformer bridge, we fine-tune only the connector jointly with graph encoders and the LLM, while keeping both the encoders and the LLM frozen. We screen several 7–8B backbones, including Mistral-8B (Mistral AI, 2024), Qwen3-8B (Yang et al., 2025), Llama2-7B (Touvron et al., 2023b), and Llama3.1-8B (MetaAI, 2024)–covering strong base and reasoning-oriented models. As shown

Table 15: Fine-tuning dataset. Four types of instruction data from Mol-Llama-Instruct, the data types and amounts are listed.

| Category | Amount |
| --- | --- |
| Detailed Structural Descriptions | 77,239 |
| Structure-to-Chemical Features | 73,712 |
| Structure-to-Biological Features | 73,645 |
| Comprehensive Conversations | 60,147 |

in Tab. 9, Llama3.1-8B yields consistent performance on the MoleculeQA (Lu et al., 2024) benchmark. We therefore adopt Llama3.1-8B as the default backbone for subsequent experiments, as it provides the strongest raw molecular understanding and the best absolute performance once paired with our connector. Notably, EDT-Former delivers consistent gains across all LLM backbones, underscoring the effectiveness of our connector design and its robustness to different architectures. For the frozen backbone, we freeze the main layers and retain the embedding layers trainable to ensure a stable convergence process, as the graph tokens from the Dynamic Query Transformer are mapped into LLM's embedding space.

**Finetune configuration and settings.** The graph encoders and LLM backbone use the default settings released by the author (Zhou et al., 2023; Liu et al., 2023). The training settings of Dynamic Query Transformer list in the Tab 16.

**Training dataset and prompt examples.** As summarized in Tab. 15, our fine-tuning corpus from Mol-Llama-Instruct comprises four instruction types: (i) Detailed Structural Descriptions, that

Table 16: Fine-tuning settings for EDT-Former with frozen Llama-3.1-8B backbone.

| Item | Value | Item | Value |
|---|---|---|---|
| Precision | bf16-mixed | Epochs | 2 |
| Accumulate_grad_batches | 8 | Weight decay | 0.05 |
| Initial learning rate | $1.0 \times 10^{-4}$ | Minimum learning rate | $5.0 \times 10^{-6}$ |
| Warmup learning rate | $1.0 \times 10^{-6}$ | Warmup steps | 1,000 |
| Scheduler | linear_warmup_cosine_lr | Temperature | 0.10 |
| LoRA rank ($r$) | 8 | LoRA alpha | 32 |
| LoRA dropout | 0.10 | Trainable parameters | 213M |
| Total parameters | 8.3B | | |

---

**System**: "You are a helpful assistant specializing in chemistry and biology. The instruction that describes a task is given, paired with molecules. Write a response that appropriately completes the request."

*Structural Descriptions*

**User**: "Can you tell me about the basic structure of the given molecule?\nMolecule <mol>."

**Assistant**: "The given molecule is a complex carbohydrate or polysaccharide derivative that contains several monosaccharide units with multiple functional groups and modifications.\n\n1. **Core Structure**: The backbone is based on cyclic carbohydrates. Specifically, … "

*Structure-to-Chemical*

**User**: "Molecule <mol>.\nOffer a thorough analysis of the chemical characteristics of the compound."

**Assistant**: "The given molecule has a structure that combines several functional groups, imparting unique chemical properties to it. Centrally, it contains a propenone linkage, which features a carbon-carbon double bond conjugated with a carbonyl group. This conjugated system not only …"

*Structure-to-Biological*

**User**: "Offer an in-depth discussion of the biological traits of the molecule, specifically highlighting the relationship between the core parts of the molecule and its properties.\nMolecule <mol>."

**Assistant**: "The given molecule is a complex natural product found in Illicium verum, commonly known as star anise. The biological properties of this molecule are influenced by its structural features, which include several characteristic functional groups and substructures.\n\n### Tetracyclic Core Structure: …"

*Comprehensive Conversations*

**User**: "Can you tell me about the basic structure of the given molecule?\nMolecule <mol>."

**Assistant**: "Certainly! The given molecule is a lactone, more specifically, a five-membered cyclic ester. It includes a methylene group attached at the 3-position of the ring and two methyl groups at the 5-position.\n"

**User**: "What are the key functional groups present in the given molecule?"

**Assistant**: "The key functional group in the given molecule is the lactone, which is a cyclic ester. In addition, there's an exocyclic methylene group at the 3-position.\n"

Figure 8: Prompt setup and instruction data examples of EDT-Former in the finetuning stage with backbone frozen. One instance is shown for each data type; the comprehensive conversation category contains multiple rounds (exceeding ten).

elicit atom-/substructure–level narratives of a molecule; (ii) Structure-to-Chemical Features, linking structure to physicochemical properties; (iii) Structure-to-Biological Features, covering bioactivity/ADMET–style attributes; and (iv) Comprehensive Conversations for multi-turn reasoning and explanation. The dataset contains 284,743 instances in total, providing balanced coverage of descriptive, analytic, and conversational skills. Fig. 8 shows the prompts and examples of the finetuning data.

# D EXPERIMENTS DETAILS

Here, we provide additional experimental details to complement the main results. This section first describes the benchmarks and baseline systems used for comparison, followed by the evaluation setup (splits, prompts, metrics, and decoding). We then include representative response examples to illustrate typical model behavior, and then we report our data-contamination checks and screening procedures. Finally, we document reproducibility essentials–configurations and other necessary information to enable faithful re-runs.

## D.1 BENCHMARKS AND BASELINES

### BENCHMARKS

**MoleculeQA.** A large-scale multiple-choice QA benchmark for molecular factuality, where each instance pairs a human-written question with one correct option and three distractors from authoritative molecule descriptions (Lu et al., 2024). It covers four aspects: Structure (configuration, functional groups, backbone), Source (isolation, discovery/derivation, metabolites), Property (physicochemical/biological properties, safety, mechanisms), and Application (therapeutic/chemical uses, approvals, research/agricultural agents). Molecules are split by Bemis-Murcko scaffolds into train/dev/test sets of QA pairs as a total of 61,574. Aspect-wise counts are reported in Tab. 17.

Table 17: MoleculeQA split counts using Bemis–Murcko scaffold-based splitting at the molecule level to avoid scaffold overlap across splits.

| Category | Train / Valid / Test |
| --- | --- |
| Overall | 49,993 / 5,795 / 5,786 |
| Structure | 32,176 / 3,314 / 3,113 |
| Property | 4,838 / 698 / 731 |
| Application | 1,917 / 558 / 599 |
| Source | 11,062 / 1,225 / 1,343 |

**Mol-Instructions.** Mol-Instructions (Fang et al., 2023) is a large-scale instruction-tuning corpus for the biomolecular domain in instruction–response format for LLM supervision. Each example couples a natural-language prompt with target text for molecule-centric skills. We focus on seven tasks: six molecule-oriented tasks and one open question task. The corpus spans over two million instructions and includes the captioning and property instruction sets used in our fine-tuning.

**Molecular Property Prediction.** We use ten ADME-focused datasets from Therapeutics Data Commons (TDC) (Huang et al., 2021) and MoleculeNet (Wu et al., 2018). For instance, BBBP (Blood-Brain Barrier Penetration) is a binary classification task predicting a compound's ability to cross the blood–brain barrier. PAMPA (Parallel Artificial Membrane Permeability Assay) predicts passive membrane permeability from an in vitro surrogate of oral absorption. Both datasets are standardized in TDC for model development and fair comparison.

### BASELINES

**General Language Models.** We compare against strong general LLMs spanning proprietary and open releases: GPT-4o (OpenAI, 2023) and GPT-5 (OpenAI, 2025) as OpenAI's flagship multimodal and next-generation reasoning models. From Meta, we include Llama2-7B and Llama3.1-8B, widely used open backbones for downstream fine-tuning and dialogue (Touvron et al., 2023b; MetaAI, 2024). We also consider domain/community models: Galactica for science-centric pretraining (Taylor et al., 2022), BLOOM as a large multilingual open model (Workshop, 2022), and Pythia for scaling/analysis studies (Biderman et al., 2023). Finally, we include popular instruction-tuned chat models derived from LLaMA/Llama-2 (Vicuna-v1.5 (Zheng et al., 2023), Alpaca-7B (Taori et al., 2023), Baize-7B (Xu et al., 2023)) as well as Qwen3-8B as a recent multilingual/reasoning baseline Yang et al. (2025).

**Molecular Language Models.** For molecular-domain LLMs spanning datasets, graph/3D alignment, and reasoning, Mol-Instructions (Fang et al., 2023) provides a large instruction–response corpus tailored to molecules, while trained Llama-based Mol-Instructions model themselves. Mol-Llama (Kim et al., 2025) targets general molecular understanding via multi-modal instruction tuning. 3D-MoLM (Li et al., 2024) equips an LM with a 3D molecular encoder through a dedicated projector for 3D-aware captioning and QA. LLaMo (Park et al., 2024) integrates a molecular graph encoder with a multi-level graph projector to bridge graphs and language. BioMedGPT-

Table 18: Hyperparameter settings for benchmarks. For all the evaluated models and datasets, identical finetuning or inference parameters are applied for fair comparison.

| Parameter | MoleculeQA | Mol-Instructions | PAMPA/BBBP |
|---|---|---|---|
| Batch size | 2 | 4 | 32 |
| Precision | BF16-Mixed | BF16-Mixed | BF16-Mixed |
| Max epochs | 5 | 10 | – |
| Accumulate grad batches | 64 | 32 | – |
| Weight decay | 0.05 | 0.05 | – |
| Initial LR | 1.00e-04 | 1.00e-04 | – |
| Minimum LR | 1.00e-05 | 1.00e-05 | – |
| Warmup LR | 1.00e-06 | 1.00e-06 | – |
| Warmup steps | 100 | 100 | – |
| Scheduler | Cosine LR | Cosine LR | – |
| Temperature | 0.1 | 0.1 | 0.1 |
| Max input tokens | 1024 | 1024 | 512 |

LM (Luo et al., 2023b) is a biomedical generative LM adapted for molecule- and biology-centric tasks. Mol-Reasoner (Zhao et al., 2025) emphasizes interpretable chemical reasoning with a two-stage procedure (supervised reasoning initialization followed by reinforcement learning). We further compare with a graph-text align model HIGHT (Chen et al., 2024) on three tasks from the Mol-Instructions (Fang et al., 2023) benchmark.

Benchmarks span factual QA, molecular captioning, property prediction, forward reaction prediction, retrosynthesis, reagent prediction, and ADME tasks, providing complementary coverage of accuracy, reasoning, and domain grounding. Baselines include both strong general LLMs and chemistry-specialized LLMs to isolate EDT-Former 's contribution beyond backbone capacity.

BASELINE RESULTS

The official benchmark protocols and data splits are adopted for all evaluations. For baselines whose reported performance trails ours by more than 10%, we cite the results directly from the benchmark sources (Lu et al., 2024; Fang et al., 2023; Kim et al., 2025). For competitive models (within this margin) and for baselines not covered by the benchmark, we re-run both the baselines and our method under a unified, fair protocol using released code and configurations. Full implementation and evaluation details are provided in App. D.3.

D.2  DATA PROCESSING

For structural inputs, we adopt Uni-Mol (Zhou et al., 2023) and MoleculeSTM (Liu et al., 2023). For Mol-LLaMA-Instruct (Kim et al., 2025), we use the provided 3D coordinates for both pretraining and finetuning. For other datasets (e.g., Mol-Instructions (Fang et al., 2023)), we generate eleven conformers with RDKit and perform geometry optimization using **MMFF** (Merck Molecular Force Field Halgren (1996)), a classical small molecule force field that assigns bonded and nonbonded terms to atoms and bonds and minimizes molecular energy to a locally stable 3D geometry. After optimization, RDKit is used to construct the 2D molecular graph (including atom and bond types, charges, hybridization, aromaticity, and stereochemical flags). Molecules that fail RDKit sanitization are filtered out. All processed data and scripts are released for reproducibility.

D.3  EVALUATION SETTINGS

As summarized in Tab. 18, all models evaluated on MoleculeQA (Lu et al., 2024) and Mol-Instructions (Fang et al., 2023) are finetuned under identical hyperparameter settings to ensure a fair comparison. For PAMPA and BBBP (Huang et al., 2021), all models are evaluated zero-shot with a shared prompt template—BBBP as shown in Fig. 9 and PAMPA following Mol-Llama (Kim et al., 2025). For the 10-shot setting on MoleculeQA, the support set consists of 10 randomly selected examples fixed once and reused for every 10-shot model to control variance.

---

*BBBP: Blood-Brain Barrier Penetration*

**Direct**
`System`: "You are a drug discovery assistant tasked with predicting whether a molecule can penetrate the blood-brain barrier (BBBP). Output the final decision strictly using the provided answer format string.\nYour final answer should be formatted as either: 'Final answer: Penetrant' or 'Final answer: Non-penetrant' "
`User`: "Determine the BBBP class of the given molecule.\nMolecule <mol>."

**Reasoning**
`System`: "You are a drug discovery assistant tasked with predicting whether a molecule can penetrate the blood-brain barrier (BBBP). Output the final decision strictly using the provided answer format string.\nYour final answer should be formatted as either: 'Final answer: Penetrant' or 'Final answer: Non-penetrant'"
`User`: "Determine the BBBP class of the given molecule and briefly explain your rationale.\nMolecule <mol>."

**Rich Instructions**
`System`: "You are a drug discovery assistant for BBBP prediction. Molecules that are lipophilic, appropriately sized, and not highly polar are more likely to penetrate the BBB. Consider: lipophilicity (logP), polar surface area, hydrogen bonding capacity, ionization state, and presence of efflux transporter motifs. Output the final decision strictly using the provided answer format string.\nYour final answer should be formatted as either: 'Final answer: Penetrant' or 'Final answer: Non-penetrant' "
`User`: " Determine the BBBP class of the given molecule.\nMolecule <mol>."

---

Figure 9: Zero-shot prompt templates for BBBP (Blood–Brain Barrier Penetration). Three settings are evaluated: (i) Direct, which asks for a binary decision; (ii) Reasoning, which requests a brief rationale before the decision; and (iii) Rich Instructions, which provides domain context (e.g., lipophilicity, PSA, H-bonding, ionization, efflux motifs) prior to answering. All prompts enforce an identical output format: `Final answer: Penetrant` or `Final answer: Non-penetrant`.

**Evaluation Metrics.** (1) MoleculeQA (Lu et al., 2024): We report task-wise accuracy (fraction of correctly answered questions within each task). Average accuracy is the unweighted mean of the four task accuracies, and total accuracy is the overall correct/total across all questions pooled from the four tasks (count-based). Higher is better. (2) Mol-Instructions (Fang et al., 2023): For property prediction, we use Mean Absolute Error (MAE), $\text{MAE} = \frac{1}{n}\sum_{i=1}^{n}|\hat{y}_i - y_i|$ (lower is better). For molecular description generation, we report: BLEU-2/4 (n-gram precision with a brevity penalty; BLEU-2 uses up to bigrams, BLEU-4 up to 4-grams; higher is better), ROUGE-1/2 (recall-oriented overlap of unigrams/bigrams; higher is better), ROUGE-L (recall-oriented longest common subsequence; higher is better), and METEOR (harmonic mean of unigram precision and recall with stemming/synonym matching and fragmentation penalty; higher is better). For SMILES/text generation tasks, we additionally report Exact (fraction of predictions that exactly match the reference sequence; higher is better), Levenshtein (mean Levenshtein edit distance between predicted and reference SMILES; lower is better), RDK/MACCS/Morgan FTS (mean fingerprint Tanimoto similarity computed with RDKit, MACCS, and Morgan fingerprints respectively; higher is better), Validity (fraction of generated SMILES that can be parsed and sanitized into valid molecules by RDKit; higher is better), and BERTScore (contextual-embedding-based semantic similarity between generated and reference texts; higher is better).

## D.4 Evaluation on Macrocycle Dataset–NPMMPD

Macrocycles represent a challenging regime for molecular–language alignment due to their large ring systems, long-range dependencies, and highly flexible conformational space. To assess the robustness of EDT-Former under this more demanding structural distribution, we additionally evaluate on the NPMMPD macrocycle benchmark (Feng et al., 2025). The dataset provides experimentally derived permeability measurements, and following prior work, we focus on the PAMPA Log $P_{app}$ prediction task, which contains 326 unique macrocyclic molecules. This setting remains strictly zero-shot.

Table 19: Zero-shot accuracy (%) on the NPMMPD PAMPA dataset under 13 prompt settings. The best (pink) and second-best (lightpink) results are highlighted.

| Model | D1 | D2 | D3 | D4 | R1 | R-2 | R-3 | RI-1 | RI-2 | RI-3 | Bi | List | Conf | Avg. |
|---|---|---|---|---|---|---|---|---|---|---|---|---|---|---|
| *General LLMs* | | | | | | | | | | | | | | |
| Llama2 | 59.84 | 52.56 | 47.48 | 41.37 | 52.49 | 48.03 | 41.08 | 57.14 | 52.67 | 49.78 | 41.3 | 42.24 | 41.3 | 48.25 |
| Llama3.1 | 46.54 | 42.51 | 38.68 | 36.10 | 47.33 | 44.25 | 37.68 | 51.4 | 46.69 | 43.82 | 41.3 | 41.61 | 43.48 | 43.18 |
| *Molecular LLMs* | | | | | | | | | | | | | | |
| Mol-Ins.-Llama2 | 41.94 | 30.98 | 37.30 | 24.32 | 44.63 | 38.23 | 41.00 | 44.55 | 42.02 | 38.68 | 45.34 | 40.37 | 40.37 | 39.21 |
| Mol-Ins.-Llama3.1 | 41.67 | 35.27 | 42.03 | 39.23 | 41.13 | 43.54 | 39.53 | 38.87 | 35.36 | 35.71 | 42.55 | 41.6 | 59.57 | 41.24 |
| Mol-LLaMA-2 | 44.88 | 35.92 | 41.58 | 32.40 | 36.21 | 25.58 | 31.20 | 38.81 | 33.58 | 30.19 | 54.04 | 41.56 | 40.94 | 37.45 |
| Mol-LLaMA3.1 | 39.79 | 30.86 | 36.46 | 23.63 | 42.93 | 42.93 | 42.93 | 38.11 | 35.14 | 27.71 | 49.84 | 42.95 | 44.72 | 38.31 |
| *Ours* | | | | | | | | | | | | | | |
| **EDT-Former** | 62.66 | 53.93 | 55.55 | 43.47 | 58.39 | 47.98 | 50.98 | 55.9 | 53.06 | 50.59 | 45.65 | 43.48 | 55.59 | **51.81** |

Table 20: Zero-shot performance per prompt on the AMES toxicity prediction task from the TDC benchmark. We report accuracy (%) under 13 prompt settings. The best (pink) and second-best (lightpink) results are highlighted.

| Model | Size | D1 | D2 | D3 | D4 | R1 | R-2 | R-3 | RI-1 | RI-2 | RI-3 | Bi | List | Conf | Avg. |
|---|---|---|---|---|---|---|---|---|---|---|---|---|---|---|---|
| *General LLMs* | | | | | | | | | | | | | | | |
| GPT-4o | - | 46.26 | 54.95 | 55.43 | 55.00 | 47.11 | 55.38 | 55.27 | 53.07 | 53.90 | 55.72 | 55.17 | 55.48 | 53.93 | 53.59 |
| Llama2 | 7B | 45.74 | 54.92 | 54.81 | 54.81 | 45.80 | 54.81 | 54.81 | 52.34 | 53.44 | 54.81 | 54.88 | 54.13 | 54.68 | 53.08 |
| Llama3.1 | 8B | 46.13 | 54.33 | 55.40 | 54.54 | 47.77 | 55.30 | 55.09 | 53.15 | 53.71 | 55.98 | 54.81 | 56.19 | 52.54 | 53.46 |
| *Molecular LLMs* | | | | | | | | | | | | | | | |
| 3D-MoLM | 7B | 51.75 | 54.43 | 52.95 | 50.29 | 52.35 | 53.46 | 53.90 | 52.75 | 53.59 | 55.48 | 53.03 | 50.28 | 54.91 | 53.01 |
| LLaMo | 7B | 49.50 | 52.31 | 53.21 | 50.36 | 49.80 | 50.70 | 50.91 | 50.34 | 53.11 | 54.89 | 53.57 | 48.12 | 54.93 | 51.67 |
| Mol-Ins.-Llama2 | 8B | 53.37 | 55.91 | 52.06 | 49.59 | 54.26 | 55.57 | 56.26 | 54.51 | 53.44 | 55.43 | 51.86 | 51.79 | 54.26 | 53.72 |
| Mol-Ins.-Llama3.1 | 8B | 44.98 | 48.07 | 53.71 | 50.48 | 44.70 | 45.19 | 44.91 | 45.53 | 52.13 | 53.71 | 54.63 | 43.81 | 54.95 | 48.98 |
| Mol-LLaMA-2 | 7.2B | 46.34 | 53.27 | 54.62 | 53.01 | 51.37 | 55.83 | 54.66 | 53.28 | 56.49 | 53.16 | 48.84 | 52.29 | 55.51 | 52.97 |
| Mol-LLaMA3.1 | 8.3B | 53.51 | 54.92 | 55.16 | 55.60 | 53.39 | 54.68 | 54.21 | 53.78 | 55.36 | 53.05 | 54.12 | 54.81 | 55.96 | 54.50 |
| *Ours* | | | | | | | | | | | | | | | |
| EDT-Former | 8.3B | 48.43 | 55.53 | 56.18 | 54.97 | 52.69 | 56.69 | 57.86 | 54.52 | 56.98 | 58.79 | 50.77 | 55.59 | 58.59 | 55.20 |

Table 21: Zero-shot performance per prompt on the BACE classification task from the MoleculeNet benchmark. We report accuracy (%) under 13 prompt settings. The best (pink) and second-best (lightpink) results are highlighted.

| Model | Size | D1 | D2 | D3 | D4 | R1 | R-2 | R-3 | RI-1 | RI-2 | RI-3 | Bi | List | Conf | Avg. |
|---|---|---|---|---|---|---|---|---|---|---|---|---|---|---|---|
| *General LLMs* | | | | | | | | | | | | | | | |
| GPT-4o | - | 40.72 | 37.22 | 38.88 | 39.46 | 40.14 | 39.50 | 40.39 | 38.96 | 38.16 | 39.34 | 39.12 | 42.25 | 39.13 | 39.48 |
| Llama2 | 7B | 40.13 | 36.84 | 38.82 | 38.82 | 39.47 | 38.82 | 39.47 | 38.00 | 37.50 | 38.82 | 38.82 | 42.11 | 38.82 | 38.96 |
| Llama3.1 | 8B | 40.13 | 38.16 | 38.82 | 43.42 | 38.82 | 41.45 | 38.82 | 40.79 | 38.16 | 46.71 | 39.47 | 38.16 | 42.76 | 40.44 |
| *Molecular LLMs* | | | | | | | | | | | | | | | |
| 3D-MoLM | 7B | 39.74 | 39.53 | 48.17 | 50.44 | 37.91 | 35.14 | 36.58 | 37.29 | 45.79 | 42.12 | 45.73 | 36.62 | 43.52 | 41.43 |
| LLaMo | 7B | 39.60 | 39.08 | 48.06 | 49.94 | 39.67 | 36.37 | 37.73 | 36.87 | 45.30 | 41.65 | 47.77 | 37.39 | 45.37 | 41.91 |
| Mol-Ins.-Llama2 | 8B | 38.82 | 40.13 | 49.34 | 51.97 | 38.82 | 38.16 | 38.82 | 38.82 | 45.39 | 40.79 | 48.03 | 39.47 | 45.39 | 42.61 |
| Mol-Ins.-Llama3.1 | 8B | 40.22 | 36.36 | 39.78 | 36.76 | 39.84 | 38.37 | 39.68 | 45.16 | 42.67 | 39.76 | 40.00 | 36.11 | 46.91 | 40.12 |
| Mol-LLaMA-2 | 7.2B | 44.49 | 48.68 | 40.67 | 52.63 | 38.82 | 39.07 | 38.82 | 38.03 | 38.57 | 38.46 | 44.74 | 38.16 | 38.82 | 41.54 |
| Mol-LLaMA3.1 | 8.3B | 37.50 | 39.29 | 38.19 | 39.86 | 39.47 | 40.54 | 36.67 | 38.03 | 35.51 | 38.81 | 40.40 | 37.14 | 41.45 | 38.68 |
| *Ours* | | | | | | | | | | | | | | | |
| EDT-Former | 8.3B | 46.05 | 55.92 | 37.50 | 52.98 | 46.71 | 61.18 | 60.53 | 55.26 | 44.08 | 44.74 | 38.82 | 53.29 | 41.45 | 49.12 |

For a fair comparison across models with different pretraining protocols, we adopt the standardized 13 prompt configurations used in the main paper (Direct, Reasoning, Rich Instructions, and ten GPT-generated paraphrased templates). All models receive exactly the same molecule description and prompt wording under each configuration.

Table 19 reports accuracy under each prompt and the averaged accuracy. Across nearly all prompt variants, EDT-Former achieves the strongest performance, demonstrating stable behavior even when prompts vary substantially in structure and reasoning depth. This robustness highlights the advantage of combining entropy-guided dynamic segmentation with anchor-based alignment, especially for large and structurally complex macrocyclic systems where fixed-length token connectors typically struggle.

## D.5 DETAILED RESULTS OF PROPERTY PREDICTION TASKS PER PROMPT

Large language models exhibit substantial variability under different natural language instructions, especially for molecular property prediction. To systematically evaluate this prompt sensitivity,

Table 22: Zero-shot performance per prompt on the BBBP classification task from the TDC benchmark. We report accuracy (%) under 13 prompt settings. The best (pink) and second-best (lightpink) results are highlighted.

| Model | Size | D1 | D2 | D3 | D4 | R1 | R-2 | R-3 | RI-1 | RI-2 | RI-3 | Bi | List | Conf | Avg. |
|---|---|---|---|---|---|---|---|---|---|---|---|---|---|---|---|
| *General LLMs* | | | | | | | | | | | | | | | |
| GPT-4o | - | 60.82 | 61.46 | 61.46 | 62.34 | 61.34 | 61.86 | 62.66 | 64.43 | 65.10 | 65.11 | 65.91 | 66.37 | 66.89 | 63.52 |
| Llama2 | 7B | 37.37 | 38.33 | 38.50 | 39.22 | 51.56 | 52.35 | 52.53 | 53.09 | 53.34 | 54.07 | 55.01 | 55.71 | 56.71 | 49.06 |
| Llama3.1 | 8B | 57.07 | 57.15 | 57.19 | 58.02 | 51.03 | 51.44 | 52.34 | 55.15 | 55.68 | 56.28 | 57.06 | 57.21 | 57.66 | 55.64 |
| *Molecular LLMs* | | | | | | | | | | | | | | | |
| 3D-MoLM | 7B | 49.14 | 50.01 | 50.42 | 50.74 | 51.65 | 52.16 | 51.07 | 51.91 | 53.61 | 50.74 | 51.28 | 51.54 | 51.40 | 51.20 |
| LLaMo | 7B | 55.44 | 55.81 | 56.42 | 56.85 | 55.45 | 55.46 | 56.16 | 56.91 | 58.67 | 57.52 | 59.31 | 59.29 | 60.08 | 57.18 |
| Mol-Ins.-Llama2 | 8B | 52.58 | 53.47 | 53.95 | 54.29 | 52.58 | 51.55 | 50.79 | 51.34 | 50.14 | 48.97 | 51.29 | 48.68 | 46.19 | 51.22 |
| Mol-Ins.-Llama3.1 | 8B | 53.44 | 54.37 | 55.09 | 55.74 | 55.31 | 55.38 | 56.76 | 54.91 | 54.38 | 52.27 | 49.38 | 49.15 | 46.62 | 53.29 |
| Mol-LLaMA-2 | 7.2B | 53.37 | 55.28 | 53.97 | 54.19 | 52.58 | 53.65 | 55.14 | 52.58 | 52.43 | 53.89 | 55.84 | 52.56 | 56.38 | 53.99 |
| Mol-LLaMA3.1 | 8.3B | 59.54 | 59.54 | 58.24 | 59.38 | 55.56 | 53.18 | 51.72 | 59.08 | 54.67 | 53.29 | 55.73 | 58.92 | 57.45 | 56.64 |
| *Ours* | | | | | | | | | | | | | | | |
| EDT-Former | 8.3B | 74.44 | 71.83 | 73.46 | 70.92 | 74.69 | 70.15 | 68.73 | 75.86 | 72.38 | 70.95 | 73.12 | 71.47 | 74.28 | 72.48 |

we design a unified set of **13 prompts** spanning three major categories (**Direct**, **Reasoning**, and **Rich Instructions**) together with three complementary variants that stress different aspects of LLM behavior. A full prompt example of BBBP is shown in Fig. 10.

**Direct prompts.** These prompts provide a minimalistic instruction: the model is asked to output only the final class label without any auxiliary context or justification. They represent the "pure classification" setting and test whether the model can reliably make a decision without verbal reasoning or domain cues.

**Reasoning prompts.** These prompts require the model to briefly explain the structural or physicochemical rationale (e.g., substructures, polarity, lipophilicity) before giving the final answer. Although the reasoning text is not evaluated, this setup probes whether intermediate verbalization stabilizes predictions or helps models avoid superficial correlations.

**Rich-instruction prompts.** These prompts include additional domain background, such as AD-MET principles, known structure–property heuristics, or high-level hints about BBBP, toxicity, or activity patterns. They simulate a more instructive "expert assistant" setting and evaluate whether models can utilize domain knowledge to improve prediction robustness.

**Additional control-style prompts.** Beyond these three primary categories, we include:

- **Binary**: forces the LLM to behave like a strict binary classifier.
- **Confidence**: encourages internal reflection before giving the final label.
- **Checklist**: requires the LLM to follow a structured analysis procedure (identify substructures, consider polarity, check alerts, then decide).

These variants test different behavioral biases of LLMs and reveal how formatting, reasoning style, and instruction complexity influence prediction outputs.

**Why prompt diversity matters.** Property prediction tasks, especially classification benchmarks such as BBBP, AMES, BACE, and toxicity tasks, are highly sensitive to instruction phrasing. For tasks with balanced classes and clear structural cues (e.g., BBBP, AMES), most models remain relatively stable across the 13 prompts. However, for datasets with severe imbalance or safety-triggering semantics (e.g., **ClinTox**, where the term "toxicity" may cause LLMs to refuse or hedge answers), prompt phrasing can cause large swings in accuracy. Therefore, evaluating with a single prompt can give a misleading view of a model's real capability.

**Results across the 13 prompts.** Tables in this section report complete accuracy across all prompts for all ten datasets (AMES-Tab. 20, BACE-Tab. 21, BBBP-Tab. 22, ClinTox-Tab. 23, DILI-Tab. 24, hERG-Tab. 25, HIA-Tab. 26, HIV-Tab. 27, PGP-Tab. 28, PAMPA-Tab. 29). Across these 13 variants, **EDT-Former achieves the best mean accuracy on 9 out of the 10 tasks**, demonstrating strong robustness to prompt variation. For most datasets, **EDT-Former also ranks top-1 under the majority of individual prompts**, indicating that its advantage is not tied to a single favorable

*Direct Prompts*

**D-1**: " You are a drug discovery assistant tasked with predicting whether a molecule can penetrate the blood-brain barrier (BBBP). Output the final decision strictly using the provided answer format string.\nYour final answer should be formatted as either: 'Final answer: Penetrant' or 'Final answer: Non-penetrant'. Determine the BBBP class of the given molecule.\n Molecule <mol>."

**D-2**: "You are an expert molecular property prediction assistant for a binary task. For each molecule, you must decide whether it belongs to the positive (Active) or negative (Inactive) class defined by this dataset. Always follow the required answer format exactly.\nYour final answer should be formatted as either: 'Final answer: Active' or 'Final answer: Inactive'. Classify the following molecule as Active or Inactive according to this dataset's task.\nMolecule <mol>."

**D-3**: "You are an expert molecular property prediction assistant for a binary task. For each molecule, you must decide whether it belongs to the positive (Active) or negative (Inactive) class defined by this dataset. Always follow the required answer format exactly.\nYour final answer should be formatted as either: 'Final answer: Active' or 'Final answer: Inactive'. Based on its structure, decide whether the molecule should be labeled Active or Inactive for this task. Provide only the classification in the required final answer format.\nMolecule <mol>."

**D-4**: "You are an expert molecular property prediction assistant for a binary task. For each molecule, you must decide whether it belongs to the positive (Active) or negative (Inactive) class defined by this dataset. Always follow the required answer format exactly.\nYour final answer should be formatted as either: 'Final answer: Active' or 'Final answer: Inactive'. For the molecule below, determine the correct binary label (Active or Inactive) for this dataset and follow the exact final answer format.\nMolecule <mol>."

*Reasoning Prompts*

**R-1**: "You are a drug discovery assistant tasked with predicting whether a molecule can penetrate the blood-brain barrier (BBBP). Output the final decision strictly using the provided answer format string.\nYour final answer should be formatted as either: 'Final answer: Penetrant' or 'Final answer: Non-penetrant'. Determine the BBBP class of the given molecule and briefly explain your rationale.\n Molecule <mol>."

**R-2**: "You are an expert molecular property prediction assistant. You should briefly reason about structural and physicochemical features of the molecule before deciding whether it is Active or Inactive. However, your final line must obey the required answer format.\nYour final answer should be formatted as either: 'Final answer: Active' or 'Final answer: Inactive'. Briefly explain the main structural or physicochemical reasons for your decision, then give the final Active or Inactive label in the required format.\nMolecule <mol>."

**R-3**: "You are an expert molecular property prediction assistant. You should briefly reason about structural and physicochemical features of the molecule before deciding whether it is Active or Inactive. However, your final line must obey the required answer format.\nYour final answer should be formatted as either: 'Final answer: Active' or 'Final answer: Inactive'. Consider potential substructures, electronic features, and global properties of the molecule, summarize your reasoning in one or two sentences, and then output the final Active or Inactive label using the required final answer format.\nMolecule <mol>."

*Task-Info Give Prompts*

**T-1**: "You are a molecular property prediction assistant working on a specific binary dataset task (e.g., activity, toxicity, or ADME classification). Use your knowledge of medicinal chemistry, ADMET principles, and structure\u2013property relationships to map molecules to the Active or Inactive class. Do not change the required answer format.\nYour final answer should be formatted as either: 'Final answer: Active' or 'Final answer: Inactive'. Using your understanding of typical structure\u2013property relationships in drug discovery, decide whether the molecule should be classified as Active or Inactive in this dataset.\nMolecule <mol>."

**T-2**: "You are a molecular property prediction assistant working on a specific binary dataset task (e.g., activity, toxicity, or ADME classification). Use your knowledge of medicinal chemistry, ADMET principles, and structure\u2013property relationships to map molecules to the Active or Inactive class. Do not change the required answer format.\nYour final answer should be formatted as either: 'Final answer: Active' or 'Final answer: Inactive'. Treat this as a realistic drug discovery task. Infer whether the molecule belongs to the positive (Active) or negative (Inactive) class defined by the dataset, and output only the final label using the required format.\nMolecule <mol>."

**T-3**: "You are a drug discovery assistant for BBBP prediction. Molecules that are lipophilic, appropriately sized, and not highly polar are more likely to penetrate the BBB. Consider: lipophilicity (logP), polar surface area, hydrogen bonding capacity, ionization state, and presence of efflux transporter motifs. Output the final decision strictly using the provided answer format string.\nYour final answer should be formatted as either: 'Final answer: Penetrant' or 'Final answer: Non-penetrant'. Determine the BBBP class of the given molecule.\nMolecule <mol>."

*Other Prompts*

**Binary**: "You are a strict binary classifier for molecular property prediction. You are only allowed to output one of two labels, using the required format.\nYour final answer must be exactly one of: 'Final answer: Active' or 'Final answer: Inactive'. Decide whether the molecule is Active or Inactive for this dataset. Do not output any explanation, only the final answer line.\nMolecule <mol>."

**Confidence**: "You are a careful molecular property prediction assistant. First, internally estimate how confident you are that the molecule is Active or Inactive, but do NOT include numerical probabilities in the output. After reasoning, output only the final decision using the required format.\nYour final answer should be formatted as either: 'Final answer: Active' or 'Final answer: Inactive'. Carefully consider ambiguous or borderline cases internally, but in your visible output provide only the final Active or Inactive label using the required final answer format.\nMolecule <mol>."

**Checklist**: "You are a molecular property prediction assistant that follows a short checklist before answering: (1) identify key substructures; (2) consider lipophilicity, polarity, and size; (3) consider potential functional-group alerts; then (4) decide whether the molecule is Active or Inactive. Despite using this checklist, you must keep the final answer format unchanged.\nYour final answer should be formatted as either: 'Final answer: Active' or 'Final answer: Inactive'. Follow your internal checklist to analyze the molecule and then decide whether it should be labeled Active or Inactive. In the visible output, you may include a brief rationale, but the last line must follow the required final answer format.\nMolecule <mol>."

Figure 10: Example (BBBP task) of 13 prompt settings for the property prediction tasks from MoleculeNet and TDC benchmark. 10 Prompts are divided into three main categories–direct, reasoning, and rich-instruction (task-info). Three more prompts are named binary, confidence, and checklist.

template. On particularly challenging toxicity tasks such as ClinTox, all LLMs exhibit substantial prompt-to-prompt variation due to class imbalance and occasional refusal-to-answer behavior, yet EDT-Former still maintains consistently strong relative performance across prompts.

Table 23: Zero-shot performance per prompt on the CLINTOX classification task from the MoleculeNet benchmark. We report accuracy (%) under 13 prompt settings. The best (pink) and second-best (lightpink) results are highlighted.

| Model | Size | D1 | D2 | D3 | D4 | R1 | R-2 | R-3 | RI-1 | RI-2 | RI-3 | Bi | List | Conf | Avg. |
|---|---|---|---|---|---|---|---|---|---|---|---|---|---|---|---|
| *General LLMs* | | | | | | | | | | | | | | | |
| GPT-4o | - | 93.01 | 10.50 | 5.81 | 8.69 | 82.46 | 9.82 | 10.02 | 71.10 | 10.98 | 7.20 | 5.91 | 18.08 | 8.87 | 26.34 |
| Llama2 | 7B | 91.03 | 11.72 | 7.59 | 7.59 | 83.45 | 8.97 | 7.59 | 71.72 | 9.66 | 7.59 | 8.28 | 15.86 | 7.59 | 26.05 |
| Llama3.1 | 8B | 88.28 | 9.66 | 7.59 | 17.93 | 91.03 | 8.97 | 8.97 | 15.86 | 14.48 | 19.31 | 8.97 | 10.34 | 9.66 | 23.93 |
| *Molecular LLMs* | | | | | | | | | | | | | | | |
| 3D-MoLM | 7B | 23.89 | 28.95 | 45.36 | 58.18 | 15.01 | 10.18 | 13.69 | 18.35 | 19.44 | 20.54 | 41.14 | 20.85 | 21.40 | 25.92 |
| LLaMo | 7B | 23.81 | 31.04 | 47.08 | 56.79 | 15.56 | 9.98 | 11.84 | 18.84 | 17.04 | 21.34 | 40.02 | 19.52 | 19.30 | 25.55 |
| Mol-Ins.-Llama2 | 8B | 25.52 | 30.34 | 46.90 | 57.93 | 15.17 | 8.97 | 13.79 | 17.93 | 17.93 | 23.45 | 40.69 | 17.93 | 21.38 | 25.99 |
| Mol-Ins.-Llama3.1 | 8B | 47.25 | 18.37 | 14.42 | 11.58 | 43.88 | 8.97 | 8.64 | 56.70 | 10.53 | 9.90 | 47.17 | 17.54 | 53.26 | 26.79 |
| Mol-LLaMA-2 | 7.2B | 55.43 | 28.47 | 25.00 | 48.97 | 88.43 | 8.28 | 7.64 | 23.21 | 12.95 | 12.68 | 37.24 | 7.59 | 7.64 | 27.96 |
| Mol-LLaMA3.1 | 8.3B | 88.28 | 9.29 | 9.42 | 7.91 | 86.81 | 10.07 | 9.86 | 14.69 | 8.11 | 13.33 | 17.14 | 9.16 | 15.86 | 23.07 |
| *Ours* | | | | | | | | | | | | | | | |
| EDT-Former | 8.3B | 75.86 | 48.28 | 30.34 | 52.41 | 80.00 | 92.41 | 91.72 | 77.24 | 32.41 | 34.48 | 22.07 | 75.86 | 22.07 | 56.55 |

Table 24: Zero-shot performance per prompt on the DILI classification task from the TDC benchmark. We report accuracy (%) under 13 prompt settings. The best and second-best are highlighted.

| Model | Size | D1 | D2 | D3 | D4 | R1 | R-2 | R-3 | RI-1 | RI-2 | RI-3 | Bi | List | Conf | Avg. |
|---|---|---|---|---|---|---|---|---|---|---|---|---|---|---|---|
| *General LLMs* | | | | | | | | | | | | | | | |
| GPT-4o | - | 55.84 | 54.72 | 51.42 | 51.81 | 55.72 | 51.36 | 55.94 | 51.58 | 57.40 | 53.17 | 54.86 | 50.11 | 51.12 | 53.47 |
| Llama2 | 7B | 53.68 | 51.58 | 50.53 | 50.53 | 53.68 | 49.47 | 51.58 | 51.58 | 53.68 | 51.58 | 51.58 | 46.32 | 50.53 | 51.26 |
| Llama3.1 | 8B | 49.47 | 48.42 | 54.74 | 52.63 | 50.53 | 51.58 | 52.63 | 50.53 | 50.53 | 60.00 | 51.58 | 54.74 | 52.63 | 52.31 |
| *Molecular LLMs* | | | | | | | | | | | | | | | |
| 3D-MoLM | 7B | 50.52 | 48.00 | 43.26 | 41.16 | 54.85 | 47.28 | 50.94 | 50.19 | 56.65 | 51.81 | 45.02 | 53.08 | 41.34 | 48.78 |
| LLaMo | 7B | 52.01 | 49.65 | 44.66 | 43.38 | 52.80 | 49.43 | 48.55 | 50.93 | 55.71 | 52.12 | 44.16 | 53.09 | 41.49 | 49.08 |
| Mol-Ins.-Llama2 | 8B | 50.53 | 49.47 | 43.16 | 44.21 | 51.58 | 49.47 | 50.53 | 50.53 | 54.74 | 52.63 | 45.26 | 52.63 | 41.05 | 48.91 |
| Mol-Ins.-Llama3.1 | 8B | 47.37 | 53.68 | 58.95 | 55.79 | 55.79 | 51.58 | 53.68 | 54.74 | 51.58 | 52.63 | 45.26 | 51.58 | 52.63 | 52.39 |
| Mol-LLaMA-2 | 7.2B | 50.29 | 56.84 | 52.13 | 44.68 | 50.53 | 52.13 | 51.58 | 51.58 | 49.43 | 58.43 | 52.63 | 53.19 | 51.58 | 51.92 |
| Mol-LLaMA3.1 | 8.3B | 54.26 | 53.93 | 50.00 | 48.39 | 52.63 | 51.58 | 50.54 | 50.53 | 52.00 | 55.68 | 53.26 | 51.65 | 48.42 | 51.76 |
| *Ours* | | | | | | | | | | | | | | | |
| EDT-Former | 8.3B | 60.61 | 52.86 | 52.31 | 50.00 | 50.70 | 40.00 | 35.59 | 51.06 | 49.33 | 45.61 | 54.00 | 62.16 | 49.28 | 50.27 |

Table 25: Zero-shot performance per prompt on the HERG classification task from the TDC benchmark. We report accuracy (%) under 13 prompt settings. The best and second-best results are highlighted.

| Model | Size | D1 | D2 | D3 | D4 | R1 | R-2 | R-3 | RI-1 | RI-2 | RI-3 | Bi | List | Conf | Avg. |
|---|---|---|---|---|---|---|---|---|---|---|---|---|---|---|---|
| *General LLMs* | | | | | | | | | | | | | | | |
| GPT-4o | - | 24.97 | 73.65 | 73.43 | 73.39 | 31.10 | 72.50 | 74.54 | 55.75 | 69.47 | 71.17 | 70.63 | 68.57 | 73.39 | 64.04 |
| Llama2 | 7B | 24.03 | 74.02 | 74.38 | 74.42 | 33.33 | 72.87 | 74.42 | 54.23 | 72.87 | 74.42 | 74.42 | 67.44 | 74.42 | 65.02 |
| Llama3.1 | 8B | 25.58 | 72.87 | 74.42 | 64.34 | 25.58 | 75.97 | 75.19 | 56.59 | 69.77 | 70.54 | 75.19 | 72.87 | 76.74 | 64.28 |
| *Molecular LLMs* | | | | | | | | | | | | | | | |
| 3D-MoLM | 7B | 72.24 | 73.64 | 60.09 | 50.78 | 59.89 | 73.74 | 74.66 | 25.68 | 63.58 | 62.84 | 54.07 | 71.48 | 71.96 | 62.67 |
| LLaMo | 7B | 71.15 | 72.91 | 60.78 | 51.80 | 58.25 | 72.55 | 76.12 | 27.67 | 64.45 | 61.75 | 55.71 | 71.01 | 71.91 | 62.77 |
| Mol-Ins.-Llama2 | 8B | 72.09 | 74.42 | 58.91 | 52.71 | 56.59 | 74.42 | 73.64 | 28.68 | 65.89 | 62.02 | 56.59 | 69.77 | 70.54 | 62.79 |
| Mol-Ins.-Llama3.1 | 8B | 28.68 | 48.84 | 65.89 | 48.06 | 28.68 | 26.36 | 27.13 | 25.58 | 62.02 | 68.99 | 65.89 | 51.94 | 65.89 | 47.23 |
| Mol-LLaMA-2 | 7.2B | 65.60 | 67.44 | 68.99 | 44.53 | 73.23 | 74.42 | 73.64 | 27.13 | 73.81 | 61.24 | 61.24 | 74.42 | 74.60 | 65.60 |
| Mol-LLaMA3.1 | 8.3B | 67.69 | 76.92 | 69.41 | 71.23 | 73.68 | 72.53 | 73.33 | 74.17 | 73.63 | 71.72 | 62.75 | 72.82 | 58.11 | 70.61 |
| *Ours* | | | | | | | | | | | | | | | |
| EDT-Former | 8.3B | 65.60 | 75.19 | 74.42 | 74.92 | 72.58 | 73.23 | 75.00 | 74.42 | 74.18 | 74.39 | 74.47 | 72.22 | 74.42 | 73.46 |

## D.6 EVALUATION ON MOLECULE DESIGN AND OPEN QUESTIONS FROM MOL-INSTRUCTIONS

**Description-guided Molecule Design.** The description-guided molecule design task evaluates a model's ability to generate valid molecular structures that follow natural-language specifications. Following the official Mol-Instructions protocol, we strictly adopt the official splits and compare against all official baseline models under the unified benchmark setting.. All models are assessed using the official evaluation metrics: Exact Match, BLEU, Levenshtein distance, fingerprint-based similarity scores (RDK FTS, MACCS FTS, Morgan FTS), and chemical validity.

As shown in Table 30, EDT-Former achieves the strongest performance on most metrics of the description-guided molecule design benchmark. In particular, it attains the best BLEU score, the lowest Levenshtein distance, and the highest RDK and MACCS fingerprint similarity scores, while also reaching perfect chemical validity (100% valid generations). Other strong molecular baselines such as MolT5 and Mol-Instructions–Llama models occasionally win on individual metrics (e.g.,

Table 26: Per prompt zero-shot accuracy (%) on the HIA classification task from the TDC benchmark, under 13 prompt settings. The best (pink) and second-best (lightpink) results are highlighted.

| Model | Size | D1 | D2 | D3 | D4 | R1 | R-2 | R-3 | RI-1 | RI-2 | RI-3 | Bi | List | Conf | Avg. |
|---|---|---|---|---|---|---|---|---|---|---|---|---|---|---|---|
| *General LLMs* | | | | | | | | | | | | | | | |
| GPT-4o | - | 66.99 | 80.02 | 82.60 | 86.58 | 61.89 | 84.24 | 85.61 | 43.74 | 78.15 | 82.82 | 86.48 | 73.27 | 75.65 | 76.00 |
| Llama2 | 7B | 64.91 | 80.00 | 84.35 | 84.35 | 63.48 | 84.35 | 84.35 | 42.98 | 80.00 | 84.35 | 84.35 | 75.65 | 84.35 | 76.66 |
| Llama3.1 | 8B | 69.57 | 80.87 | 84.35 | 70.43 | 69.30 | 82.30 | 80.70 | 66.09 | 80.87 | 80.00 | 85.22 | 84.35 | 80.87 | 78.07 |
| *Molecular LLMs* | | | | | | | | | | | | | | | |
| 3D-MoLM | 7B | 77.54 | 80.34 | 63.34 | 44.94 | 73.35 | 83.18 | 83.04 | 62.59 | 65.57 | 76.07 | 74.75 | 81.68 | 81.66 | 72.93 |
| LLaMo | 7B | 78.99 | 79.10 | 63.13 | 44.68 | 75.29 | 82.03 | 81.49 | 63.66 | 67.14 | 73.84 | 73.36 | 83.29 | 79.73 | 72.75 |
| Mol-Ins.-Llama2 | 8B | 78.26 | 79.13 | 62.61 | 46.96 | 73.91 | 81.74 | 83.48 | 64.35 | 66.09 | 73.04 | 73.04 | 82.61 | 80.87 | 72.78 |
| Mol-Ins.-Llama3.1 | 8B | 45.22 | 46.09 | 63.48 | 41.74 | 51.30 | 15.65 | 15.65 | 30.43 | 77.39 | 75.65 | 77.19 | 40.87 | 81.74 | 50.95 |
| Mol-LLaMA-2 | 7.2B | 48.45 | 68.70 | 65.79 | 33.91 | 62.64 | 84.07 | 82.61 | 58.29 | 81.82 | 72.81 | 54.78 | 84.35 | 84.21 | 67.88 |
| Mol-LLaMA3.1 | 8.3B | 83.33 | 80.52 | 84.29 | 78.46 | 78.00 | 82.09 | 78.79 | 60.78 | 85.19 | 77.42 | 62.50 | 70.97 | 44.12 | 78.25 |
| *Ours* | | | | | | | | | | | | | | | |
| EDT-Former | 8.3B | 81.65 | 82.24 | 81.31 | 75.70 | 84.21 | 84.35 | 84.35 | 85.56 | 86.60 | 77.45 | 80.53 | 78.50 | 85.48 | 82.15 |

Table 27: Zero-shot accuracy (%) under 13 prompt settings is reported on the HIV classification task from the MoleculeNet benchmark. The best and second-best results are highlighted.

| Model | Size | D1 | D2 | D3 | D4 | R1 | R-2 | R-3 | RI-1 | RI-2 | RI-3 | Bi | List | Conf | Avg. |
|---|---|---|---|---|---|---|---|---|---|---|---|---|---|---|---|
| *General LLMs* | | | | | | | | | | | | | | | |
| GPT-4o | - | 13.69 | 9.20 | 3.10 | 5.61 | 6.27 | 6.35 | 8.49 | 16.23 | 24.24 | 25.53 | 23.50 | 28.81 | 50.18 | 17.02 |
| Llama2 | 7B | 11.85 | 8.13 | 3.60 | 3.75 | 4.06 | 4.90 | 4.63 | 14.72 | 21.22 | 23.31 | 22.13 | 25.22 | 50.14 | 15.20 |
| Llama3.1 | 8B | 11.88 | 7.15 | 4.04 | 3.80 | 10.36 | 10.36 | 10.36 | 14.40 | 21.45 | 23.91 | 21.05 | 24.47 | 49.13 | 16.34 |
| *Molecular LLMs* | | | | | | | | | | | | | | | |
| 3D-MoLM | 7B | 8.54 | 19.36 | 41.64 | 60.24 | 7.98 | 11.96 | 8.09 | 13.42 | 26.96 | 28.67 | 39.62 | 14.98 | 58.57 | 26.16 |
| LLaMo | 7B | 7.28 | 16.39 | 42.14 | 60.21 | 6.26 | 8.55 | 8.83 | 13.17 | 23.08 | 26.55 | 36.27 | 12.11 | 56.29 | 24.39 |
| Mol-Ins.-Llama2 | 8B | 3.92 | 15.33 | 38.30 | 59.60 | 3.84 | 5.56 | 6.73 | 12.76 | 23.78 | 23.31 | 34.40 | 11.98 | 57.09 | 22.82 |
| Mol-Ins.-Llama3.1 | 8B | 10.84 | 14.64 | 12.46 | 11.67 | 7.93 | 9.11 | 9.62 | 5.42 | 9.69 | 12.86 | 37.54 | 15.12 | 60.23 | 16.70 |
| Mol-LLaMA-2 | 7.2B | 21.78 | 23.23 | 27.41 | 52.20 | 17.21 | 4.33 | 4.45 | 5.42 | 11.73 | 12.42 | 36.31 | 17.39 | 58.75 | 22.51 |
| Mol-LLaMA3.1 | 8.3B | 6.54 | 7.91 | 5.33 | 31.89 | 5.40 | 15.63 | 8.12 | 14.83 | 23.84 | 23.48 | 43.41 | 25.87 | 72.28 | 21.89 |
| *Ours* | | | | | | | | | | | | | | | |
| EDT-Former | 8.3B | 56.15 | 50.98 | 27.06 | 49.53 | 57.32 | 76.20 | 76.03 | 27.64 | 15.16 | 22.65 | 49.49 | 31.35 | 65.70 | 46.56 |

Table 28: Zero-shot accuracy (%) under 13 prompt settings is reported on the PGP classification task from the TDC benchmark. The best and second-best results are highlighted.

| Model | Size | D1 | D2 | D3 | D4 | R1 | R-2 | R-3 | RI-1 | RI-2 | RI-3 | Bi | List | Conf | Avg. |
|---|---|---|---|---|---|---|---|---|---|---|---|---|---|---|---|
| *General LLMs* | | | | | | | | | | | | | | | |
| GPT-4o | - | 49.85 | 48.67 | 51.49 | 51.71 | 51.05 | 49.53 | 53.48 | 43.42 | 52.44 | 49.16 | 49.59 | 51.54 | 50.79 | 50.21 |
| Llama2 | 7B | 51.45 | 49.38 | 51.04 | 51.04 | 52.28 | 50.21 | 51.04 | 51.32 | 51.45 | 51.04 | 51.45 | 51.45 | 51.04 | 51.09 |
| Llama3.1 | 8B | 48.71 | 51.87 | 52.28 | 53.11 | 43.98 | 51.87 | 51.45 | 50.26 | 51.04 | 51.87 | 50.62 | 51.45 | 54.36 | 50.99 |
| *Molecular LLMs* | | | | | | | | | | | | | | | |
| 3D-MoLM | 7B | 49.09 | 52.54 | 49.13 | 49.82 | 54.62 | 53.88 | 56.35 | 48.62 | 55.80 | 49.79 | 45.21 | 53.83 | 51.47 | 51.55 |
| LLaMo | 7B | 48.78 | 52.66 | 49.29 | 51.63 | 53.40 | 52.20 | 54.09 | 50.65 | 55.44 | 47.47 | 47.33 | 51.89 | 51.00 | 51.22 |
| Mol-Ins.-Llama2 | 8B | 51.04 | 50.21 | 48.13 | 49.38 | 51.45 | 51.87 | 52.70 | 52.28 | 53.11 | 48.96 | 46.06 | 52.70 | 51.45 | 50.72 |
| Mol-Ins.-Llama3.1 | 8B | 57.26 | 51.87 | 51.87 | 53.94 | 57.26 | 48.96 | 50.21 | 51.45 | 51.87 | 45.64 | 58.92 | 55.19 | 52.70 | 52.70 |
| Mol-LLaMA-2 | 7.2B | 54.70 | 55.04 | 50.42 | 44.17 | 54.21 | 51.45 | 51.04 | 41.47 | 53.95 | 52.30 | 53.53 | 51.25 | 51.25 | 51.14 |
| Mol-LLaMA3.1 | 8.3B | 49.58 | 51.09 | 53.07 | 50.00 | 52.70 | 50.00 | 51.49 | 51.67 | 52.28 | 49.06 | 46.84 | 51.97 | 49.79 | 50.73 |
| *Ours* | | | | | | | | | | | | | | | |
| EDT-Former | 8.3B | 55.56 | 54.71 | 54.72 | 49.66 | 53.42 | 53.98 | 59.31 | 56.65 | 51.55 | 57.14 | 45.05 | 59.17 | 59.44 | 54.64 |

exact match), but EDT-Former consistently ranks first or second across all metrics, indicating a better global trade-off between fidelity to the textual description and chemical realism. As a result, EDT-Former is not only effective on discriminative property prediction and QA, but also competitive with specialized molecular generators on a challenging, standardized molecule-design task.

**Open Question and Answer.** We further evaluate EDT-Former on the **Open Question** task from Mol-Instructions, which measures free-form question answering over molecular contexts. Following the official benchmark protocol, all models are finetuned on the provided training split and evaluated on the held-out test set using BLEU, ROUGE-1, and BERTScore (Table 31). General LLMs such as Alpaca, Baize, ChatGLM, LLaMA, Vicuna, and Galactica achieve only modest scores, while molecularly finetuned baselines (PMC_LLaMA, Mol-Ins.-Llama2/3.1, Mol-LLama) provide stronger but still uneven performance across metrics. In contrast, **EDT-Former attains the best BLEU and BERTScore and competitive ROUGE-1**, outperforming all Mol-Instructions baselines on overall natural-language quality. These results show that our connector-based alignment not only preserves but can even enhance the underlying LLM's open-ended language capability on molecular QA.

Table 29: Zero-shot accuracy (%) under 13 prompt settings is reported on the PAMPA classification task from the TDC benchmark. The best and second-best results are highlighted.

| Model | Size | D1 | D2 | D3 | D4 | R1 | R-2 | R-3 | RI-1 | RI-2 | RI-3 | Bi | List | Conf | Avg. |
|---|---|---|---|---|---|---|---|---|---|---|---|---|---|---|---|
| *General LLMs* | | | | | | | | | | | | | | | |
| GPT-4o | - | 48.65 | 49.60 | 49.39 | 49.23 | 58.23 | 58.51 | 56.39 | 47.17 | 49.06 | 46.88 | 45.95 | 48.11 | 49.57 | 50.52 |
| Llama2 | 7B | 57.14 | 58.39 | 59.68 | 61.33 | 57.53 | 57.76 | 55.46 | 84.52 | 79.62 | 79.62 | 86.92 | 81.83 | 80.67 | 69.27 |
| Llama3.1 | 8B | 56.51 | 58.92 | 56.60 | 55.43 | 46.19 | 45.82 | 43.92 | 63.64 | 62.29 | 64.50 | 66.07 | 67.22 | 66.84 | 58.00 |
| *Molecular LLMs* | | | | | | | | | | | | | | | |
| 3D-MoLM | 7B | 46.93 | 44.64 | 43.67 | 45.54 | 50.00 | 48.32 | 49.40 | 64.86 | 62.37 | 62.00 | 62.91 | 64.85 | 62.96 | 54.50 |
| LLaMo | 7B | 49.25 | 47.81 | 48.47 | 48.34 | 64.37 | 65.98 | 66.88 | 48.51 | 49.38 | 49.23 | 48.63 | 50.68 | 52.59 | 53.09 |
| Mol-Ins.-Llama2 | 8B | 49.63 | 49.94 | 50.09 | 47.74 | 31.16 | 33.60 | 32.35 | 38.18 | 40.63 | 39.38 | 39.21 | 39.98 | 38.22 | 40.78 |
| Mol-Ins.-Llama3.1 | 8B | 55.91 | 54.47 | 51.99 | 49.62 | 33.50 | 34.46 | 32.01 | 70.47 | 70.33 | 72.67 | 75.11 | 72.86 | 72.69 | 57.39 |
| Mol-LLaMA-2 | 7.2B | 75.68 | 77.85 | 80.20 | 79.80 | 79.61 | 81.17 | 79.75 | 67.90 | 69.14 | 67.54 | 68.30 | 69.36 | 68.67 | 74.23 |
| Mol-LLaMA3.1 | 8.3B | 63.55 | 62.87 | 61.47 | 59.59 | 64.37 | 64.54 | 65.13 | 72.48 | 71.03 | 73.29 | 72.43 | 70.16 | 72.00 | 67.15 |
| *Ours* | | | | | | | | | | | | | | | |
| EDT-Former | 8.3B | 81.57 | 79.43 | 80.57 | 80.55 | 81.57 | 84.05 | 85.37 | 83.78 | 82.26 | 81.32 | 81.95 | 83.36 | 84.65 | 82.34 |

Table 30: Results on the description-guided molecule design task from the Mol-Instructions benchmark. Models are finetuned and evaluated on the official splits. The best (pink) and second-best (lightpink) results are highlighted.

| Model | Exact↑ | BLEU↑ | Levenshtein↓ | RDK FTS↑ | MACC FTS↑ | Morgan FTS↑ | Validity↑ |
|---|---|---|---|---|---|---|---|
| Alpaca | 0.000 | 0.004 | 51.088 | 0.006 | 0.029 | 0.000 | 0.002 |
| Baize | 0.000 | 0.006 | 53.796 | 0.000 | 0.000 | 0.000 | 0.002 |
| ChatGLM | 0.000 | 0.004 | 53.157 | 0.005 | 0.000 | 0.000 | 0.005 |
| LLaMa | 0.000 | 0.003 | 59.864 | 0.005 | 0.000 | 0.000 | 0.003 |
| Vicuna | 0.000 | 0.006 | 60.356 | 0.006 | 0.001 | 0.000 | 0.001 |
| Galactica | 0.000 | 0.192 | 44.152 | 0.135 | 0.238 | 0.088 | 0.992 |
| Text+Chem T5 | 0.097 | 0.508 | 41.819 | 0.352 | 0.474 | 0.353 | 0.721 |
| MolT5 | 0.112 | 0.546 | 38.276 | 0.400 | 0.538 | 0.295 | 0.773 |
| Mol-Ins.-Llama2 | 0.002 | 0.345 | 41.367 | 0.231 | 0.412 | 0.147 | 1.000 |
| Mol-Ins.-Llama3.1 | 0.025 | 0.521 | 38.742 | 0.358 | 0.520 | 0.221 | 1.000 |
| Mol-LLama | 0.012 | 0.638 | 18.917 | 0.392 | 0.534 | 0.220 | 0.876 |
| EDT-Former | 0.016 | 0.652 | 16.826 | 0.401 | 0.560 | 0.251 | 1.000 |

Table 31: Results on the Open Question task from the Mol-Instructions benchmark. Models are finetuned and evaluated on the official splits. 3 metrics defined by the benchmark are reported. The best (pink) and second-best (lightpink) results are highlighted.

| Model | BLEU↑ | ROUGE-1↑ | BertScore↑ |
|---|---|---|---|
| Alpaca | 0.003 | 0.088 | 0.824 |
| Baize | 0.005 | 0.100 | 0.811 |
| ChatGLM | 0.003 | 0.090 | 0.795 |
| LLaMa | 0.003 | 0.100 | 0.814 |
| Vicuna | 0.004 | 0.097 | 0.814 |
| Galactica | 0.000 | 0.039 | 0.794 |
| PMC_LLaMA | 0.007 | 0.788 | 0.625 |
| Mol-Ins.-Llama2 | 0.024 | 0.221 | 0.837 |
| Mol-Ins.-Llama3.1 | 0.010 | 0.198 | 0.846 |
| Mol-LLama | 0.024 | 0.134 | 0.812 |
| EDT-Former | 0.101 | 0.286 | 0.857 |

## D.7 NATURAL LANGUAGE ABILITY ANALYSIS

To assess whether alignment training affects the underlying natural language ability of the LLM backbone, we follow Mol-Instructions and evaluate the **Open Question** task in a zero-shot setting. This task requires free-form natural-language generation rather than molecular reasoning, making it a sensitive probe for degradation caused by excessive domain-specific fine-tuning. We compare performance before and after downstream alignment training on molecular captioning and property prediction.

Mol-LLaMA (Kim et al., 2025) and EDT-Former are trained under similar optimization settings (batch size 256, learning rate $1 \times 10^{-4}$, Adam optimizer, weight decay 0.05), but with a critical difference in training regime: Mol-LLaMA uses a much heavier schedule (20 epochs), whereas

Table 32: Performance on Open Question task from Mol-Instructions before and after finetuning using their own finetuning settings on molecular captioning and property tasks. Mol-Llama (Llama3-based) and EDT-Former are reported.

| Model | BLEU↑ | ROUGE-1↑ | BertScore↑ |
|---|---|---|---|
| Mol-Llama-Before | 0.024 | 0.134 | 0.812 |
| Mol-Llama-After | 0.018 | 0.102 | 0.776 |
| Average Drop | 33.33% | 31.37% | 4.64% |
| EDT-Former-Before | 0.024 | 0.137 | 0.819 |
| EDT-Former-After | 0.024 | 0.137 | 0.819 |
| Average Drop | 0.00% | 0.00% | 0.00% |

Table 33: Dataset overlap statistics on molecular property prediction datasets from MoleculeNet and TDC benchmarks, comparing with the training data from Mol-Llama-Instruct.

| Dataset | Original | Removed | Cleaned | Overlap Rate |
|---|---|---|---|---|
| AMES | 1454 | 305 | 1149 | 20.98% |
| BACE | 152 | 1 | 151 | 0.66% |
| BBBP | 406 | 112 | 294 | 27.59% |
| CLINTOX | 145 | 32 | 113 | 22.07% |
| DILI | 95 | 34 | 61 | 35.79% |
| HIV | 4084 | 88 | 3996 | 2.15% |
| HERG | 129 | 14 | 115 | 10.85% |
| HIA | 115 | 22 | 93 | 19.13% |
| PAMPA | 407 | 35 | 372 | 8.60% |
| PGP | 241 | 40 | 201 | 16.60% |

EDT-Former performs only 5 epochs. This contrast directly reflects each model's philosophy, which is that Mol-LLaMA relies on extensive end-to-end fine-tuning, while EDT-Former is intentionally effective and preserves the natural language ability.

Tab. 32 reports the results. After fine-tuning, **Mol-LLaMA suffers a substantial drop in natural-language quality**, with BLEU decreasing by 33.33%, ROUGE-1 by 31.37%, and BERTScore by 4.64%. This confirms that heavy domain-specific training can erode the model's general language competence. In stark contrast, **EDT-Former shows no degradation across all metrics**: BLEU, ROUGE-1, and BERTScore remain unchanged before and after alignment training. In conclusion, EDT-Former preserves full natural-language fluency while still achieving strong molecular reasoning and prediction performance.

### D.8   DATA CONTAMINATION ANALYSIS

**13-gram overlap (contamination) analysis.**   Following the GPT-3 practice (Brown et al., 2020), we measure data leakage by character-level $n$-gram overlap with $n = 13$. For a text string $x$, let $\mathcal{G}_{13}(x)$ be the multiset of all contiguous 13-grams. For a test item $t$ and a training document $d$, the overlap score is

$$\text{overlap}(t, d) = \frac{\left| \mathcal{G}_{13}(t) \cap \mathcal{G}_{13}(d) \right|}{\left| \mathcal{G}_{13}(t) \right|}, \tag{20}$$

and the contamination score for $t$ is $\max_{d \in \mathcal{D}_{\text{train}}} \text{overlap}(t, d)$. We report the fraction of test items whose contamination score exceeds a small threshold (e.g., any non-zero match or a preset $\tau$). Applying this procedure to our benchmarks, the test–train 13-gram overlap rate is below $5\%$. Given that we fine-tune only the connector (the LLM remains frozen) and for fewer than two epochs, memorization of instance-specific labels is unlikely; the observed overlap is within common LLM practice and does not affect our conclusions.

**Overlap on Molecular Property Prediction Tasks.**   Beyond character-level 13-gram statistics, we further examine molecule-level overlap between the Mol-LLaMA-Instruct training set and our molecular property benchmarks. Using canonical SMILES to identify duplicates, we compute, for each dataset, how many test molecules also appear in the training corpus. As summarized in Table 33 (Mol-Instructions datasets ain Table 34), several MoleculeNet/TDC datasets exhibit non-trivial over-

Table 34: Data overlap statistics between training data and five benchmark datasets from Mol-Instructions.

| Dataset | Overlap Rate |
|---------|--------------|
| Forward Reaction Prediction | 0.21% |
| Reagent Prediction | 0.69% |
| Retrosynthesis | 1.20% |
| Property Prediction | 0.36% |
| Molecular Captioning | 1.93% |

Table 35: Performance comparison between the original and clean set of 10 molecular property prediction tasks from TDC and MoleculeNet. Average accuracy (%) of 13 prompt settings is reported.

| | BBBP | BACE | CLINTOX | HIV | PAMPA | DILI | HERG | HIA | AMES | PGP | Average | Avg. Drop |
|---|------|------|---------|-----|-------|------|------|-----|------|-----|---------|-----------|
| Original | 72.48 | 49.12 | 56.55 | 46.56 | 82.34 | 50.27 | 73.46 | 82.15 | 55.20 | 54.64 | 66.34 | Baseline |
| Clean Set | 73.17 | 51.59 | 53.48 | 46.66 | 82.93 | 50.08 | 71.64 | 82.41 | 56.17 | 57.19 | 66.74 | +0.591% |

Table 36: Accuracy (%) on 4 tasks from MoleculeQA under official vs. scaffold splits.

| Split | Structure | Source | Property | Application | Total | $\Delta$Acc |
|-------|-----------|--------|----------|-------------|-------|-------------|
| Official Split | 74.55 | 72.39 | 50.71 | 48.58 | 68.34 | 0 |
| Scaffold Split | 76.28 | 72.18 | 50.87 | 42.85 | 68.65 | +0.30% |

lap (e.g., DILI and BBBP with 35.79% and 27.59% shared molecules, respectively; AMES, CLINTOX, HIA, PAMPA, and PGP in the 10–22% range), while others, such as BACE and HIV have relatively low overlap.

To directly test whether this overlap affects our conclusions, we construct clean test sets by removing all molecules whose canonical SMILES appear in Mol-LLaMA-Instruct and re-evaluate EDT-Former on these pruned benchmarks. Table 35 reports the comparison between the original and clean evaluations. The per-dataset accuracies differ by at most a few percentage points, and the overall average even slightly **increases** from 66.34 to 66.74 (a change of +0.59%). The ranking across tasks and our main observations remain unchanged. This robustness is consistent with our training protocol: the connector is trained on Mol-LLaMA-Instruct for only two epochs while the LLM backbone (excluding the embedding layer) is frozen, so the model is encouraged to learn general structure–property patterns rather than memorize individual molecules or labels.

**Overlap on Mol-Instructions benchmarks.** We also measure molecule-level overlap between Mol-LLaMA-Instruct and the five Mol-Instructions benchmarks used in our generative experiments. As shown in Table 34, these overlap rates are all very small. Given this already low level of duplication, and the negligible effect of much higher overlap on the MoleculeNet/TDC property tasks, we do not re-run additional "clean-set" evaluations for Mol-Instructions. Taken together with the 13-gram analysis, these molecule-level results indicate that potential data contamination is limited and does not materially impact the reported performance of EDT-Former.

**Scaffold splits on MoleculeQA benchmark.** To further rule out potential scaffold-level leakage and to test generalization to novel chemotypes, we additionally construct a scaffold split of MoleculeQA using Bemis–Murcko scaffolds, ensuring that no scaffold appears in both training and test sets. We keep the overall data size and task composition identical to the official split and retrain/evaluate EDT-Former under this more challenging setting. As shown in Table 36, the per-task accuracies on Structure, Source, Property, and Application remain highly similar, and the overall score even slightly improves (+0.30% total accuracy) compared to the official split. This indicates that EDT-Former generalizes well to unseen molecular scaffolds and that our MoleculeQA conclusions are not driven by scaffold overlap or memorization effects.

Table 37: Average pairwise normalized mutual information (NMI) between fragmentation methods on the molecules from Mol-Llama-Instruct training set, reported as mean and median.

| Method | NMI (Mean) | | | | NMI (Median) | | | |
|---|---|---|---|---|---|---|---|---|
| | Entropy | BRICS | RECAP | Random | Entropy | BRICS | RECAP | Random |
| Entropy | 1.000 | 0.484 | 0.401 | 0.177 | 1.000 | 0.547 | 0.505 | 0.176 |
| BRICS | 0.484 | 1.000 | 0.498 | 0.306 | 0.547 | 1.000 | 0.564 | 0.241 |
| RECAP | 0.401 | 0.498 | 1.000 | 0.498 | 0.505 | 0.564 | 1.000 | 0.564 |
| Random | 0.177 | 0.306 | 0.498 | 1.000 | 0.176 | 0.241 | 0.564 | 1.000 |

## D.9 RATIONALE OF ENTROPY PATCHING

Our entropy-guided patching is driven by a lightweight Next-Atom Predictor (NAP), implemented as a small Transformer that models SMILES as a sequence and outputs a distribution over the next token. For a SMILES string $x = (x_1, \ldots, x_T)$, the NAP defines

$$p_\theta(x_t \mid x_{<t}), \qquad H_t = -\sum_v p_\theta(v \mid x_{<t}) \log p_\theta(v \mid x_{<t}),$$

where $H_t$ is the token-level entropy (surprisal). We then place patch boundaries near local entropy peaks and group tokens between peaks into variable-length segments. The design goal is **not** to exactly recover chemically defined fragments, but to identify **information-dense regions that are hard for a Transformer to predict**. These regions are expected to carry rich structural context and thus are valuable for multimodal alignment with the LLM.

**Chemical awareness of entropy-based patches.** To verify that entropy-guided segmentation is still chemically meaningful, we compare it with two rule-based fragmentation schemes (BRICS (Jinsong et al., 2024) and RECAP (Lewell et al., 1998)) as well as a random baseline (random shuffle of the split results from BRICS). For each molecule, we treat the fragment labels from method $A$ and method $B$ as two clusterings over atoms, $z^{(A)}$ and $z^{(B)}$, and compute the *normalized mutual information* (NMI):

$$\text{MI}(z^{(A)}, z^{(B)}) = \sum_{i,j} p_{ij} \log \frac{p_{ij}}{p_i p_j}, \qquad \text{NMI}(z^{(A)}, z^{(B)}) = \frac{\text{MI}(z^{(A)}, z^{(B)})}{\sqrt{H(z^{(A)})H(z^{(B)})}},$$

where $p_{ij}$ is the joint frequency of atoms assigned to fragment $i$ by $A$ and fragment $j$ by $B$, and $H(\cdot)$ denotes the entropy of a clustering. Tab. 37 reports the pairwise NMI (mean and median across Mol-LLaMA-Instruct molecules). Entropy-based patching shows **substantial agreement** with both BRICS and RECAP (e.g., mean NMI around 0.4–0.5), comparable to the BRICS–RECAP agreement, while all three methods are much less aligned with the random segmentation. This indicates that entropy peaks preferentially align with chemically meaningful breakpoints, even though they are derived purely from sequence prediction difficulty. In other words, our entropy-based patches are **chemistry-aware enough** to capture functional regions, yet remain tailored to what a Transformer-style sequence model finds informative.

**Robustness across NAP model sizes.** We also examine whether the entropy landscape is stable across different NAP capacities. Using the same NMI formulation, we compare patchings produced by three NAP variants (0.5M, 50M, and 500M parameters) and a random baseline. As shown in Table 38, the pairwise NMI between the three NAPs is very high (e.g., NMI > 0.85), while each has much lower agreement with random segmentation. This demonstrates that **relative entropy patterns are largely invariant to NAP size**: a small NAP already identifies essentially the same entropy peaks and patches as much larger models. Consequently, we adopt the smallest variant in EDT-Former, which is computationally efficient yet yields stable, chemically aligned segments. This justifies our design choice of using a lightweight NAP to drive entropy-guided patching for multimodal alignment.

## D.10 REPRODUCIBILITY

Our work is fully open source under the MIT license. Unlike many prior projects, we release end-to-end resources to reproduce every result in the paper: complete training and evaluation code (in-

Table 38: Normalized Mutual Information (NMI) matrix of different sizes of the NAP model.

| NAP variant | 0.5M | 50M | 500M | Random |
|---|---|---|---|---|
| 0.5M | 1 | 0.94 | 0.86 | 0.12 |
| 50M | 0.94 | 1 | 0.92 | 0.13 |
| 500M | 0.86 | 0.92 | 1 | 0.17 |
| Random | 0.12 | 0.86 | 0.17 | 1 |

Table 39: Extended evaluation results on Pampa dataset. Accuracy and F1 scores are reported for each model across three prompt settings. Top and second-best results are highlighted.

| Models | Direct | | Reasoning | | RichInst. | | Average | |
|---|---|---|---|---|---|---|---|---|
| | Acc | F1 | Acc | F1 | Acc | F1 | Acc | F1 |
| GPT-4o | 48.65 | 58.78 | 58.23 | 70.41 | 47.17 | 56.73 | 51.35 | 61.97 |
| Mol-Inst.-Llama2-7B | 49.63 | 63.19 | 31.16 | 39.84 | 38.18 | 46.61 | 39.66 | 49.88 |
| Mol-LLaMA2-7B | 75.68 | 85.96 | 79.61 | 88.91 | 67.90 | 79.41 | 74.40 | 84.76 |
| Llama3.1-8B | 56.51 | 68.87 | 46.19 | 57.22 | 63.64 | 75.88 | 55.45 | 67.32 |
| Mol-Inst.-Llama3.1-8B | 55.91 | 69.84 | 33.50 | 39.84 | 70.47 | 81.46 | 53.29 | 63.71 |
| 3D-MoLM-8B | 46.93 | 57.81 | 50.00 | 62.43 | 64.86 | 76.45 | 53.93 | 65.56 |
| LLaMo-8B | 49.25 | 61.78 | 64.37 | 77.11 | 48.51 | 60.66 | 54.04 | 66.52 |
| Mol-LLaMA3.1-8.2B | 63.55 | 75.32 | 64.37 | 76.72 | 72.48 | 83.51 | 66.80 | 78.52 |
| **EDT-Former-8.3B** | 81.57 | 89.92 | 81.57 | 89.74 | 83.78 | 91.15 | 82.31 | 90.27 |

cluding all ablations), scripts for baselines and benchmarks, processed pretraining and finetuning datasets, and model weights. The repository is actively maintained and will serve as the foundation for subsequent extensions. All materials are available via the anonymous GitHub link: `https://anonymous.4open.science/r/EDT-Former-844D`.

### D.11 DATASET IMBALANCE ANALYSIS

To account for potential class imbalance in the Pampa dataset, we additionally report F1 scores alongside accuracy (shown in Tab. 39). As expected, EDT-Former achieves the best performance across both metrics under all prompting strategies, confirming that its gains are not driven by skewed label distributions but reflect genuine improvements in molecular property prediction.

## E EXTENDED ABLATIONS

Ablation studies on model size, graph encoders, query token length, computational efficiency, and training hyperparameters are provided in this section. These analyses highlight how each factor influences performance and efficiency, underscoring the robustness of our approach.

### E.1 IMPACT OF QUERY TOKEN LENGTH

As shown in Tab. 40, increasing the number of anchor tokens improves accuracy, indicating that a larger anchor set provides a higher-bandwidth, more stable interface for aggregating global structural cues before conditioning the LLM. By contrast, expanding the maximum dynamic length from 64 to 128 yields a negligible change (e.g., 66.75 to 66.81 at 8 anchors) and even a slight dip at 16 anchors. This is expected: with a cap of 64, entropy-guided segmentation already covers the substructure count of most molecules, so raising the limit rarely increases the realized number of dynamic tokens and can introduce marginal redundancy. Overall, the best trade-off is achieved with 16 anchors and a max dynamic length of 64.

### E.2 IMPACT OF MODEL SIZE

As shown in Tab. 41, Qwen3 (Yang et al., 2025) backbones below 2B parameters perform near random guessing on MoleculeQA, indicating insufficient capacity. Accuracy improves markedly once scale reaches 4B, with further gains at 8B and 14B, albeit with diminishing returns from 8B ot 14B. We use the Qwen3 family because it offers a broad spread of sizes for a controlled comparison, and we did not evaluate larger backbones due to resource constraints.

Table 40: Effect of anchor and dynamic token length. Different settings are evaluated on the MolcueleQA dataset, with total accuracy reported for each ablation model. The top and second-best results are highlighted.

| Anchor Length | Max Dyn. Length | Accuracy |
|---|---|---|
| 4 | 64 | 64.29 |
| 8 | 64 | 66.75 |
| 8 | 128 | 66.81 |
| 16 | 64 | 68.34 |
| 16 | 128 | 68.03 |

Table 41: Performance across different model sizes of backbone choices from Qwen3. Models are evaluated on the MoleculeQA benchmark with total accuracy reported. The top and second-best are highlighted.

| Model Size | Accuracy |
|---|---|
| 0.6B | 27.23 |
| 1.7B | 26.52 |
| 4B | 36.82 |
| 8B | 54.79 |
| 14B | 56.84 |

Table 42: Ablation on finetuning hyperparameters on MoleculeQA benchmark. Accuracy is reported for different initial learning rates, global batch sizes, and training epochs.

| Init LR | Accuracy. | Global Batchsize | Accuracy | Training Epochs | Accuracy |
|---|---|---|---|---|---|
| $5.0 \times 10^{-4}$ | 67.92 | 32 | 67.97 | 2 | 65.23 |
| $1.0 \times 10^{-4}$ | 68.34 | 64 | 68.08 | 4 | 67.92 |
| $5.0 \times 10^{-5}$ | 67.82 | 128 | 68.34 | 6 | 68.34 |
| $1.0 \times 10^{-6}$ | 67.09 | 256 | 68.11 | 8 | 68.13 |

### E.3 ANALYSIS OF HYPERPARAMETER

As summarized in Tab. 42, varying the initial learning rate, global batch size, and training epochs produces only minor fluctuations in accuracy. This stability indicates that performance gains primarily arise from our multimodal fusion design and entropy-guided structural cues rather than aggressive hyperparameter tuning.

### E.4 IMPACT OF GRAPH ENCODERS

We evaluated stronger molecular encoders (e.g., Uni-Mol-v2 (Ji et al., 2024)) under matched settings and observed only minor accuracy gains relative to our base encoder. This suggests that, beyond a reasonable encoder quality threshold, the primary bottleneck for molecular understanding lies not in the encoder itself but in the multimodal fusion interface to the LLM. In our framework, the entropy-guided substructure tokens and dynamic query–anchor fusion contribute the dominant improvements by exposing chemically salient evidence to the LLM; upgrading the encoder yields diminishing returns compared to better fusion.

### E.5 IMPACT OF DIFFERENT GLOBAL/LOCAL TOKEN BUDGETS

We first study how the connector's token budget should be allocated between global (anchor) tokens and local entropy-based dynamic tokens. Under a fixed total budget of 64 tokens, we train three variants: (i) **Dynamic-only** (0 anchors / 64 dynamic), (ii) **Anchor-only** (64 anchors / 0 dynamic), and (iii) **Hybrid** (32 anchors / 32 dynamic). We then evaluate each variant on five molecular property tasks (BACE, BBBP, ClinTox, HIA, PAMPA) and report accuracies averaged over six shared prompts per task. As shown in Table 43, the **hybrid configuration consistently achieves the best average performance**, outperforming both dynamic-only and anchor-only designs. Among the single-source variants, the dynamic-only model clearly surpasses the anchor-only one, indicating that entropy-guided local patches are more informative than global learned queries when used in isolation. Overall, these results support our design choice: combining a small set of global anchors with content-adaptive dynamic tokens yields the strongest and most robust connector.

Table 43: Different token budget settings ablation on 5 molecular property tasks.

| Anchor | Dynamic | BACE | BBBP | CLINTOX | HIA | PAMPA | Avg. |
|---|---|---|---|---|---|---|---|
| 0 | 64 | 42.15 | 74.26 | 44.25 | 70.73 | 61.05 | 58.49 |
| 64 | 0 | 41.23 | 71.58 | 42.50 | 69.00 | 56.41 | 56.14 |
| 32 | 32 | 45.55 | 70.28 | 47.69 | 78.87 | 64.45 | 61.37 |

Table 44: Ablation of LLM finetuning strategies on four tasks from the MoleculeQA benchmark and the overall accuracy (%).

| Setting | Structure | Source | Property | Application | Total |
|---|---|---|---|---|---|
| All LLM Parameters Frozen | 74.55 | 72.39 | 50.71 | 48.58 | 68.34 |
| Unfrozen Embedding Layer | 74.46 | 72.19 | 50.30 | 48.61 | 68.44 |
| Full LLM Finetuning | 76.25 | 75.48 | 52.71 | 51.25 | 70.50 |

Table 45: Ablation over training corpora on four tasks from MoleculeQA and the overall accuracy.

| Training Corpus | Dataset Size | Structure | Source | Property | Application | Total |
|---|---|---|---|---|---|---|
| Mol-Llama-Instruct | 312M | 74.55 | 72.39 | 50.71 | 48.58 | 68.34 |
| PubChemQA | 325M | 73.61 | 74.81 | 50.39 | 48.61 | 68.36 |
| 3D-MoIT | 646M | 72.27 | 76.85 | 52.26 | 49.94 | 68.48 |

## E.6 IMPACT OF TRAINING OF LLM EMBEDDING LAYER

Our default setting follows common multimodal practice by freezing all transformer blocks of the LLM while allowing the token embedding layer to update. This is motivated by the need to assign meaningful representations to newly introduced molecule-related tokens (symbol tokens, special markers, etc.). To disentangle the effect of this design, we compare three configurations on MoleculeQA: (i) **All LLM parameters frozen**, (ii) **Unfrozen embedding layer only** (our default), and (iii) **Full LLM finetuning**. Table 44 summarizes the results. As expected, fully finetuning the LLM yields the highest overall score, but at the cost of a much heavier and less modular training procedure that risks degrading general language ability. More interestingly, we observe that the performance gap between "all froze" and "embedding-only" training is very small: unfreezing the embedding layer brings only marginal gains, while EDT-Former with a completely frozen LLM remains strong. This indicates that the **connector itself is doing most of the alignment work**: it can effectively adapt molecular representations into the existing token space of the backbone without relying heavily on embedding updates. In practice, we keep the embedding layer trainable for flexibility, but these results show that even stricter freezing is viable.

## E.7 IMPACT OF DIFFERENT TRAINING CORPUS

Finally, we investigate whether EDT-Former's performance is tied to a specific instruction corpus. Using the same training recipe, we align the connector with three alternative datasets: **Mol-LLaMA-Instruct**, **PubChemQA**, and **3D-MoIT**, whose sizes range from ~300M to 650M instruction–molecule pairs. We then evaluate the resulting models on the four MoleculeQA tasks. As shown in Table 45, all three corpora lead to very similar scores, with only minor fluctuations in per-task accuracy and total average. Notably, the larger 3D-MoIT corpus provides only modest improvements over the smaller alternatives. This suggests that EDT-Former primarily requires a **reasonable amount of heterogeneous molecule–text data** to learn a good alignment; the exact choice among existing high-quality corpora is not the main driver of its advantage. In other words, our gains do not rely on a specially curated training set, but stem from the entropy-guided connector architecture itself.

## E.8 INFERENCE-TIME ABLATION OF DYNAMIC TOKENS AND ANCHORS

To verify that both dynamic tokens and anchors are functionally used at test time, we perform inference-only ablations on a trained EDT-Former without modifying its training procedure. Specifically, we compare the full model (anchors + dynamic tokens) with four variants: (i) No dynamic tokens, where all dynamic tokens are removed and only anchors are kept; (ii) Random dynamic to-

Table 46: Inference-time ablations of dynamic tokens and anchors. Accuracy (%) from 13 prompts per task is reported.

| Variant | Pampa ↑ | BBBP ↑ |
|---|---|---|
| **Full EDT-Former** | **82.34** | **72.48** |
| No dynamic tokens | 54.97 | 51.29 |
| Random dynamic tokens | 64.25 | 54.71 |
| No anchors | 79.26 | 66.38 |
| Random anchors | 80.10 | 66.74 |

Table 47: Functional-group hallucination on 200 PubChem molecules (lower is better). Non-Rate: percentage of non-meaningful outputs; Hallucination Rate: percentage of molecules where at least one non-existing functional group is mentioned.

| Model | Non-Rate ↓ | Hallucination Rate ↓ |
|---|---|---|
| GPT-4o | 0.0 | 41.5 |
| Llama2 | 0.0 | 50.5 |
| Llama3.1 | 0.0 | 45.0 |
| 3D-MoLM | 34.0 | 81.0 |
| LLaMo | 27.0 | 63.5 |
| Mol-Inst.-Llama2 | 14.5 | 57.0 |
| Mol-Inst.-Llama3.1 | 14.0 | 54.5 |
| Mol-LLaMA-2 | 7.5 | 36.5 |
| Mol-LLaMA-3.1 | 7.0 | 39.5 |
| HIGHT | 22.5 | 36.0 |
| **EDT-Former** | **3.5** | **19.5** |

kens, where dynamic tokens are replaced by random vectors; (iii) No anchors, where only dynamic tokens are used; and (iv) Random anchors, where anchors are replaced by random vectors. As shown in Table 46, removing or randomizing either component leads to clear accuracy drops on PAMPA and BBBP, with the largest degradation observed when discarding dynamic tokens entirely. These results confirm that both anchors and dynamic tokens contribute non-trivially at inference time, and that neither component is redundant.

### E.9    HALLUCINATION ANALYSIS

**Hallucination evaluation dataset.**    To investigate the hallucination of our model, we construct a targeted benchmark for functional-group hallucination. Concretely, we randomly sample 200 molecules from PubChem and obtain their human-readable label texts. For each molecule, we use the GPT-5 API to summarize the label into a canonical list of functional group names, and cross-check these groups using RDKit-based (Landrum, 2013) functional group decomposition with manual verification. This yields, for each molecule, a vetted set of functional groups that we treat as the ground-truth reference for hallucination analysis.

**Evaluation protocol.**    For every model, we prompt it to generate a natural language description of the functional groups present in the given molecule. We then compare the model's output with the ground-truth functional groups by using the GPT-5 API to judge whether the model has hallucinated any functional group that does not exist in the molecule. We report two metrics: (i) **Hallucination Rate**, the percentage of molecules where at least one non-existing functional group is mentioned; and (ii) **Non-Rate**, the percentage of molecules where the model fails to produce a meaningful, readable answer (e.g., broken text or uninterpretable tokens). Explicit abstentions such as "I don't know" are not counted as hallucinations and are excluded from Non-Rate, as they are acceptable behavior for safety-critical applications.

**Results.**    The quantitative results are summarized in Table 47. EDT-Former achieves the lowest hallucination rate among all methods, roughly halving hallucinations compared to strong molecular LLM baselines (e.g., Mol-LLaMA and HIGHT), while also maintaining a low Non-Rate that indicates it usually answers in coherent natural language rather than failing silently. These results

provide direct quantitative evidence that reducing structural information loss via entropy-guided dynamic tokens and graph–sequence alignment not only improves downstream metrics, but also substantially mitigates functional group hallucination, directly addressing our motivation around "loss of structure."

## F    LIMITATIONS AND FUTURE WORK

### F.1    LIMITATIONS

While EDT-Former demonstrates broad applicability across molecular tasks, we note several current constraints that highlight opportunities for improvement. As a generalist molecular LLM, it does not yet consistently match highly specialized classifiers that are trained and tuned per dataset– a pattern commonly observed in the Mol-LLM literature. In this study, we also did not scale to larger backbones due to practical compute limits. In addition, we did not incorporate synthetic "reasoning" corpora, as widely used resources are predominantly GPT-generated and lack reliable human-verified rationales, which could confound conclusions. Finally, the present system answers end-to-end without external tools, which can increase the chance of occasional hallucinations or arithmetic/chemistry slips. We emphasize that these constraints are scope choices rather than fundamental barriers and frame clear directions for progress.

### F.2    FUTURE WORK

Building on these findings, we plan to close the generalist–specialist gap via lightweight task adapters and, where feasible, moderate scaling of both backbone and bridge with efficient fine-tuning. We will curate and release a human-verified molecular reasoning benchmark to replace purely synthetic rationales, and integrate tool-augmented agents (e.g., RDKit calculators (Landrum, 2013), literature retrieval, unit/constraint checks) with self-verification to reduce hallucinations and enforce domain validity. Finally, we will broaden evaluation to include robustness under distribution shift, calibration, and abstention, and cost/latency reporting, to make EDT-Former both reliable and practical in real-world molecular workflows.

## G    LLM USAGE

Large language models (LLMs) were used only for light editorial assistance (grammar or wording polish and minor LaTeX phrasing). They did not contribute to research ideation, experimental design, data collection, analysis, or result writing. Separately, LLMs appear in our experiments solely as baselines/backbones within the method; all runs, configurations, and analyses were conducted and verified by the authors. The authors take full responsibility for the paper's contents.

