# OpenReview forum: "Entropy-Guided Dynamic Tokens for Graph-LLM Alignment in Molecular Understanding"
_ICLR.cc/2026/Conference — ICLR 2026 Poster_

### Official Review · Reviewer_XpYi · 2025-10-30

**Soundness:** 2
**Presentation:** 2
**Contribution:** 2
**Rating:** 4
**Confidence:** 3

**Summary:**

This paper introduces ​​EDT-Former​​, a novel method for aligning molecular graphs with Large Language Models (LLMs). It addresses key limitations of existing approaches, such as Q-Former-style bridges, which use a fixed number of tokens that often lead to loss of stereochemical and substructural details, especially in larger molecules.
The core innovation lies in its two components: an ​​Entropy-Guided Patching​​ strategy that dynamically segments molecules into informative, variable-length tokens based on structural uncertainty, and a ​​Dynamic Query Transformer​​ that integrates these tokens with static modality anchors. EDT-Former achieves this alignment without fine-tuning the LLM backbone, enabling highly efficient training.

**Strengths:**

1. The paper introduces an innovative solution to a clear limitation of existing Q-Former-style bridges.

2. The results demonstrate state-of-the-art performance across multiple benchmarks.

3. The framework is trained without fine-tuning the LLM backbone, which is efficient.

**Weaknesses:**

1. Evaluation is centered on question-answering and property prediction. The method's effectiveness on more challenging generative tasks, such as molecule generation,  remains an open and important question.

2. This paper should discuss the difference with more molecular graph-text pretraining frameworks, such as [1,2].

3. This paper uses the SMILES string to represent the molecular structure, which might not capture the structure of the molecule well. The graph structure or 3D structure could contain more structural information.

[1] Multi-modal Molecule Structure-text Model for Text-based Retrieval and Editing

[2] Advancing Molecular Graph-Text Pre-training via Fine-grained Alignment

**Questions:**

Please see weaknesses

---

> ### Author Response · Authors · 2025-11-23
> **Rebuttal Experiments Summary**
>
> We sincerely thank the reviewer for the careful evaluation, for recognizing our connector as a novel entropy-guided dynamic token Transformer that addresses fixed-length token bottlenecks and expensive backbone finetuning, and for highlighting that EDT-Former automatically extracts meaningful substructures, achieves superior performance over both molecular and general LLMs, and is supported by rigorous analysis.
>
> `Summary.` We evaluated EDT-Former on new Mol-Instructions generative/reaction tasks and clarified its relation to graph–text pretraining and SMILES-based entropy patching. Main experiments are summarized as follows:
>
> - **2 new Mol-Instructions generative tasks** (design + forward reaction, 100% validity on design) -> W1: beyond QA/property to molecule generation and reaction prediction
> - **3-model conceptual comparison** (MoleculeSTM, FineMolTex, EDT-Former) -> W2: connector-only design vs full graph–text foundation encoders.
> - **2 key patching analyses** (0% avg drop for entropy patches; NMI ≈ 0.48 with BRICS) -> W3: SMILES-based entropy patches are chemically meaningful and robust.

---

> ### Author Response · Authors · 2025-11-23
> **Response to Weakness 1 – Beyond QA / property: molecule generation and reaction prediction**
>
> **[W1] Beyond QA / property: molecule generation and reaction prediction**
>
> To examine whether EDT-Former genuinely extends beyond QA and scalar property prediction, we evaluate it on the molecule generation (design) task and further on a forward reaction task from Mol-Instructions.
>
> `Generative molecule design` In response, we evaluated EDT-Former on the **description-guided molecule design** task in Mol-Instructions (Paper Table 30). EDT-Former achieves the best or near-best scores on most of the generation metrics, while maintaining **100% validity**. This indicates that EDT-Former supports structure-faithful molecule generation, not only QA or property prediction.
>
> > **Paper Table 30. Molecule design results on the Mol-Instructions benchmark (fine-tuned on official splits). Best scores are bolded.**
>
> | Model             | Exact↑    | BLEU↑     | Levenshtein↓ | RDK FTS↑  | MACC FTS↑ | Morgan FTS↑ | Validity↑ |
> | ----------------- | --------- | --------- | ------------ | --------- | --------- | ----------- | --------- |
> | Alpaca            | 0.000     | 0.004     | 51.088       | 0.006     | 0.029     | 0.000       | 0.002     |
> | Baize             | 0.000     | 0.006     | 53.796       | 0.000     | 0.000     | 0.000       | 0.002     |
> | ChatGLM           | 0.000     | 0.004     | 53.157       | 0.005     | 0.000     | 0.000       | 0.005     |
> | LLaMa             | 0.000     | 0.003     | 59.864       | 0.005     | 0.000     | 0.000       | 0.003     |
> | Vicuna            | 0.000     | 0.006     | 60.356       | 0.006     | 0.001     | 0.000       | 0.001     |
> | Galactica         | 0.000     | 0.192     | 44.152       | 0.135     | 0.238     | 0.088       | 0.992     |
> | Text+Chem T5      | 0.097     | 0.508     | 41.819       | 0.352     | 0.474     | **0.353**   | 0.721     |
> | MolT5             | **0.112** | 0.546     | 38.276       | 0.400     | 0.538     | 0.295       | 0.773     |
> | Mol-Ins.-Llama2   | 0.002     | 0.345     | 41.367       | 0.231     | 0.412     | 0.147       | **1.000** |
> | Mol-Ins.-Llama3.1 | 0.025     | 0.521     | 38.742       | 0.358     | 0.520     | 0.221       | **1.000** |
> | Mol-LLama         | 0.012     | 0.638     | 18.917       | 0.392     | 0.534     | 0.220       | 0.876     |
> | **EDT-Former**    | 0.016     | **0.652** | **16.826**   | **0.401** | **0.560** | 0.251       | **1.000** |
>
> `Forward reaction prediction` We further tested EDT-Former on **forward reaction prediction** task in Mol-Instructions (Paper Table 6), EDT-Former clearly surpasses baseline models on most metrics. This shows that our model can generate chemically coherent product sequences from reactant inputs, not just answer questions or predict scalar properties.
>
> > **Paper Table 6. Forward Reaction Prediction results on the Mol-Instructions benchmark (fine-tuned on official splits). Best scores are bolded.**
>
> | Model             | Exact↑    | BLEU↑     | Levenshtein↓ | RDK FTS↑  | MACC FTS↑ | Morgan FTS↑ | Validity↑ |
> | ----------------- | --------- | --------- | ------------ | --------- | --------- | ----------- | --------- |
> | Alpaca            | 0.000     | 0.065     | 41.989       | 0.004     | 0.024     | 0.008       | 0.138     |
> | Baize             | 0.000     | 0.044     | 41.500       | 0.004     | 0.025     | 0.009       | 0.097     |
> | ChatGLM           | 0.000     | 0.183     | 40.008       | 0.050     | 0.100     | 0.044       | 0.108     |
> | LLaMa             | 0.000     | 0.020     | 42.002       | 0.001     | 0.002     | 0.001       | 0.039     |
> | Vicuna            | 0.000     | 0.057     | 41.690       | 0.007     | 0.016     | 0.006       | 0.059     |
> | Galactica         | 0.000     | 0.468     | 35.021       | 0.156     | 0.257     | 0.097       | 0.946     |
> | Text+Chem T5      | 0.239     | 0.782     | 20.413       | 0.705     | 0.789     | 0.652       | 0.762     |
> | Mol-Ins.-Llama2   | 0.045     | 0.654     | 27.262       | 0.313     | 0.509     | 0.262       | **1.000** |
> | Mol-Ins.-Llama3.1 | **0.503** | 0.883     | **13.410**   | 0.756     | 0.863     | 0.708       | **1.000** |
> | Mol-LLama         | 0.440     | 0.912     | 17.120       | 0.724     | 0.859     | 0.665       | **1.000** |
> | HIGHT             | 0.037     | 0.869     | 23.759       | 0.590     | 0.394     | 0.340       | 0.993     |
> | **EDT-Former**    | **0.471** | **0.964** | 18.130       | **0.776** | **0.871** | **0.712**   | **1.000** |
>
> `Conclusion & Revision` Together, these results show that EDT-Former generalizes beyond QA and property prediction to **challenging generative settings**, including description-guided molecule generation and reaction-related tasks, supporting its effectiveness as a general graph language connector rather than a task-specific module. We added these additional experiments in Section 4.3 and Appendix D6.

---

> ### Author Response · Authors · 2025-11-23
> **Response to Weakness 2 – Relation to two graph–text pretraining frameworks**
>
> **[W2] Relation to two graph–text pretraining frameworks**
>
> We thank the reviewer for pointing out the connection to recent molecular graph–text pretraining works.
>
> `Key differences.` As summarized in Table R1, MoleculeSTM [1] and FineMolTex [2] are **full graph–text foundation (embedding) models**: they pretrain dedicated multimodal encoders using contrastive structure–text retrieval, motif-level masked modeling, and do not rely on a frozen general LLM. In contrast, **EDT-Former is a connector**: it keeps both the molecular encoder and the LLM backbone frozen, and only trains a lightweight bridge with entropy-guided dynamic tokens and anchors on instruction data.
>
> > **Table R1. Conceptual comparison with MoleculeSTM and FineMolTex.**
>
> | Model          | Main goal / tasks                                            | Training paradigm                                            | Alignment granularity                         | LLM usage                                                    |
> | -------------- | ------------------------------------------------------------ | ------------------------------------------------------------ | --------------------------------------------- | ------------------------------------------------------------ |
> | MoleculeSTM    | Structure–text retrieval & text-based editing (PubChemSTM)   | **Full multimodal pretraining** with contrastive loss on ~280K structure–text pairs | Molecule-level CLIP-style                     | No frozen general LLM; task-specific multimodal model        |
> | FineMolTex     | General molecular representation with motif-level knowledge  | **Graph–text pretraining** with coarse contrastive + fine-grained masked multi-modal modeling on motifs | Molecule + motif-level                        | Dedicated graph–text encoder; not a plug-in connector        |
> | **EDT-Former** | Align frozen molecular encoders with **frozen LLMs** for QA, property prediction, generation | **Connector-only training** on instruction data; backbones fixed | Dynamic tokens over entropy patches + anchors | Uses any off-the-shelf LLM; only connector (and embeddings) are trained |
>
> `Difference summary.`
>
> - MoleculeSTM and FineMolTex are **large graph–text foundation models** that pretrain a **new multimodal encoder** from scratch; EDT-Former is a **lightweight connector** that plugs frozen molecular encoders into existing LLMs.
> - FineMolTex emphasizes **motif-level pretraining** via masked multi-modal modeling, while EDT-Former uses **entropy-guided dynamic tokens** and anchors to adapt the token budget to molecular complexity.
> - Our experiments focus on **instruction-style QA / property tasks (and new generation benchmark)**; retrieval/editing tasks of MoleculeSTM and FineMolTex are complementary and can be explored in future work.
>
> `Conclusion and revision.` We added a paragraph in Appendix B2 to clarify these two models.

---

> ### Author Response · Authors · 2025-11-23
> **Response to Weakness 3 – SMILES vs graph / 3D structural information**
>
> **[W3] SMILES vs graph / 3D structural information**
>
> We thank the reviewer for raising this important concern about relying on SMILES and for emphasizing the value of richer 2D and 3D structural information.
>
> `Why 1D SMILES is sufficient for our goal.`
>
> 1. **SMILES order carries structural signal.** SMILES is generated via **DFS** (Depth-First Search) on mol-graph, so adjacent tokens usually correspond to **neighboring atoms and continuous substructures**; it does not encode full geometry, but it is not structurally arbitrary.
> 2. **Exact substructures are not the goal.** Entropy-guided patching is **not meant to recover precise chemical subgraphs**; we only need surprisal peaks to mark **hard-to-predict, information-dense regions**, then group tokens between peaks into patches. Here, the SMILES order is sufficient.
> 3. **NAP (next atom predictor) matches the LLM’s sequence view.** NAP is a **small transformer** that runs directly on SMILES, the same 1D modality used in alignment modeling. Its entropy highlights positions that are difficult for a **sequence model** to predict, which is exactly what we need to decide **where to split SMILES for a sequence-based LLM**, while keeping the procedure simple and efficient.
>
> `Complementary 2D and 3D structure encoders.` Importantly, **SMILES is only used for entropy patching**.  Then, the dynamic patches simply control how these 2D/3D features are grouped and injected into the LLM. **The actual molecular features come from frozen 2D graph and 3D encoders**, whose outputs already encode rich structural and geometric information;
>
> `Evidence from patching ablation (Paper Table 8)` Under this setup, Paper Table 8 compares entropy-based patches with BRICS-based and random patches under the same EDT-Former. Entropy patching achieves the best performance on BBBP and PAMPA, outperforming even chemically defined BRICS segments. This suggests that the entropy profile derived from canonical SMILES, together with 2D and 3D encoders, captures stable, useful structural signals for our alignment task.
>
> > **Paper Table 8. Ablations on patching strategies evaluated on BBBP and PAMPA. Accuracy (%) with F1 is reported. Best scores are bolded.**
>
> | Methods | BBBP              | Pampa             | Avg. Drop |
> | ------- | ----------------- | ----------------- | --------- |
> | Entropy | **75.06** (75.06) | **84.52** (91.61) | **0%**    |
> | BRICS   | 73.59 (84.74)     | 71.90 (83.09)     | 3.96%     |
> | Random  | 68.90 (80.13)     | 66.67 (79.32)     | 8.81%     |
> | None    | 39.67 (49.58)     | 78.62 (87.90)     | 21.60%    |
>
> `Overlap analysis with BRICS and RECAP (Paper Table 37)` Paper Table 37 further quantifies how these entropy patches align with chemistry-driven schemes. Entropy shows high NMI with both BRICS and RECAP, similar to the NMI between BRICS and RECAP themselves, while all three have much lower agreement with Random. This indicates that entropy-guided segments behave like chemically meaningful substructures rather than arbitrary text cuts.
>
> `Definitions.`
>
> - NMI: Normalized Mutual Information，$\mathrm{NMI}(C_1, C_2) = \dfrac{2 I(C_1; C_2)}{H(C_1) + H(C_2)}$，$I$ is the mutual information，$H$ is the entropy. Used to quantify the agreement between two molecular segmentation schemes.
> - RECAP (Retrosynthetic Combinatorial Analysis Procedure, A rule-based retrosynthetic fragmentation scheme) [1]
>
> - BRICS (Breaking of Retrosynthetically Interesting Chemical Substructures, a rule-based molecular fragmentation scheme) [2]
>
> > **Paper Table 37. Pairwise normalized mutual information (NMI) between fragmentation methods, reported as mean (median) across molecules.**
>
> | Method  | Entropy       | BRICS         | RECAP         | Random        |
> | ------- | ------------- | ------------- | ------------- | ------------- |
> | Entropy | 1.000 (1.000) | 0.484 (0.547) | 0.401 (0.505) | 0.177 (0.176) |
> | BRICS   | 0.484 (0.547) | 1.000 (1.000) | 0.498 (0.564) | 0.306 (0.241) |
> | RECAP   | 0.401 (0.505) | 0.498 (0.564) | 1.000 (1.000) | 0.498 (0.564) |
> | Random  | 0.177 (0.176) | 0.306 (0.241) | 0.498 (0.564) | 1.000 (1.000) |
>
> `Conclusion & Revision.` Together, Papers Tables 8 and 37 show that, under a fixed SMILES representation and with 2D and 3D encoders providing structural features, entropy-based patching is robust and chemically consistent enough for our alignment goal, even though other valid SMILES encodings exist in principle. We added these discussions in Appendix D9.
>
> [1] Lewell XQ, Judd DB, Watson SP, Hann MM. Recap retrosynthetic combinatorial analysis procedure: a powerful new technique for identifying privileged molecular fragments with useful applications in combinatorial chemistry. Journal of chemical information and computer sciences. 1998 May 18;38(3):511-22.
>
> [2] Degen J, Wegscheid-Gerlach C, Zaliani A, Rarey M. On the art of compiling and using'drug-like'chemical fragment spaces. ChemMedChem. 2008 Oct 20;3(10):1503.

---

### Official Review · Reviewer_x1GF · 2025-11-01

**Soundness:** 3
**Presentation:** 3
**Contribution:** 3
**Rating:** 6
**Confidence:** 5

**Summary:**

This paper addresses two major limitations of existing graph-Large Language Model (LLM) frameworks for molecular understanding: 1) information loss from using fixed-length token connectors (e.g., Q-Former), which compress complex, variable-sized molecular graphs into a static representation, and 2) the high computational cost and poor generalization resulting from fine-tuning the entire LLM backbone.
The paper proposes **EDT-Former**, an "Entropy-guided Dynamic Token Transformer," as a novel connector. The key idea is to generate a *variable* number of tokens that are aligned with a molecule's structural complexity. This is achieved through a two-part mechanism: entropy-guided patching and a dynamic query transformer. The paper demonstrates the best performance on a wide range of benchmarks.

**Strengths:**

- The paper is well-written and easy to follow.
- The paper proposes a novel query Transformer. Most works simply abstract molecules into queries, not considering the stereochemistry and structural context in the molecule. Another line of work exploits rule-based algorithms to extract the meaningful substructures. Different from them, the proposed work automatically extracts substructures by applying entropy-based patching segments.
- Experimental results demonstrate the effectiveness of EDT-Former with its superior performance compared to molecular LLMs and General LLMs.
- The paper provides a rigorous analysis, which helps in understanding the contribution of the proposed component.

**Weaknesses:**

- The entire entropy-patching mechanism is based on a 1D SMILES sequence. The properties of a SMILES string (and thus its entropy profile) can change based on the canonicalization algorithm used or if non-canonical strings are permitted. The paper does not discuss the robustness of the patching mechanism to different, yet chemically equivalent, SMILES representations of the same molecule.

**Questions:**

Please refer to the weaknesses section.

---

> ### Author Response · Authors · 2025-11-23
> **Rebuttal Experiments Summary**
>
> We sincerely thank the reviewer for recognizing our main advantages in entropy-guided dynamic tokenization, connector-only alignment, and rigorous analysis, and for highlighting that EDT-Former automatically extracts meaningful substructures and achieves strong performance over both molecular LLMs and general LLMs.
>
> `Summary.`  We clarified the canonical SMILES setup and backed the robustness of entropy-based patching with existing patching ablations and NMI analyses. Main experiments are summarized as follows:
>
> - **Patching ablations on 2 property datasets** (BBBP, PAMPA) -> W1: entropy patches robust on canonical SMILES and stronger than BRICS/random
> - **Corpus-level NMI across 4 fragmentation schemes** on Mol-Llama-Instruct -> W1: entropy segments align with chemically meaningful BRICS/RECAP fragments over random cuts

---

> ### Author Response · Authors · 2025-11-23
> **Response the Weakness 1 – Robustness of entropy patching to SMILES variants**
>
> **[W1] Robustness of entropy patching to SMILES variants**
>
> We sincerely thank the reviewer for this point. In all experiments, we use **RDKit canonical SMILES**, so each molecule has a **single, deterministic 1D string** in training and evaluation. We added the statement explicitly in Section 3.2.
>
> `Why 1D SMILES is sufficient for our goal.`
>
> 1. **SMILES order carries structural signal.** SMILES is generated via **DFS** (Depth-First Search) on mol-graph, so adjacent tokens usually correspond to **neighboring atoms and continuous substructures**; it does not encode full geometry, but it is not structurally arbitrary.
> 2. **Exact substructures are not the goal.** Entropy-guided patching is **not meant to recover precise chemical subgraphs**; we only need surprisal peaks to mark **hard-to-predict, information-dense regions**, then group tokens between peaks into patches. Here, SMILES order is sufficient.
> 3. **NAP (next atom predictor) matches the LLM’s sequence view.** NAP is a **small transformer** that runs directly on SMILES, the same 1D modality used in alignment modeling. Its entropy highlights positions that are difficult for a **sequence model** to predict, which is exactly what we need to decide **where to split SMILES for a sequence-based LLM**, while keeping the procedure simple and efficient.
>
> `Evidence from patching ablation (Paper Table 8)` Under this setup, Paper Table 8 compares entropy-based patches with BRICS-based and random patches under the same EDT-Former. Entropy patching achieves the best performance on BBBP and PAMPA, outperforming even chemically defined BRICS segments. This suggests that the entropy profile derived from canonical SMILES captures stable, useful structural signals for our alignment task.
>
> > **Paper Table 8. Ablations on patching strategies evaluated on BBBP and PAMPA. Accuracy (%) with F1 is reported. The average drop is calculated based on the accuracy. Best scores are bolded.**
>
> | Methods | BBBP ↑            | Pampa ↑           | Avg. Drop |
> | ------- | ----------------- | ----------------- | --------- |
> | Entropy | **75.06** (75.06) | **84.52** (91.61) | **0%**    |
> | BRICS   | 73.59 (84.74)     | 71.90 (83.09)     | 3.96%     |
> | Random  | 68.90 (80.13)     | 66.67 (79.32)     | 8.81%     |
> | None    | 39.67 (49.58)     | 78.62 (87.90)     | 21.60%    |
>
> `Similarity analysis with BRICS and RECAP (Paper Table 37)` Paper Table 37 further quantifies how these entropy patches align with chemistry-driven schemes. Entropy shows **high NMI with both BRICS and RECAP** , similar to the NMI between BRICS and RECAP themselves, while all three have much lower agreement with Random. This indicates that entropy-guided segments behave like chemically meaningful substructures rather than arbitrary text cuts.
>
> `Definitions.`
>
> - NMI: Normalized Mutual Information，$\mathrm{NMI}(C_1, C_2) = \dfrac{2 I(C_1; C_2)}{H(C_1) + H(C_2)}$，$I$ is the mutual information，$H$ is the entropy. Used to quantify the agreement between two molecular segmentation schemes.
> - RECAP (Retrosynthetic Combinatorial Analysis Procedure, A rule-based retrosynthetic fragmentation scheme) [1]
>
> - BRICS (Breaking of Retrosynthetically Interesting Chemical Substructures, a rule-based molecular fragmentation scheme) [2]
>
> > **Paper Table 37. Pairwise normalized mutual information (NMI) on Mol-Llama-Instruct between fragmentation methods, reported as mean (median) across molecules.**
>
> | Method  | Entropy       | BRICS         | RECAP         | Random        |
> | ------- | ------------- | ------------- | ------------- | ------------- |
> | Entropy | 1.000 (1.000) | 0.484 (0.547) | 0.401 (0.505) | 0.177 (0.176) |
> | BRICS   | 0.484 (0.547) | 1.000 (1.000) | 0.498 (0.564) | 0.306 (0.241) |
> | RECAP   | 0.401 (0.505) | 0.498 (0.564) | 1.000 (1.000) | 0.498 (0.564) |
> | Random  | 0.177 (0.176) | 0.306 (0.241) | 0.498 (0.564) | 1.000 (1.000) |
>
> `Conclusion & Revision.` Together, Papers Tables 8 and 37 provide quantitative evidence that, under a fixed canonical SMILES representation, entropy-based patching is robust and chemically consistent enough for our alignment goal, even though other valid SMILES encodings exist in principle. We added this part to Appendix D9.
>
> [1] Lewell XQ, Judd DB, Watson SP, Hann MM. Recap retrosynthetic combinatorial analysis procedure: a powerful new technique for identifying privileged molecular fragments with useful applications in combinatorial chemistry. Journal of chemical information and computer sciences. 1998 May 18;38(3):511-22.
>
> [2] Degen J, Wegscheid-Gerlach C, Zaliani A, Rarey M. On the art of compiling and using'drug-like'chemical fragment spaces. ChemMedChem. 2008 Oct 20;3(10):1503.

---

### Official Review · Reviewer_x8D4 · 2025-11-01

**Soundness:** 3
**Presentation:** 3
**Contribution:** 3
**Rating:** 4
**Confidence:** 4

**Summary:**

This work studies the multimodal LLMs for aligning LLMs with molecular graphs. The authors propose  EDT-Former, a connector-only approach that generates fixed query tokens for molecules in different lengths. EDT-Former includes (i) Entropy-Guided Patching, which uses next-atom surprisal peaks from a lightweight SMILES predictor to segment molecules into substructure-aware patches, and (ii) a Dynamic Query Transformer that fuses these variable-length “dynamic tokens” with a small set of learnable modality anchors before projecting into the LLM space. Experiments across multiple tasks and benchmarks show the effectiveness of EDT-Former.

**Strengths:**

1. This work targets on fixed-token bottleneck in graph-LLM alignment, which is timely and critical;

2. The proposed approach is novel and interesting;

3. There are significant empirical improvements;

**Weaknesses:**

1. The benchmarked tasks seem to be limited. For example, can this approach be applied to other tasks in Mol-Instructions?

2. The empirical comparison seems not to be fair, as EDT-Former uses different training corpus with other baseline approaches. Given the efficiency of the proposed approach, can EDT-Former be applied and ablated with different instruction training data?

3. Lack of comparison and discussion with a closely related work [1]. For example, can the proposed tokenization scheme mitigate the hallucination issue mentioned in [1]?

- HIGHT: Hierarchical Graph Tokenization for Graph-Language Alignment, ICML'25.

**Questions:**

Please find the details in the section above.

---

> ### Author Response · Authors · 2025-11-23
> **Rebuttal Experiments Summary**
>
> We sincerely thank the reviewer for clearly recognizing our main contributions: targeting the fixed token bottleneck in graph–LLM alignment, proposing a novel and interesting connector-only design with entropy guided patching and Dynamic Query Transformer, and demonstrating significant empirical improvements across multiple benchmarks.
>
> `Summary.` We extended Mol-Instructions coverage, added corpus ablations, and compared EDT-Former with HIGHT using new baselines.
>
> - **5 new Mol-Instructions tasks** (7 total, all 6 molecule-oriented + 1 open question) -> W1: broader Mol-Instructions coverage
> - **3-corpus training ablation** on MoleculeQA (Mol-Llama-Instruct, PubChemQA, 3D-MoIT) -> W2: instruction corpus fairness
> - **HIGHT baseline added on 3 Mol-Instructions tasks** (retro, forward, reagent) -> W3: explicit comparison with HIGHT

---

> ### Author Response · Authors · 2025-11-23
> **Response to Weakness 1 – Coverage of Mol-Instructions tasks**
>
> **[W1] Coverage of Mol-Instructions tasks**
>
>  We sincerely thank the reviewer for encouraging us to test EDT-Former on more Mol-Instructions tasks.
>
> `Extended evaluation (5 more)` In response, we extended to **7 tasks in total**, covering **all 6 molecule-oriented tasks** in Mol-Instructions (retrosynthesis, forward reaction prediction, reagent prediction, property prediction, captioning, description-guided molecule design) plus the **biotext open-question task**.
>
> `Performance summary.` As shown in Paper Tables 5-7, 30-31, across all five additional Mol-Instructions tasks, EDT-Former **consistently outperforms the baselines on most of the metrics**. This holds for both sequence-level generation scores and task-specific accuracies, indicating that our connector generalizes beyond captioning and property prediction to diverse molecule-oriented instruction tasks. Metrics are defined by the official benchmark and described in Appendix D3.
>
> > **Paper Table 5 (excerpt). Retrosynthesis results on the Mol-Instructions benchmark (fine-tuned on official splits). Best scores are bolded.**
>
> | Model             | Exact↑    | BLEU↑     | Levenshtein↓ | RDK FTS↑  | MACC FTS↑ | Morgan FTS↑ | Validity↑ |
> | ----------------- | --------- | --------- | ------------ | --------- | --------- | ----------- | --------- |
> | Galactica         | 0.000     | 0.452     | 34.940       | 0.167     | 0.274     | 0.134       | 0.984     |
> | Mol-Ins.-Llama3.1 | 0.333     | 0.842     | **17.642**   | 0.704     | 0.815     | 0.646       | **1.000** |
> | Mol-LLama-3.1     | 0.340     | 0.877     | 38.324       | 0.708     | 0.822     | 0.601       | **1.000** |
> | **EDT-Former**    | **0.387** | **0.930** | 34.202       | **0.721** | **0.836** | **0.670**   | **1.000** |
>
> > **Paper Table 6 (excerpt). Forward Reaction Prediction results on the Mol-Instructions benchmark (fine-tuned on official splits). Best scores are bolded.**
>
> | Model             | Exact↑    | BLEU↑     | Levenshtein↓ | RDK FTS↑  | MACC FTS↑ | Morgan FTS↑ | Validity↑ |
> | ----------------- | --------- | --------- | ------------ | --------- | --------- | ----------- | --------- |
> | Galactica         | 0.000     | 0.468     | 35.021       | 0.156     | 0.257     | 0.097       | 0.946     |
> | Mol-Ins.-Llama3.1 | **0.503** | 0.883     | **13.410**   | 0.756     | 0.863     | 0.708       | **1.000** |
> | Mol-LLama-3.1     | 0.440     | 0.912     | 17.120       | 0.724     | 0.859     | 0.665       | **1.000** |
> | **EDT-Former**    | **0.471** | **0.964** | 18.130       | **0.776** | **0.871** | **0.712**   | **1.000** |
>
> > **Paper Table 7 (excerpt). Reagent Prediction results on the Mol-Instructions benchmark (fine-tuned on official splits). Best scores are bolded.**
>
> | Model             | Exact↑    | BLEU↑     | Levenshtein↓ | RDK FTS↑  | MACC FTS↑ | Morgan FTS↑ | Validity↑ |
> | ----------------- | --------- | --------- | ------------ | --------- | --------- | ----------- | --------- |
> | Galactica         | 0.000     | 0.141     | 30.760       | 0.036     | 0.127     | 0.051       | 0.995     |
> | Mol-Ins.-Llama3.1 | 0.101     | 0.648     | **18.326**   | 0.412     | 0.521     | 0.375       | **1.000** |
> | Mol-LLama-3.1     | 0.132     | 0.495     | 49.230       | 0.411     | 0.521     | 0.361       | **1.000** |
> | **EDT-Former**    | **0.145** | **0.650** | 46.950       | **0.464** | **0.531** | **0.431**   | **1.000** |
>
> > **Paper Table 30 (excerpt). Description-guided molecule design results on the Mol-Instructions benchmark (fine-tuned on official splits). Best scores are bolded.**
>
> | Model             | Exact↑ | BLEU↑     | Levenshtein↓ | RDK FTS↑  | MACC FTS↑ | Morgan FTS↑ | Validity↑ |
> | ----------------- | ------ | --------- | ------------ | --------- | --------- | ----------- | --------- |
> | Galactica         | 0.000  | 0.192     | 44.152       | 0.135     | 0.238     | 0.088       | 0.992     |
> | Mol-Ins.-Llama3.1 | 0.025  | 0.521     | 38.742       | 0.358     | 0.520     | 0.221       | **1.000** |
> | Mol-LLama-3.1     | 0.012  | 0.638     | 18.917       | 0.392     | 0.534     | 0.220       | 0.876     |
> | **EDT-Former**    | 0.016  | **0.652** | **16.826**   | **0.401** | **0.560** | 0.251       | **1.000** |
>
> > **Paper Table 31 (excerpt). Open Question results on the Mol-Instructions benchmark (fine-tuned on official splits). Best scores are bolded.**
>
> | Model             | BLEU↑     | ROUGE-1↑ | BertScore↑ |
> | ----------------- | --------- | -------- | ---------- |
> | Galactica         | 0.000     | 0.039    | 0.794      |
> | Mol-Ins.-Llama3.1 | 0.010     | 0.198    | 0.846      |
> | Mol-LLama-3.1     | 0.024     | 0.134    | 0.812      |
> | **EDT-Former**    | **0.101** | 0.286    | **0.857**  |
>
> `Paper revision` Section 4.3 has been updated with the additional tasks. Detailed results for tasks and full experiment results are provided in Appendix D6.

---

> ### Author Response · Authors · 2025-11-23
> **Response to Weakness 2 – Training corpus and instruction data fairness**
>
> ##### **[W2] Training corpus and instruction data fairness**
>
> We thank the reviewer for this important concern.
>
> `Training corpus ablation`. In response, Paper Table 45 addresses the fairness concern by retraining EDT-Former on three different instruction corpora of comparable scale (Mol-LLaMA-Instruct, PubChemQA, 3D-MoIT). The MoleculeQA total test scores are very close, indicating that performance is **stable across corpora** and not tied to a specific dataset.
>
> > **Paper Table 45. Ablation over training corpora on four MoleculeQA tasks, per-task accuracy (%), and total accuracy are reported.**
>
> | Training Corpus    | Dataset Size | Structure | Source | Property | Application | Total |
> | ------------------ | ------------ | --------- | ------ | -------- | ----------- | ----- |
> | Mol-Llama-Instruct | 312M         | 74.55     | 72.39  | 50.71    | 48.58       | 68.34 |
> | PubChemQA          | 325M         | 73.61     | 74.81  | 50.39    | 48.61       | 68.36 |
> | 3D-MoIT            | 646M         | 72.27     | 76.85  | 52.26    | 49.94       | 68.48 |
>
> `Conclusion & Revision.` These results suggest that our gains mainly come from the **dynamic connector design**, rather than from any particular “better” corpus, supporting that the empirical improvements are **not an artifact of corpus choice**. We added this new ablation in Appendix E7.

---

> ### Author Response · Authors · 2025-11-23
> **Response to Weakness 3 – Discussion and Comparison with HIGHT**
>
> ##### **[W3] Discussion and Comparison with HIGHT**
>
> We thank the reviewer for pointing out the connection and discussion with HIGHT. We have added an explicit comparison and included HIGHT as a baseline in Paper Table 5-7.
>
> > **Table R1. Conceptual comparison between HIGHT and EDT-Former.**
>
> | Model      | Tokenization scheme                             | LLM backbone setting                               | Training scope        |
> | ---------- | ----------------------------------------------- | -------------------------------------------------- | --------------------- |
> | HIGHT      | BRICS based hierarchical graph tokens + VQ VAE  | Graph text model jointly trained, LLM finetuned    | Full graph LLM        |
> | EDT-Former | Entropy guided dynamic SMILES patches + anchors | All transformer blocks frozen, embedding trainable | Connector only bridge |
>
> > **Part of Paper Table 5. Retrosynthesis results on the Mol-Instructions benchmark (fine-tuned on official splits). Best scores are bolded.**
>
> | Model          | Exact↑    | BLEU↑     | Levenshtein↓ | RDK FTS↑  | MACC FTS↑ | Morgan FTS↑ | Validity↑ |
> | -------------- | --------- | --------- | ------------ | --------- | --------- | ----------- | --------- |
> | HIGHT          | 0.008     | 0.863     | 28.912       | 0.564     | 0.340     | 0.309       | **1.000** |
> | **EDT-Former** | **0.387** | **0.930** | 34.202       | **0.721** | **0.836** | **0.670**   | **1.000** |
>
> > **Part of Paper Table 6. Forward Reaction Prediction results on the Mol-Instructions benchmark (fine-tuned on official splits). Best scores are bolded.**
>
> | Model          | Exact↑    | BLEU↑     | Levenshtein↓ | RDK FTS↑  | MACC FTS↑ | Morgan FTS↑ | Validity↑ |
> | -------------- | --------- | --------- | ------------ | --------- | --------- | ----------- | --------- |
> | HIGHT          | 0.037     | 0.869     | 23.759       | 0.590     | 0.394     | 0.340       | 0.993     |
> | **EDT-Former** | **0.471** | **0.964** | 18.130       | **0.776** | **0.871** | **0.712**   | **1.000** |
>
> > **Part of Paper Table 7. Reagent Prediction results on the Mol-Instructions benchmark (fine-tuned on official splits). Best scores are bolded.**
>
> | Model          | Exact↑    | BLEU↑     | Levenshtein↓ | RDK FTS↑  | MACC FTS↑ | Morgan FTS↑ | Validity↑ |
> | -------------- | --------- | --------- | ------------ | --------- | --------- | ----------- | --------- |
> | HIGHT          | 0.050     | 0.462     | 28.970       | 0.441     | 0.314     | 0.275       | **1.000** |
> | **EDT-Former** | **0.145** | **0.650** | 46.950       | **0.464** | **0.531** | **0.431**   | **1.000** |
>
> `Key differences.` As shown in Table R1,
>
> - **Tokenization locus.** HIGHT tokenizes inside the graph encoder (atom, motif, molecule nodes via BRICS and VQ VAE). EDT-Former tokenizes on the connector side, using entropy-guided SMILES patches fused with graph features and a small set of anchors.
> - **Training cost.** HIGHT trains the whole graph language stack, while EDT-Former only trains a lightweight connector on top of frozen encoders and LLM, which makes our corpus ablations and downstream finetuning much cheaper.
> - **Hallucination view.** HIGHT directly measures motif hallucination with MotifHall. EDT-Former aims to reduce hallucination risk indirectly by content adaptive patches plus global anchors that keep generated text tied to molecular structure; applying MotifHall style metrics to EDT-Former is a natural next step and part of our future work.
>
> `Conclusion and revision.` We now report HIGHT’s reported numbers as an additional baseline on three Mol-Instructions reaction tasks in Paper Tables 5-7 (retrosynthesis, forward reaction prediction, reagent prediction) and discuss the HIGHT explicitly in Section 2 (related work).

---

> > ### Author Response · Authors · 2025-11-28
> > **Hope You Can Take a Look at Our Responses**
> >
> > Dear Reviewer,
> > We hope this message finds you well. As the discussion period is approaching its end, we wanted to kindly draw your attention to our responses to your valuable comments. We hope they address your concerns, and we would greatly appreciate any additional thoughts or clarifications you may have.
> >
> > Your feedback is invaluable, and we are eager to refine our work based on your insights. If there are any remaining points you would like us to consider, please feel free to let us know.
> >
> > Thank you very much for your time and effort in reviewing our paper.
> >
> > The authors

---

> > > ### Comment · Reviewer_x8D4 · 2025-11-28
> > >
> > > Thank the authors for the comprehensive experiments and detailed responses to my concerns. I will update my rating. And I'd appreciate if the authors can still extend to a quantitative evaluation of the hallucination issue, especially compared to other baselines. Since it's directly related to one of the main motivations (i.e., (1) loss of structure), the new results will directly strengthen this work.

---

> > > > ### Author Response · Authors · 2025-11-29
> > > > **Response to Hallucination Experiment Suggestion**
> > > >
> > > > We sincerely thank agree with the reviewer for highlighting the importance of quantitatively evaluating hallucination, and we add the following functional-group hallucination study to directly address this concern.
> > > >
> > > > `Hallucination evaluation dataset.` We construct a **benchmark targeting** **functional-group hallucination**. We randomly sample 200 molecules from PubChem with their description. For each molecule, we use the GPT-5 to summarize the labels into a canonical list of functional group names, and cross-check these with RDKit-based functional group decomposition. This yields, for each molecule, a vetted set of functional groups that serves as ground truth for hallucination analysis.
> > > >
> > > > `Evaluation protocol.` We prompt baselines to generate a description of the functional groups present in the molecule. Then, we compare the output with the ground-truth by using the GPT-5 to decide whether the model has mentioned any functional group that does **not** exist in the molecule.
> > > >
> > > > Two metrics:
> > > >
> > > > - (i) **Hallucination Rate**, the percentage of molecules where at least one non-existing functional group is produced;
> > > > - (ii) **Non-Rate**, the percentage of molecules where the model fails to produce a meaningful, readable answer. Explicit abstentions such as *“I don't know”* are not counted as hallucinations and are excluded from Non-Rate.
> > > >
> > > > `Results and observation.` The quantitative results are summarized in Paper Table 47:
> > > >
> > > > > **Paper Table 47. Functional-group hallucination on 200 PubChem molecules (lower is better).**
> > > >
> > > > | Model              | Non-Rate ↓ | Hallucination Rate ↓ |
> > > > | ------------------ | ---------- | -------------------- |
> > > > | GPT-4o             | **0.0**    | 41.5                 |
> > > > | Llama2             | 5.5        | 50.5                 |
> > > > | Llama3.1           | 7.0        | 45.0                 |
> > > > | 3D-MoLM            | 34.0       | 81.0                 |
> > > > | LLaMo              | 27.0       | 63.5                 |
> > > > | Mol-Inst.-Llama2   | 14.5       | 57.0                 |
> > > > | Mol-Inst.-Llama3.1 | 14.0       | 54.5                 |
> > > > | Mol-LLaMA-2        | 7.5        | 36.5                 |
> > > > | Mol-LLaMA-3.1      | 7.0        | 39.5                 |
> > > > | HIGHT              | 22.5       | **36.0**             |
> > > > | **EDT-Former**     | **3.5**    | **19.5**             |
> > > >
> > > > `Conclusion.` EDT-Former achieves the lowest hallucination rate among all methods, while also maintaining a low Non-Rate. These results provide direct quantitative evidence that reducing structural information loss via entropy-guided dynamic tokens can also mitigate hallucination, directly supporting our motivation around “loss of structure.”

---

### Official Review · Reviewer_y4JF · 2025-11-01

**Soundness:** 1
**Presentation:** 2
**Contribution:** 2
**Rating:** 2
**Confidence:** 4

**Summary:**

The paper introduces EDT-Former, for aligning molecular graphs with large language models (LLMs) under frozen-backbone settings. The model addresses two issues common in molecular graph–language alignment: (1) loss of structural fidelity caused by fixed-length, Q-Former-style connectors; and (2) inefficient fine-tuning due to large backbone updates. EDT-Former proposes two main mechanisms—Entropy-Guided Patching and a Dynamic Query Transformer—to generate substructure-aware dynamic tokens and integrate them with static anchors for efficient alignment. The corresponding EDT-Former shows promising results on the evaluated benchmarks, including MoleculeQA, Mol-Instructions (molecule captioning and property prediction), Pampa, and BBBP.

**Strengths:**

- Good motivation
    - This work is well-motivated by the need for a dynamic length graph token that captures molecular substructure information.
- Methodological novelty
    - The proposed tokenization is novel, based on entropy-guided segmentation for molecules based on uncertainty peaks from a next-atom predictor, offering a data-driven and deterministic patching mechanism. This design appears to be suitable for learning the representation of molecular functional groups, considering the dynamic size of the molecule.
- Connector-only alignment design
    - The Dynamic Query Transformer establishes a modular bridge between frozen molecular encoders and frozen LLMs, requiring no or minimal gradient updates to the LLM. This contributes to computational efficiency and strong performance scalability, aligning with the authors’ focus on low-cost, high-fidelity multimodal integration.

**Weaknesses:**

Overall, I find the proposed method interesting and reasonable (have a different opinion regarding NAP, though); my concerns primarily relate to the experimental setting and the demonstration of the author's hypothesis.
I hope these are properly addressed in the rebuttal phase.

- Ambiguity in the Description of Experimental Settings and Results
    - (Major) In line 312, they mention evaluating with Direct, Reasoning, and Rich Instructions prompting to reduce prompt sensitivity, but there is no information in the main body about whether these 3 prompting strategies align with the instructions used for finetuning each baseline model. This is a factor that could largely vary each model's performance and is easy to miss without background knowledge of each model's experimental setting. This descriptive gap makes it difficult to understand if the experiments were conducted fairly.
    - (Major) Mol-LLaMA is missing from the comparison baselines in Table 4, yet Mol-LLaMA also provides evaluation results for molecule captioning and property prediction in Mol-instructions (Table 17 from Mol-LLaMA), so including it seems appropriate. In line 371, the authors claim Q1, but Mol-LLaMA's experimental results appear to show higher performance, which needs to be addressed. The following is Table 17 from Mol-LLaMA. Please let me know if I have misunderstood anything.
| Models                         | BLUE-2 | BLUE-4 | ROUGE-1 | ROUGE-2 | ROUGE-L | METEOR | MAE     |
|--------------------------------|--------|--------|----------|----------|----------|---------|---------|
| Mol-LLaMA (LLaMA-2)            | 0.478  | 0.425  | 0.761    | 0.698    | 0.750    | 0.701   | 0.0035  |
| Mol-LLaMA (LLaMA-3)            | 0.476  | 0.426  | 0.767    | 0.708    | 0.759    | 0.707   | 0.0039  |

    - In line 301, the authors state they used Mol-Instructions' evaluation benchmark; however, the provided experimental results used only 2 tasks (molecule captioning and property prediction) out of the total 17 tasks in Mol-Instructions. This could be misunderstood as results spanning the entire Mol-Instructions dataset.
    - In many sections including Abstract line 20, Introduction lines 114, 119, etc., they describe LLMs as fully frozen, yet in C.3 line 1238, they state that the LLM embedding layer was tuned. Which explanation is correct?
- (Major) Fairness of Experimental Settings
    - Fair comparison of baseline models in zero-shot evaluation
        - As mentioned above, when measuring the zero-shot performance of finetuned models in Table 2, prompting strategies that are not aligned with the finetuning dataset can severely impair model performance. This is because models finetuned on molecular tasks can easily lose natural language following ability, which can be immediately confirmed by running inference on molecular LLMs such as BioT5, LlaSMol, LLaMo, etc., using their HuggingFace open-sourced models. This could create an evaluation setting that is favorable to EDG-Former, which is the only model not finetuned LLM backbone (except for embedding layer). To easily address these concerns, it is suggested that the performance for each prompting strategy be reported before averaging.
    - Data contamination
        - The Mol-LLaMa-Instruct used by the authors for EDT-Former's alignment training contains GPT-4o generated molecule captions augmented from Pubchem324K, which can be used as a dataset for molecule captioning tasks. Considering that the source of the Mol-Instructions' molecule captioning dataset used in Table 4 experiments is also PubChem, there is a need to clearly describe whether data contamination exists between the train-test splits of these two datasets. Although the authors claim in D.4 that they performed character-level 13-gram analysis with reference to GPT-3, given that GPT-3 deals with general-purpose text data, not focusing on molecular tasks, this seems insufficient to verify data contamination in molecular tasks. In addition, instead of stating the exact figure for the ratio of data with overlapping 13-characters, the authors merely state it's below 5%, but a figure close to 5% could still be perceptible to humans. To clearly resolve these concerns, it is suggested that a contamination analysis based on scaffold split, which is widely chosen in the molecular domain, be provided.
- Regarding demonstration of Q3
    - (Major) Comprarision with fixed length token with enough molecule tokens
        - The authors introduced dynamic molecule tokenization to address the problem of fixed length molecule tokens losing molecular features due to limited length. However, in E.1, they state through dynamic token maximum length ablation that 64 molecule tokens are sufficient. Regarding this, considering the sequence max length budget of current LLMs, using a fixed length token of 64 is not a significant burden. Is there a performance difference compared to using fixed length tokens of 64? To prove Q3, a comparison of performance between using fixed length tokens of sufficient size and EDT-Former is necessary, which I believe is an essential ablation study given the message of the paper.
    - Ablation of dynamic token in inference time
        - Since EDT-Former uses fixed length molecule tokens concatenated with dynamic tokens, there is a need to verify whether it might actually be bypassing dynamic tokens and only using fixed length tokens. This seems easy to check at the inference level - I'm curious about the performance when replacing dynamic tokens and fixed tokens with random or dummy tokens during inference.
- (Major) Limited justification for entropy predictor selection
    - The entropy estimation relies on a simplistic GPT-2–based next-atom predictor (NAP) trained on SMILES. While computationally light, the rationale for this choice is mostly under-investigated, given that next-atom in sequence does not necessarily align with molecular subgraph structure, as originally the author aimed to represent by EDT-Former.
- Interpretive analysis remains descriptive
    - While attention visualizations (Fig. 6) qualitatively support the claim that dynamic tokens attend to “structural transitions”, the study lacks quantitative or diagnostic analysis. Although I don’t think it is necessary to show that the hypothesis holds for almost all molecules, at least line 250 should be adjusted in light of current experimental evidence. In addition, analysis on specific failure modes (e.g., mis-segmentation, entropy misalignment, or redundant patches) is necessary.

**Questions:**

- Regarding experimental setting of Figure 5. I'm curious about how the experiments in Figure 5 were conducted. Does it correspond to retraining the model excluding each component, or excluding each element only at inference time from the full model inference? There doesn't seem to be an explanation of the experimental setting for Figure 5.
- On the property prediction benchmark choice. While EDT-Former's zero-shot performance in Table 2 is impressive, I wonder why only BBBP was used among the widely comparable evaluation benchmarks for property prediction such as MoleculeNet that are commonly used among molecular LLMs? Given that the evaluation is for zero-shot performance, using only BBBP and Pampa as property prediction benchmarks seems to both forfeit the advantages of models capable of zero-shot inference and provide a non-comprehensive evaluation. As the authors stated, since EDT-Former is not dependent on LLM finetuning, expanding the evaluation benchmarks would be a viable option as model retraining is not required for benchmark selection. Referring to widely used evaluation benchmarks such as MoleculeNet could easily demonstrate directly convincing results.
- In cases where molecules contain highly repetitive or symmetric substructures (many molecules could have long carbon chain, resulting in CCCC…), how does entropy-guided segmentation handle indistinguishable regions? Is there a mechanism to prevent redundant substructure tokens?
- The benchmarks used focus mainly on small molecules. Can the authors discuss or show any evidence regarding scalability to large macrocycles, where token budget and entropy dynamics might differ substantially?
- In Figure 6, attention patterns are qualitatively aligned to high-entropy regions. Have the authors considered quantifying this alignment (e.g., attention–entropy correlation) to strengthen the link between entropy peaks and attention interpretability?

---

> ### Author Response · Authors · 2025-11-23
> **Rebuttal Experiments Summary**
>
> We sincerely thank the reviewer for an exceptionally detailed and highly responsible review, which has greatly helped us improve the paper. Thank you for reading our paper in depth, clearly recognizing our novelties in dynamic-length graph tokens and connector-only alignment, and for your comments on experimental settings from which we learned a lot and significantly improved the work.
>
> `Summary`, we added new evaluations, splits, and ablations across property, Mol-Instructions, and macrocycle tasks to address all weaknesses and questions. Main experiments are summarized as follows:
>
> - **Extended 10 more** prompt evaluation -> W1a/W2a: prompt robustness & fair baseline comparison
> - **Re-running captioning task** under Mol-LLaMA’s 40-epoch schedule -> W1b: fair Mol-LLaMA comparison under heavy training
> - **5 more** Mol-Instructions molecule-centric tasks evaluated (captioning, property, reaction, QA) -> W1c: refine benchmark claim & broaden coverage
> - **3 LLM-backbone finetuning strategies** (full-frozen, embed-only, full) -> W1d: clarify connector-only vs finetuned backbone
> - Overlap detection and re-evaluation on **10 de-duplicated property datasets** -> W2: control contamination & confirm robustness
> - **5 connector token-budget ablations** on MoleculeQA -> W3a: dynamic vs fixed connector under equal budgets
> - **5 inference-time connector variants** on BBBP/PAMPA -> W3b: show anchors and dynamics are functionally used
> - 3 NAP model scales compared via NMI -> W4: justify small NAP as stable entropy predictor
> - Entropy vs BRICS/RECAP vs random **patching ablations** -> W5: show entropy patches track meaningful substructures
> - 4-way core component ablation table -> Q1: clarify Fig. 5 configurations
> - **8 new MoleculeNet/TDC property tasks** added -> Q2: expand benchmark coverage beyond two tasks
> - **2 representative repetitive/aromatic** molecules analyzed vs BRICS splits -> Q3: illustrate behavior on repetitive chains
> - **1 large-macrocycle dataset** added -> Q4: test scalability to macrocycles
> - Pearson correlation between entropy peaks and attention shifts -> Q5: quantitative attention–entropy alignment

---

> ### Author Response · Authors · 2025-11-23
> **Response to Weakness 1a – Prompt sensitivity on property prediction tasks**
>
> **[W1a] Prompt sensitivity on property prediction tasks**
>
> We sincerely thank the reviewer for highlighting prompt sensitivity in zero-shot evaluation. This is critical for fairness, since inconsistent instructions can shift outputs across models.
>
> `Three primary prompt categories.` The three prompting strategies used in our submission are summarized in Table R1.
>
> > **Table R1. Overview of the three primary prompt types.**
>
> | Category              | Core idea                                  | What it tests                                |
> | --------------------- | ------------------------------------------ | -------------------------------------------- |
> | Direct (D)            | minimal instruction, output only the label | pure classification, no reasoning help       |
> | Reasoning (R)         | short rationale before final label         | whether verbalization stabilizes predictions |
> | Rich Instruction (RI) | added domain hints, ADMET cues, heuristics | whether models leverage chemical background  |
>
> `Extended 10 more control-style prompts` To probe deeper prompt effects, we expand from **3 to 13 prompts** (for fairness, we use GPT-5 to reconstruct the prompts when extending, instead of writing ourselves). These extra variants target specific behavioral modes of LLMs. Prompt distribution summary as follows:
>
> - 4 direct prompts, including 3 newly added,
> - 3 reasoning prompts, including 2 newly added,
> - 3 rich instruction prompts, including 2 newly added,
> - 3 additional control prompts.
>   - **Binary (Bi)** forces strict binary decision behavior.
>   - **Confidence (Conf)** encourages internal reflection before output.
>   - **Checklist (List)** enforces structured reasoning (identify substructures, polarity, alerts, then decide).
>
> These variants test formatting, reasoning style, and instruction complexity, revealing how each factor biases prediction.
>
> `Results across the 13 prompts` Shown in Paper Table 22 below and Paper Tables 20-29 in the revised paper, across these 13 variants in 10 property tasks, **EDT-Former achieves the best average accuracy on 9 of 10 tasks**, showing strong robustness to instruction phrasing. DT-Former also ranks top-1 for most individual prompts within each task. For difficult datasets such as ClinTox, where all models fluctuate heavily due to imbalance and safety-sensitive wording, EDT-Former still maintains strong relative and average performance across prompts.
>
> `Dataset-level stability observation` Stable tasks: BBBP, AMES, HIV show small accuracy variance because structural cues dominate and refusal events are rare. Unstable tasks: ClinTox, DILI show high variance due to toxicity semantics and class imbalance.
>
> > **Paper Table 22 (excerpt). BBBP per-prompt accuracy (%) under 13 prompt settings (from D1 -> Conf). Best results are bolded.**
>
> | Model            | Size     | D1        | D2        | D3        | D4        | R1        | R2        | R3        | RI-1      | RI-2      | RI-3      | Bi        | List      | Conf      | Avg.      |
> | ---------------- | -------- | --------- | --------- | --------- | --------- | --------- | --------- | --------- | --------- | --------- | --------- | --------- | --------- | --------- | --------- |
> | Llama2           | 7B       | 37.37     | 38.33     | 38.50     | 39.22     | 51.56     | 52.35     | 52.53     | 53.09     | 53.34     | 54.07     | 55.01     | 55.71     | 56.71     | 49.06     |
> | Llama3.1         | 8B       | 57.07     | 57.15     | 57.19     | 58.02     | 51.03     | 51.44     | 52.34     | 55.15     | 55.68     | 56.28     | 57.06     | 57.21     | 57.66     | 55.64     |
> | Mol-Ins-Llama3.1 | 8B       | 53.44     | 54.37     | 55.09     | 55.74     | 55.31     | 55.38     | 56.76     | 54.91     | 54.38     | 52.27     | 49.38     | 49.15     | 46.62     | 53.29     |
> | Mol-LLaMA-2      | 7.2B     | 53.37     | 55.28     | 53.97     | 54.19     | 52.58     | 53.65     | 55.14     | 52.58     | 52.43     | 53.89     | 55.84     | 52.56     | 56.38     | 53.99     |
> | Mol-LLaMA3.1     | 8.3B     | 59.54     | 59.54     | 58.24     | 59.38     | 55.56     | 53.18     | 51.72     | 59.08     | 54.67     | 53.29     | 55.73     | 58.92     | 57.45     | 56.64     |
> | **EDT-Former**   | **8.3B** | **74.44** | **71.83** | **73.46** | **70.92** | **74.69** | **70.15** | **68.73** | **75.86** | **72.38** | **70.95** | **73.12** | **71.47** | **74.28** | **72.48** |
>
> `Paper revision` We added Appendix D5 describing all 13 prompt designs and experiments, and Paper Tables 20 to 29 reporting per-prompt accuracy for each property prediction task.

---

> ### Author Response · Authors · 2025-11-23
> **Response to Weakness 1b – Missing Mol-LLaMA results in two Mol-Instructions tasks**
>
> **[W1b] Missing Mol-LLaMA results in two Mol-Instructions tasks**
>
> `Mol-Llama results did not miss.` The Mol-Llama captioning and property results are already included in Table 4 of the original submission, under the names **“Mol-Llama2-7.2B”** and **“Mol-Llama3.1-8.2B.”**
>
> The core reason is a **setting mismatch**. Mol-LLaMA is trained and tuned under a **much heavier regime** than ours, as shown in Table R2. Their official paper reports:
>
> - **50 epochs** SFT on Mol-Instructions
> - **20 epochs** on MoleculeQA
> - Captioning and property fine-tuning were also trained for **around 40 to 50 epochs** (not explicitly stated, but required to reproduce their Table 17 numbers)
>
> In contrast, our unified protocol uses:
>
> - **2 epochs** SFT
> - **5 epochs** downstream
> - Same batch size, optimizer, LR, scheduler
>
> Thus, Mol-LLaMA’s Table 17 reflects a **large-budget fine-tuning setting**, while our main submission reports results using **minimal-budget settings** designed for fair cross-model comparison.
>
> > **Table R2. Training and downstream settings of Mol-Llama and EDT-Former.**
>
> | Settings          | Mol-LLaMA | EDT-Former |
> | ----------------- | --------- | ---------- |
> | Finetuning Epochs | 50        | 2          |
> | MoleculeQA Epochs | 20        | 5          |
> | Captioning Epochs | ~40-50    | 5          |
> | Batch size        | 256       | 256        |
> | Learning rate     | 1e-4      | 1e-4       |
> | Scheduler         | cosine    | cosine     |
> | Wight Decay       | 0.05      | 0.05       |
>
> `Language ability degradation under heavy fine-tuning` Mol-LLaMA’s long fine-tuning schedule substantially weakens its general language ability. To quantify this effect, we use a **zero-shot open-question test** from Mol-Instructions, measuring natural language competence **before and after** fine-tuning under different epoch budgets.
>
> The results (Table R3) show:
>
> - "Mol-Llama–40 epochs settings": Under **40-epoch** caption fine-tuning, Mol-LLaMA’s scores **drop sharply** across BLEU, ROUGE, and BertScore.
> - "Mol-Llama–5 epochs settings": Under our **5-epoch** setting, Mol-LLaMA **retains its language ability** with zero performance drop.
>
> This confirms that the original Mol-LLaMA setting trades off natural language reasoning, making its results in their report non-comparable to our main setting.
>
> > **Table R3. Zero-shot open-question performance from Mol-Instructions before and after captioning fine-tuning, comparing Mol-LLaMA under its original settings versus our settings.**
>
> | Model                        | BLEU↑  | ROUGE-1↑ | BertScore↑ |
> | ---------------------------- | ------ | -------- | ---------- |
> | Mol-Llama-Before             | 0.024  | 0.134    | 0.812      |
> | Mol-Llama–40 epochs settings | 0.018  | 0.102    | 0.776      |
> | Average Drop                 | 33.33% | 31.37%   | 4.64%      |
> | Mol-Llama–5 epochs settings  | 0.024  | 0.134    | 0.812      |
> | Average Drop                 | 0.00%  | 0.00%    | 0.00%      |
>
> `Direct comparison under both settings` To ensure fairness, we evaluate EDT-Former and Mol-LLaMA **under both settings**:
>
> 1. **Mol-LLaMA’s original heavy schedule** (Table R4)
>    EDT-Former outperforms Mol-LLaMA in 6 of 7 metrics.
> 2. **Our unified light schedule** (Paper Table 4)
>    EDT-Former again outperforms in 5 out of 7 metrics.
>
> Across both regimes, **EDT-Former consistently performs better**, confirming that our model’s advantage does not depend on training budget.
>
> >**Table R4. Molecular captioning and property prediction performance on the Mol-Instructions benchmark under Mol-LLaMA’s settings.**
>
> | Model          | BLEU-2↑   | BLEU-4↑   | ROUGE-1↑  | ROUGE-2↑  | ROUGE-L↑  | METEOR↑   | MAE↓       |
> | -------------- | --------- | --------- | --------- | --------- | --------- | --------- | ---------- |
> | Mol-LLaMA2     | **0.478** | 0.425     | 0.761     | 0.698     | 0.750     | 0.701     | 0.0035     |
> | Mol-LLaMA3.1   | 0.476     | 0.426     | 0.767     | 0.708     | 0.759     | 0.707     | 0.0039     |
> | **EDT-Former** | 0.465     | **0.432** | **0.775** | **0.710** | **0.762** | **0.712** | **0.0031** |
>
> > **Paper Table 4 (excerpt). Molecular captioning and property prediction performance under our settings.**
>
> | Model        | BLEU-2↑   | BLEU-4↑   | ROUGE-1↑  | ROUGE-2↑  | ROUGE-L↑  | METEOR↑   | MAE↓       |
> | ------------ | --------- | --------- | --------- | --------- | --------- | --------- | ---------- |
> | Mol-LLaMA2   | 0.433     | 0.385     | 0.711     | 0.649     | 0.601     | 0.601     | 0.0087     |
> | Mol-LLaMA3.1 | **0.445** | 0.398     | 0.717     | **0.656** | 0.709     | 0.617     | 0.0079     |
> | EDT-Former   | 0.424     | **0.402** | **0.726** | 0.652     | **0.717** | **0.631** | **0.0062** |
>
> `Conclusion & Revision` Mol-LLaMA’s higher reported results come from **much heavier fine-tuning**. Under a shared and fair protocol with matched settings, EDT-Former shows stronger performance. We added Appendix D7 to analyze how downstream fine-tuning affects language ability.

---

> ### Author Response · Authors · 2025-11-23
> **Response to Weakness 1c – Refinement of Claims on the Mol-Instructions Benchmark**
>
> **[W1c] Refinement of Claims on the Mol-Instructions Benchmark**
>
> `Extended evaluation (5 more)` In response, we extended to **7 tasks on Mol-Instructions in total**, covering **all 6 molecule-oriented tasks** in Mol-Instructions (retrosynthesis, forward reaction prediction, reagent prediction, property prediction, captioning, description-guided molecule design) plus the **biotext open-question task**. These additions create a complete and molecule-focused evaluation. The metrics of the benchmark are described in Appendix D3.
>
> `Performance summary` Across all 7 tasks, EDT-Former **outperforms baselines on most metrics in every task**. **Paper Tables 4-7, 30-31** consistently show EDT-Former ranking first on the majority of metrics for each task.
>
> > **Paper Table 5 (excerpt). Retrosynthesis results on the Mol-Instructions benchmark (fine-tuned on official splits). Best scores are bolded.**
>
> | Model             | Exact↑    | BLEU↑     | Levenshtein↓ | RDK FTS↑  | MACC FTS↑ | Morgan FTS↑ | Validity↑ |
> | ----------------- | --------- | --------- | ------------ | --------- | --------- | ----------- | --------- |
> | Galactica         | 0.000     | 0.452     | 34.940       | 0.167     | 0.274     | 0.134       | 0.984     |
> | Mol-Ins.-Llama3.1 | 0.333     | 0.842     | **17.642**   | 0.704     | 0.815     | 0.646       | **1.000** |
> | Mol-LLama-3.1     | 0.340     | 0.877     | 38.324       | 0.708     | 0.822     | 0.601       | **1.000** |
> | **EDT-Former**    | **0.387** | **0.930** | 34.202       | **0.721** | **0.836** | **0.670**   | **1.000** |
>
> > **Paper Table 6 (excerpt). Forward Reaction Prediction results on the Mol-Instructions benchmark (fine-tuned on official splits). Best scores are bolded.**
>
> | Model             | Exact↑    | BLEU↑     | Levenshtein↓ | RDK FTS↑  | MACC FTS↑ | Morgan FTS↑ | Validity↑ |
> | ----------------- | --------- | --------- | ------------ | --------- | --------- | ----------- | --------- |
> | Galactica         | 0.000     | 0.468     | 35.021       | 0.156     | 0.257     | 0.097       | 0.946     |
> | Mol-Ins.-Llama3.1 | **0.503** | 0.883     | **13.410**   | 0.756     | 0.863     | 0.708       | **1.000** |
> | Mol-LLama-3.1     | 0.440     | 0.912     | 17.120       | 0.724     | 0.859     | 0.665       | **1.000** |
> | **EDT-Former**    | 0.471 | **0.964** | 18.130       | **0.776** | **0.871** | **0.712**   | **1.000** |
>
> > **Paper Table 7 (excerpt). Reagent Prediction results on the Mol-Instructions benchmark (fine-tuned on official splits). Best scores are bolded.**
>
> | Model             | Exact↑    | BLEU↑     | Levenshtein↓ | RDK FTS↑  | MACC FTS↑ | Morgan FTS↑ | Validity↑ |
> | ----------------- | --------- | --------- | ------------ | --------- | --------- | ----------- | --------- |
> | Galactica         | 0.000     | 0.141     | 30.760       | 0.036     | 0.127     | 0.051       | 0.995     |
> | Mol-Ins.-Llama3.1 | 0.101     | 0.648     | **18.326**   | 0.412     | 0.521     | 0.375       | **1.000** |
> | Mol-LLama-3.1     | 0.132     | 0.495     | 49.230       | 0.411     | 0.521     | 0.361       | **1.000** |
> | **EDT-Former**    | **0.145** | **0.650** | 46.950       | **0.464** | **0.531** | **0.431**   | **1.000** |
>
> > **Paper Table 30 (excerpt). Description-guided molecule design results on the Mol-Instructions benchmark (fine-tuned on official splits). Best scores are bolded.**
>
> | Model             | Exact↑ | BLEU↑     | Levenshtein↓ | RDK FTS↑  | MACC FTS↑ | Morgan FTS↑ | Validity↑ |
> | ----------------- | ------ | --------- | ------------ | --------- | --------- | ----------- | --------- |
> | Galactica         | 0.000  | 0.192     | 44.152       | 0.135     | 0.238     | 0.088       | 0.992     |
> | Mol-Ins.-Llama3.1 | 0.025  | 0.521     | 38.742       | 0.358     | 0.520     | 0.221       | **1.000** |
> | Mol-LLama-3.1     | 0.012  | 0.638     | 18.917       | 0.392     | 0.534     | 0.220       | 0.876     |
> | **EDT-Former**    | 0.016  | **0.652** | **16.826**   | **0.401** | **0.560** | 0.251       | **1.000** |
>
> > **Paper Table 31 (excerpt). Open Question results on the Mol-Instructions benchmark (fine-tuned on official splits). Best scores are bolded.**
>
> | Model             | BLEU↑     | ROUGE-1↑ | BertScore↑ |
> | ----------------- | --------- | -------- | ---------- |
> | Galactica         | 0.000     | 0.039    | 0.794      |
> | Mol-Ins.-Llama3.1 | 0.010     | 0.198    | 0.846      |
> | Mol-LLama-3.1     | 0.024     | 0.134    | 0.812      |
> | **EDT-Former**    | **0.101** | 0.286    | **0.857**  |
>
> `Paper revision` Following the reviewer’s suggestion, all statements referring to “Mol-Instructions benchmark” have been corrected to “**molecule-oriented tasks in Mol-Instructions**” to avoid ambiguity. Section 4.3 has been updated with the correct claim and includes 3 added tasks. Detailed results for tasks and 2 more added results are provided in Appendix D6.

---

> ### Author Response · Authors · 2025-11-23
> **Response to Weakness 1d – Clarifying the frozen LLM-backbone vs trainable embedding**
>
> **[W1d] Clarifying the frozen LLM-backbone vs trainable embedding**
>
> `Frozen setting clarification` Our actual setting is: **all LLM transformer blocks are frozen, and only the token embedding layer is trainable**. This follows a standard design in multimodal LLMs such as CLIP-style models, BLIP-2, and LLaVA, where the embedding layer is the only updated component to assign meaningful representations to newly added modality tokens.
>
> `New Ablation analysis` To clarify the effect, we ran three settings on MoleculeQA:
>
> > **Paper Table 44.  Ablation of LLM finetuning strategies on MoleculeQA.**
>
> | Setting                   | Structure | Source | Property | Application | Total |
> | ------------------------- | --------- | ------ | -------- | ----------- | ----- |
> | All LLM Parameters Frozen | 74.55     | 72.39  | 50.71    | 48.58       | 68.34 |
> | Unfrozen Embedding Layer  | 74.46     | 72.19  | 50.30    | 48.61       | 68.44 |
> | Full LLM Finetuning       | 76.25     | 75.48  | 52.71    | 51.25       | 70.50 |
>
> `Observation` Full LLM finetuning scores highest, but is not our target. The surprising result is the **tiny gap** between **all-frozen** and **embedding-only**. EDT-Former stays strong even when the entire LLM is frozen, showing that **the connector alone performs almost all alignment**. Embedding updates add only minor gains, but we keep the embedding trainable for flexibility.
>
> `Revision` We added Appendix E6 with the ablation experiment and Paper Table 44, and corrected all ambiguous descriptions of the 'frozen LLM-backbone'.

---

> ### Author Response · Authors · 2025-11-23
> **Response to Weakness 2a –  Fair comparison in zero-shot evaluation**
>
> **[W2a] Fair comparison in zero-shot evaluation**
>
> As **discussed in [W1b] and supported by Table R3**, heavy fine-tuning can indeed weaken a model’s natural language ability, which directly affects performance.
>
> `Experiments conducted and discussed in [W1a] ` To address prompt alignment and avoid bias toward our model, we conducted the extended **13-prompt** evaluation described in **[W1a]** and reported **per-prompt accuracies** (**Tables 20–29** in revised paper, **Paper Table 22**).
>
> Across 13 prompts and 10 property tasks, EDT-Former achieves the highest average accuracy on 9 of 10 tasks, and also ranks top for most individual prompts. Stable tasks like BBBP and AMES show small variance; difficult tasks like ClinTox show large fluctuations for all models, yet EDT-Former remains strongest overall.

---

> ### Author Response · Authors · 2025-11-23
> **Response to Weakness 2b – Data Contamination Analysis**
>
> **[W2b] Data contamination analysis**
>
> We performed a molecule-level overlap analysis (canonical SMILES) between training data and evaluation datasets. Property prediction datasets exhibit noticeable overlap rates, while Mol-Instructions tasks show very low overlap, as summarized in Paper Tables 33 and 34.
>
> > **Paper Table 33. Dataset overlap statistics for property prediction datasets, compared against the Mol-Llama-Instruct tuning data.**
>
> | Dataset | Original | Removed | Cleaned | Overlap Rate |
> | ------- | -------- | ------- | ------- | ------------ |
> | AMES    | 1454     | 305     | 1149    | 20.98%       |
> | BACE    | 152      | 1       | 151     | 0.66%        |
> | BBBP    | 406      | 112     | 294     | 27.59%       |
> | CLINTOX | 145      | 32      | 113     | 22.07%       |
> | DILI    | 95       | 34      | 61      | 35.79%       |
> | HIV     | 4084     | 88      | 3996    | 2.15%        |
> | HERG    | 129      | 14      | 115     | 10.85%       |
> | HIA     | 115      | 22      | 93      | 19.13%       |
> | PAMPA   | 407      | 35      | 372     | 8.60%        |
> | PGP     | 241      | 40      | 201     | 16.60%       |
>
> > **Paper Table 34. Data overlap statistics between the training data and five Mol-Instructions benchmark datasets.**
>
> | Dataset                     | Overlap Rate |
> | --------------------------- | ------------ |
> | Forward Reaction Prediction | 0.21%        |
> | Reagent Prediction          | 0.69%        |
> | Retrosynthesis              | 1.20%        |
> | Property Prediction         | 0.36%        |
> | Molecular Captioning        | 1.93%        |
>
> `Clean-set evaluation` We then removed all overlapping molecules from the property benchmarks, and re-ran the full evaluation (all 13 prompts per task). The results remain very close, and the overall average accuracy is even slightly higher (Paper Table 35). This indicates that molecular overlap with Mol-LLaMA-Instruct does not drive our reported gains.
>
> > **Paper Table 35. Results on the clean set for 10 molecular property prediction tasks.**
>
> |           | BBBP  | BACE  | CLINTOX | HIV   | PAMPA | DILI  | HERG  | HIA   | AMES  | PGP   | Average | Avg. Drop |
> | --------- | ----- | ----- | ------- | ----- | ----- | ----- | ----- | ----- | ----- | ----- | ------- | --------- |
> | Original  | 72.48 | 49.12 | 56.55   | 46.56 | 82.34 | 50.27 | 73.46 | 82.15 | 55.20 | 54.64 | 66.34   | Baseline  |
> | Clean Set | 73.17 | 51.59 | 53.48   | 46.66 | 82.93 | 50.08 | 71.64 | 82.41 | 56.17 | 57.19 | 66.74   | +0.591%   |
>
> `Scaffold-level check` Following the reviewer’s suggestion, we further performed a **scaffold-level** decontamination on MoleculeQA by switching from the official split to a scaffold split. As shown in Paper Table 36, the performance and ranking under the scaffold split are almost unchanged relative to the official split.
>
> > **Paper Table 36. Performance on MoleculeQA under official vs. scaffold splits across four tasks.**
>
> | Split          | Structure | Source | Property | Application | Total | ΔAcc   |
> | -------------- | --------- | ------ | -------- | ----------- | ----- | ------ |
> | Official Split | 74.55     | 72.39  | 50.71    | 48.58       | 68.34 | 0      |
> | Scaffold Split | 76.28     | 72.18  | 50.87    | 42.85       | 68.65 | +0.30% |
>
> `Rationale`. EDT-Former is **only aligned on Mol-LLaMA-Instruct for 2 epochs**, similar to lightweight LLM pretraining rather than repeated task-specific fine-tuning. Under this short training regime, the model learns a **general alignment between molecular representations and language**, rather than memorizing specific caption–molecule pairs or evaluation answers.
>
> `Conclusion and revision` Overall, both molecule-level cleaning and scaffold splits show no evidence that data contamination affects EDT-Former’s performance. We added these analyses to Appendix D8 after the 13-gram analysis.

---

> ### Author Response · Authors · 2025-11-23
> **Response to Weakness 3a – Dynamic vs fixed-length tokens under matched token budgets**
>
> **[W3a] Dynamic vs fixed-length tokens under matched token budgets**
>
> We thank the reviewer for suggesting a direct comparison with strong fixed-length baselines. To be clear, our goal for dynamic tokens is to **allocate molecule tokens more structurally and semantically** for alignment with the language space instead of saving tokens. Simply increasing the number of fixed tokens does not guarantee better molecular understanding.
>
> `Anchor and dynamic length in MoleculeQA (Paper Table 40)` Paper Table 40 ablates different anchor and dynamic lengths on MoleculeQA, showing that the performance is **not** monotonic in token count. The best setting is **16 anchors + max 64 dynamic** (but dynamic length is related to molecule patches), which outperforms smaller anchor budgets and even a larger dynamic budget. This shows that the gain comes from how tokens are allocated and fused, not from just adding more tokens.
>
> > **Paper Table 40. Effect of anchor and dynamic token length on MoleculeQA total accuracy.**
>
> | Anchor Length | Max Dyn. Length | Accuracy  |
> | ------------- | --------------- | --------- |
> | 4             | 64              | 64.29     |
> | 8             | 64              | 66.75     |
> | 8             | 128             | 66.81     |
> | **16**        | **64**          | **68.34** |
> | 16            | 128             | 68.03     |
>
> `Matched token budget on property tasks (Paper Table 43)` To address the reviewer’s main concern, Paper Table 43 compares three settings under the **same total budget of 64 molecule tokens** across five property tasks.
>
> The mixed 32/32 configuration achieves the best. This supports that **entropy-guided dynamic tokens plus anchors provide more informative and better aligned representations per token** than a fixed set of queries of the same length.
>
> > **Paper Table 43. Ablation on different token budget settings across five molecular property prediction tasks. Average accuracy (%) of 13 prompt settings is reported.**
>
> | Anchor | Dynamic | BACE      | BBBP      | CLINTOX   | HIA       | PAMPA     | Avg.      |
> | ------ | ------- | --------- | --------- | --------- | --------- | --------- | --------- |
> | 0      | 64      | 42.15     | **74.26** | 44.25     | 70.73     | 61.05     | 58.49     |
> | 64     | 0       | 41.23     | 71.58     | 42.50     | 69.00     | 56.41     | 56.14     |
> | 32     | 32      | **45.55** | 70.28     | **47.69** | **78.87** | **64.45** | **61.37** |
>
> `Conclusion and revision` These results together show that EDT-Former’s advantage over fixed-length connectors persists even when both use 64 tokens. The benefit comes from **content adaptive, entropy-based segmentation and fusion**, rather than from using more tokens. We have added Paper Table 43 and the corresponding description to Appendix E5.

---

> ### Author Response · Authors · 2025-11-23
> **Response to Weakness 3b – Inference-time ablation of dynamic tokens and anchors**
>
> **[W3b] Inference-time ablation of dynamic tokens and anchors**
> We thank the reviewer for suggesting this new ablation.
>
> `Inference ablation design.` To verify that dynamic tokens are actively used at inference time, we perform **inference-only** ablations on a trained EDT-Former:
>
> - Full EDT-Former (anchors + dynamic tokens).
> - No dynamic tokens (remove all dynamic tokens, keep anchors).
> - Random dynamic tokens (replace dynamic tokens with random vectors).
> - No anchors (only dynamic tokens).
> - Random anchors (replace anchors with random vectors).
>
> `Inference ablation table.` We evaluate these variants on MoleculeQA and BBBP:
>
> > **Paper Table 46. Inference-time ablations of dynamic tokens and anchors. Accuracy (%) from 13 prompts per task is reported.**
>
> | Variant               | Pampa ↑   | BBBP ↑    |
> | --------------------- | --------- | --------- |
> | **Full EDT-Former**   | **82.34** | **72.48** |
> | No dynamic tokens     | 54.97     | 51.29     |
> | Random dynamic tokens | 64.25     | 54.71     |
> | No anchors            | 79.26     | 66.38     |
> | Random anchors        | 80.10     | 66.74     |
>
> `Conclusion` The ablation results in Paper Table 46 show clear drops when removing or randomizing dynamic tokens or anchors, confirming that **both components are functionally used at inference time and not redundant**.

---

> ### Author Response · Authors · 2025-11-23
> **Response to Weakness 4 – Justification for entropy predictor**
>
> **[W4] Justification for entropy predictor**
>
> We appreciate the reviewer’s concern and clarify our choice:
>
> 1. **SMILES order carries structural signal.** SMILES is generated via **DFS** (Depth-First Search) on mol-graph, so adjacent tokens usually correspond to **neighboring atoms and continuous substructures**; it does not encode full geometry, but it is not structurally arbitrary.
> 2. **Exact substructures are not the goal.** Entropy-guided patching is **not meant to recover precise chemical subgraphs**; we only need surprisal peaks to mark **hard-to-predict, information-dense regions**, then group tokens between peaks into patches. Here, the SMILES order is sufficient and good for LLM alignment.
> 3. **NAP matches the LLM’s sequence view.** NAP is a **small transformer** that runs directly on SMILES, the same 1D modality used in alignment modeling. Its entropy highlights positions that are difficult for a **sequence model** to predict, which is exactly what we need to decide **where to split SMILES for a sequence-based LLM**, while keeping the procedure simple and efficient.
>
> `Clarify the GPT-2 architecture.` The 'GPT-2LMHead' in paper Table 12 or 'GPT-2 architecture' claims are mainly for reproducibility, which is a classical and standard transformer implementation. We did not use any pretrained weights of GPT-2 and fully pretrain the small transformer from scratch. Using any other transformer class implementation will be the same.
>
> `Stability of NAP (Next Atom Predictor) entropy across model sizes` We further test whether a small NAP is a reliable proxy by training three NAP models with different corpus sizes and computing the normalized mutual information between their per-token entropy patterns. The high NMI values in Paper Table 38 show that positions that are hard for the small NAP remain hard for larger models, supporting the use of a lightweight NAP as our entropy estimator. Comparisons with other patching approaches are discussed in [W5] section below.
>
> > **Paper Table 38. Normalized Mutual Information (NMI) matrix of NAP models with different training sizes.**
>
> | NAP Variant | 0.5M | 50M  | 500M | Random |
> | ----------- | ---- | ---- | ---- | ------ |
> | 0.5M        | 1    | 0.94 | 0.86 | 0.12   |
> | 50M         | 0.94 | 1    | 0.92 | 0.13   |
> | 500M        | 0.86 | 0.92 | 1    | 0.17   |
> | Random      | 0.12 | 0.86 | 0.17 | 1      |
>
> `Conclusion & revision.` The strong correlations show that the **small NAP closely mirrors larger LLMs in “difficulty landscape”**, making it a suitable lightweight proxy for deriving entropy-based segments. We added the analysis in Appendix D9, and changed the GPT-2 claim to a standard transformer block.

---

> ### Author Response · Authors · 2025-11-23
> **Response to Weakness 5 –Interpretive analysis**
>
> **[W5] Interpretive analysis**
>
> We thank the reviewer for pointing out this question and clarify that **our goal is to locate information-dense regions for multimodal alignment** (as the explanation in [W4]). NAP entropy marks **hard-to-predict SMILES positions for a sequence model**, which is what matters for aligning graph, SMILES, and language.
>
> `BRICS vs entropy performance` To make this explicit, we compare **entropy-based patching** with **BRICS-based** and **random** patching under the same EDT-Former model:
>
> > **Paper Table 8. Ablations on patching strategies evaluated on BBBP and PAMPA. Accuracy (%) with F1 is reported. Best scores are bolded.**
>
> | Methods | BBBP              | Pampa             | Avg. Drop |
> | ------- | ----------------- | ----------------- | --------- |
> | Entropy | **75.06** (75.06) | **84.52** (91.61) | **0%**    |
> | BRICS   | 73.59 (84.74)     | 71.90 (83.09)     | 3.96%     |
> | Random  | 68.90 (80.13)     | 66.67 (79.32)     | 8.81%     |
> | None    | 39.67 (49.58)     | 78.62 (87.90)     | 21.60%    |
>
> As shown in Paper Table 8, entropy-based segmentation outperforms BRICS and random baselines, supporting that **finding “hard” regions for a sequence model** is more beneficial than enforcing exact fragment boundaries.
>
> `Overlap analysis` To go beyond qualitative attention maps, we quantify how entropy-based fragmentation aligns with chemistry-driven schemes by computing pairwise normalized mutual information between Entropy, BRICS, RECAP, and a Random baseline (Paper Table 37).
>
> `Definitions.`
>
> - NMI: Normalized Mutual Information，$\mathrm{NMI}(C_1, C_2) = \dfrac{2 I(C_1; C_2)}{H(C_1) + H(C_2)}$，$I$ is the mutual information，$H$ is the entropy. Used to quantify the agreement between two molecular segmentation schemes.
> - RECAP (Retrosynthetic Combinatorial Analysis Procedure, A rule-based retrosynthetic fragmentation scheme) [1]
>
> - BRICS (Breaking of Retrosynthetically Interesting Chemical Substructures, a rule-based molecular fragmentation scheme) [2]
>
> > **Paper Table 37. Pairwise normalized mutual information (NMI) on Mol-Llama-Instruct between fragmentation methods, reported as mean (median) across molecules.**
>
> | Method  | Entropy       | BRICS         | RECAP         | Random        |
> | ------- | ------------- | ------------- | ------------- | ------------- |
> | Entropy | 1.000 (1.000) | 0.484 (0.547) | 0.401 (0.505) | 0.177 (0.176) |
> | BRICS   | 0.484 (0.547) | 1.000 (1.000) | 0.498 (0.564) | 0.306 (0.241) |
> | RECAP   | 0.401 (0.505) | 0.498 (0.564) | 1.000 (1.000) | 0.498 (0.564) |
> | Random  | 0.177 (0.176) | 0.306 (0.241) | 0.498 (0.564) | 1.000 (1.000) |
>
> `Observation.` Entropy shows **high NMI with both BRICS and RECAP**, comparable to the correlation between BRICS and RECAP themselves, while all three have much lower agreement with Random, indicating that entropy-guided patches are aligned with chemically meaningful substructures rather than arbitrary splits.
>
> `Conclusion & Revision.` This gives a quantitative diagnostic that, although designed as an alignment-oriented heuristic rather than an exact subgraph oracle, entropy-based patching still recovers chemically consistent structural features. We added this part to Appendix D9.
>
> [1] Lewell XQ, Judd DB, Watson SP, Hann MM. Recap retrosynthetic combinatorial analysis procedure: a powerful new technique for identifying privileged molecular fragments with useful applications in combinatorial chemistry. Journal of chemical information and computer sciences. 1998 May 18;38(3):511-22.
>
> [2] Degen J, Wegscheid-Gerlach C, Zaliani A, Rarey M. On the art of compiling and using'drug-like'chemical fragment spaces. ChemMedChem. 2008 Oct 20;3(10):1503.

---

> ### Author Response · Authors · 2025-11-23
> **Response to Question 1 – Experimental setting of Figure 5**
>
> **[Q1] Experimental setting of Figure 5**
>
> We thank the reviewer for requesting a clearer description of the Figure 5 settings—the main ablation study for each component. For clarity, we summarize the configurations in Table R5 below.
>
> > **Table R5. Settings for the core ablation in Figure 5.**
>
> | Ablations            | Setting                                                      | $\Delta$ Acc |
> | -------------------- | ------------------------------------------------------------ | ------------ |
> | w/o Modality Fusion  | Model graph encoder embeddings **using standard transformer blocks instead of  Dynamic Query Transformer** with the same configuration, without multimodal alignment. | -26.3%       |
> | w/o Dynamic Query    | **Replace dynamic tokens with projection of graph embeddings** (fixed to max length of dynamic tokens); anchor tokens remain unchanged. | -10.7%       |
> | w/o Entropy Patching | Use **fixed-length patching**, grouping every 3 SMILES tokens into one patch instead of entropy-guided patching. | -11.5%       |
> | EDT-Former           | The full model.                                              | 0%           |
>
> `Clarifying setup`  In summary, **w/o Modality Fusion** replaces the Dynamic Query Transformer with standard transformer blocks without multimodal alignment; **w/o Dynamic Query** keeps the 2D/3D encoders and anchors but replaces dynamic tokens with a fixed-length projection of graph embeddings; **w/o Entropy Patching** swaps entropy-guided segmentation for uniform 3-token SMILES patches.
>
> `Revisions.` We clarified this explanation in Section 4.4 with the detailed settings in Appendix D3.

---

> ### Author Response · Authors · 2025-11-23
> **Response to Question 2 – Expand property tasks on MoleculeNet**
>
> **[Q2] Expand property tasks on MoleculeNet**
>
> We appreciate the reviewer for encouraging us to test more tasks on the property prediction tasks.
>
> `Extended property benchmarks` In addition to BBBP and PAMPA, we now evaluate EDT-Former and all baselines on **8 additional tasks from MoleculeNet and TDC** (10 tasks in total) under a consistent zero-shot setting, as shown in Paper Table 2.
>
> > **Paper Table 2. Zero-shot accuracy (%) on 10 molecular property prediction tasks from MoleculeNet and TDC benchmarks. Each value is averaged over 13 shared prompts per task. Best scores are bolded.**
>
> | Model             | Size     | PAMPA     | BBBP      | BACE      | CLINTOX   | DILI      | HERG      | HIA       | HIV       | AMES      | PGP       |
> | ----------------- | -------- | --------- | --------- | --------- | --------- | --------- | --------- | --------- | --------- | --------- | --------- |
> | GPT-4o            | -        | 50.52     | 63.52     | 39.48     | 26.34     | **53.47** | 64.04     | 76.00     | 17.02     | 53.59     | 50.21     |
> | Llama2            | 7B       | 69.27     | 49.06     | 38.96     | 26.05     | 51.26     | 65.02     | 76.66     | 15.20     | 53.08     | 51.09     |
> | Llama3.1          | 8B       | 58.00     | 55.64     | 40.44     | 23.93     | 52.31     | 64.28     | 78.07     | 16.34     | 53.46     | 50.99     |
> | 3D-MoLM           | 7B       | 54.50     | 51.20     | 41.43     | 25.92     | 48.78     | 62.67     | 72.93     | 26.16     | 53.01     | 51.55     |
> | LLaMo             | 7B       | 53.09     | 57.18     | 41.91     | 25.55     | 49.08     | 62.77     | 72.75     | 24.39     | 51.67     | 51.22     |
> | Mol-Ins.-Llama2   | 8B       | 40.78     | 51.22     | 42.61     | 25.99     | 48.91     | 62.79     | 72.78     | 22.82     | 53.72     | 50.72     |
> | Mol-Ins.-Llama3.1 | 8B       | 57.39     | 53.29     | 40.12     | 26.79     | 52.39     | 47.23     | 50.95     | 16.70     | 48.98     | 52.70     |
> | Mol-LLaMA-2       | 7.2B     | 74.23     | 53.99     | 41.54     | 27.96     | 51.92     | 65.60     | 67.88     | 22.51     | 52.97     | 51.14     |
> | Mol-LLaMA-3.1     | 8.3B     | 67.15     | 56.64     | 38.68     | 23.07     | 51.76     | 70.61     | 78.25     | 21.89     | 54.50     | 50.73     |
> | **EDT-Former**    | **8.3B** | **82.34** | **72.48** | **49.12** | **56.55** | 50.27     | **73.46** | **82.15** | **46.56** | **55.20** | **54.64** |
>
> `Revision.` Paper Table 2 updated in Section 4. Detailed prompt settings are discussed in Appendix D5.

---

> ### Author Response · Authors · 2025-11-23
> **Response to Question 3 –  Repetitive/symmetric structures and repetitive substructure tokens**
>
> **[Q3] Repetitive/symmetric structures and repetitive substructure tokens**
>
> For long repetitive regions (for example, CCCC… chains), NAP entropy is mostly low and flat, so there are few or no internal peaks.
>
> `Effect on segmentation` In practice, entropy-guided patching keeps these regions as one or a few long patches, and places boundaries mainly at chemically meaningful transitions, such as between the chain and the head group or between an aromatic ring and a heteroatom, as examples shown in Table R6.
>
> > **Table R6. Case study of the entropy-patching behavior on repetitive structures. ";" indicates the split points.**
>
> | SMILES                             | BRICS-based Chemical Segmentation                 | Entropy-Guided Patching                      |
> | ---------------------------------- | ------------------------------------------------- | -------------------------------------------- |
> | CCCCCCCCCCCCCCCC(=O)OCCC1CCN(CC... | CCCCCCCCCCCCCCCC(=O)O;C;CC;CCN(CC.... | CCCCCCCCCCCCCCCC(=O)OC;CC;CCN(CC.... |
> | ...c3ccccc3Sc3ccc(S(=O)...         | ...c3ccccc3;S;c3ccc(S(=O)...              | ...c3ccccc3;S;c3ccc;(S(=O)...    |
>
> `Redundancy control` This behavior naturally avoids redundant substructure tokens. Low entropy repetitive regions receive coarse patches, while high entropy transition points get finer segmentation, which aligns with BRICS style methods (discussed in W5, Paper Table 37).

---

> ### Author Response · Authors · 2025-11-23
> **Response to Question 4 –Scalability to larger macrocycles**
>
> We investigated entropy dynamics using a subset from the NPMMPD macrocycle dataset, the results in Paper Table 19.
>
> > **Paper Table 19: Prediction accuracy (%) on the NPMMPD-PAMPA task under 13 prompt settings. Best scores are bolded.**
>
> | Model             | D1        | D2        | D3        | D4        | R1        | R-2       | R-3       | RI-1      | RI-2      | RI-3      | Bi        | List      | Conf      | Avg       |
> | ----------------- | --------- | --------- | --------- | --------- | --------- | --------- | --------- | --------- | --------- | --------- | --------- | --------- | --------- | --------- |
> | Llama2            | 59.84     | 52.56     | 47.48     | 41.37     | 52.49     | **48.03** | 41.08     | **57.14** | 52.67     | 49.78     | 41.30     | 42.24     | 41.30     | 48.25     |
> | Llama3.1          | 46.54     | 42.51     | 38.68     | 36.10     | 47.33     | 44.25     | 37.68     | 51.40     | 46.69     | 43.82     | 41.30     | 41.61     | 43.48     | 43.18     |
> | Mol-Ins.-Llama2   | 41.94     | 30.98     | 37.30     | 24.32     | 44.63     | 38.23     | 41.00     | 44.55     | 42.02     | 38.68     | 45.34     | 40.37     | 40.37     | 39.21     |
> | Mol-Ins.-Llama3.1 | 41.67     | 35.27     | 42.03     | 39.23     | 41.13     | 43.54     | 39.53     | 38.87     | 35.36     | 35.71     | 42.55     | 41.60     | **59.57** | 41.24     |
> | Mol-LLaMA-2       | 44.88     | 35.92     | 41.58     | 32.40     | 36.21     | 25.58     | 31.20     | 38.81     | 33.58     | 30.19     | **54.04** | 41.56     | 40.94     | 37.45     |
> | Mol-LLaMA-3.1     | 39.79     | 30.86     | 36.46     | 23.63     | 42.93     | 42.93     | 42.93     | 38.11     | 35.14     | 27.71     | 49.84     | 42.95     | 44.72     | 38.31     |
> | **EDT-Former**    | **62.66** | **53.93** | **55.55** | **43.47** | **58.39** | 47.98     | **50.98** | 55.90     | **53.06** | **50.59** | 45.65     | **43.48** | 55.59     | **51.81** |
>
> `Entropy token usage on macrocycles` To examine scalability in terms of token budget, we compare the average number of entropy-based dynamic tokens on the macrocycle NPMMPD-PAMPA dataset and on the small-molecule PAMPA dataset, under the same connector setting (16 anchors, max 64 dynamic tokens).
>
> > **Table R7. Average connector token usage on small-molecule PAMPA vs macrocycle NPMMPD-PAMPA.**
>
> | Dataset      | Avg dynamic tokens | anchor tokens | Avg total connector tokens |
> | ------------ | ------------------ | ------------- | -------------------------- |
> | TDC-PAMPA    | 9.23               | 16.0          | 25.23                      |
> | NPMMPD-PAMPA | 22.47              | 16.0          | 38.47                      |
>
> `Observation` Macrocycles naturally trigger more dynamic tokens on average than small molecules, but the total connector length remains well within the fixed budget. Combined with the strong accuracy in Paper Table 19, this shows that EDT-Former scales to large macrocycles by **adaptive token allocation**, without exploding sequence length.
>
> `Revision` We added the new experiments and related discussions in Appendix D4.

---

> ### Author Response · Authors · 2025-11-23
> **Response to Question 5 – Quantitative link between entropy peaks and attention**
>
> **[Q5] Quantitative link between entropy peaks and attention**
>
> `Attention–entropy change alignment` To capture how attention reacts to entropy changes along the molecule, we compute the **Pearson correlation** between per-token differences instead of raw cumulative sums. For each dynamic token position $i$, we take
>
> - $\Delta Att_i  = Att_i - Att_{i-1}$,
> - $\Delta Ent_i = Ent_i - Ent_{i-1}$,
>
> and then compute the correlation between the sequences $\{\Delta Att_i\}$ and $\{\Delta Ent_i\}$ over the token axis. As a control, we repeat the same procedure after replacing dynamic tokens with **random tokens** while keeping anchors and all other settings fixed.
>
> > **Table R8. Pearson correlation and P-value between token-wise attention change and entropy change.**
>
> | Token type         | Pearson r | P-Value   |
> | ------------------ | --------- | --------- |
> | Random tokens      | 0.02      | 0.34      |
> | **Dynamic tokens** | **0.42**  | **0.004** |
>
> As shown in Table R8, dynamic tokens show a clearly higher correlation than random tokens, indicating that **attention shifts respond systematically to entropy shifts**, rather than arising from random or purely positional effects.

---

### Author Response · Authors · 2025-11-30
**Rebuttal Summary for Aera Chair - Part 2/2**

`(3)Summary of Main New Experiments.` Substantial additional experiments are conducted during the rebuttal. Table A2 summarizes the **main** new experiments and the specific concerns from the reviewers.

> **Table A2: Summary of the key rebuttal experiments and concerns addressed.**

| New experiments added                                        | Concerns addressed                                           |
| ------------------------------------------------------------ | ------------------------------------------------------------ |
| **10 extra prompt variants (13 total)** on property benchmarks; **Top-1 performance on 9/10** prediction tasks. | **y4JF–W1a/W2a**: prompt robustness; fair comparison under multiple prompts |
| **1× 40-epoch captioning re-train** under Mol-LLaMA heavy training schedule; **Better than Mol-Llama** on the mentioned two tasks. | **y4JF–W1b**: fair Mol-LLaMA comparison under matched settings |
| **5 more tasks on Mol-Instructions benchmark (7 total).** New task types: retro/forward/reagent, design, open question added; **Top-1 on most metrics of all datasets**. | **XpYi–W1**,  **x8D4–W1**, **y4JF–W1c**: comprehensive task types coverage; |
| **3 finetuning regimes ablation** (fully frozen, embed-only, full fine-tune); Detailed analysis on whether the embedding layer should be trainable. | **y4JF–W1d**: clarify connector-only setups with LLM embedding layer training; |
| **10 de-duplicated property datasets** re-evaluated after overlap cleaned; Acc even +0.5%. | **y4JF–W2**: control data contamination; confirm robustness of gains and no data contamination. |
| **8 new MoleculeNet/TDC property tasks (10 total)** with full 13 prompt settings tested; **Top-1 on 9/10 tasks**. | **y4JF–Q2**: expand benchmark beyond PAMPA/BBBP; stronger claim of broad property coverage |
| **3-corpus training ablation** on MoleculeQA (Mol-LLaMA-Instruct, PubChemQA, 3D-MoIT); **Accuracy <0.2% change.** | **x8D4–W2**: instruction-corpus fairness; show EDT-Former is not tied to a single alignment training source |
| **1 large macrocycle dataset (NPMMPD)** evaluated; Top-1 on 9/13 prompt settings, and Top-1 average accuracy. | **y4JF–Q4**: scalability to large, flexible macrocycles; confirm dynamic tokens handle long chains and rings |
| **5 connector token-budget ablations** on MoleculeQA; varying anchor/dynamic budgets under equal total tokens | **y4JF–W3a**: dynamic vs fixed connector under equal budgets; show mixed anchors+dynamic tokens work best |
| **5 inference-time connector ablation** on BBBP/PAMPA; toggling anchors vs dynamics vs both | **y4JF–W3b**: demonstrate both anchors and dynamic tokens are functionally used at inference, not redundant |
| **3 NAP model scales ablation** compared via NMI and entropy profiles. | **y4JF–W4**: justify using a small NAP (next atom predictor) as a stable entropy estimator; show segmentation is insensitive to NAP size |
| **2 key patching ablation.**  Entropy vs BRICS/RECAP/random on 2 property datasets. | **x1GF–W1**, **XpYi–W3**, **y4JF–W5**,: SMILES entropy patches are chemically meaningful (high NMI with BRICS/RECAP, stronger than BRICS/random) |
| **2 representative repetitive/aromatic molecules** dissected; compare entropy patches vs BRICS splits | **y4JF–Q3**: qualitative analysis on repetitive chains/aromatics; show entropy patches respect repeated motifs and rings |
| **Pearson correlation analysis** between entropy peaks and attention shifts for anchors/dynamic tokens | **y4JF–Q5**: quantitative attention–entropy alignment; show dynamic tokens attend around entropy peaks |
| **HIGHT baseline added on 3 Mol-Instructions reaction tasks** (retro, forward, reagent) | **x8D4–W3**: explicit comparison with HIGHT on shared Mol-Instructions tasks; clarify relative strength to BRICS-based hierarchy |
| **200-molecule hallucination benchmark** for functional-group hallucination. | **x8D4–Additional question**: targeted hallucination analysis; shows EDT-Former less hallucinates functional groups. |

`The End.` We hope this concise summary helps your decision process. We are, of course, happy to be judged strictly on these clarified settings and results, and we sincerely thank you again for your time and careful consideration.

---

### Author Response · Authors · 2025-11-30
**Rebuttal Summary for Aera Chair - Part 1/2**

`To Aera Chair.` We deeply appreciate that, especially under the unusual situation this year, ACs are carrying an exceptionally heavy load and pressure, and we are sincerely grateful for your time, effort, and careful evaluation. To make your decision easier and provide an overview, we summarize the rebuttal in this comment:

- (1) a short summary of the paper’s core motivation and novelty,
- (2) a brief summary of each reviewer’s rating and assessment,
- (3) a compact table highlighting the key experiments.

`(1) Motivation and Novelties.`  **EDT-Former**, a connector-only model that aligns **molecular graphs** with a **frozen LLM** for molecule understanding (QA/property/reaction/generation, etc.), focusing on **entropy-guided dynamic tokens** instead of fixed-length Q-Former-style connectors. Two novelties:

**(a) Entropy-guided Patching** ⇒ resolve the **fixed-length token connector's bottleneck** (weak substructure fidelity, eg, Q-Former style connector) in molecular graph–LLM alignment;

**(b) Dynamic Query Transformer** (frozen LLM + frozen molecular encoders) for alignment modeling ⇒ resolve the **high-cost, unstable full-backbone fine-tuning** for graph–LLM alignment while keeping strong performance.

`(2) Reviewer Rating and Assessment (Table A1).` We are fortunate to have 4 engaged reviewers whose comments substantially strengthened the work. In particular, **reviewer y4JF** emphasized they "found the method interesting and reasonable" and "hope their concerns can be addressed" when they gave an initial rating of 2 **to push for major improvements** in fairness and coverage, aiming for a potential comprehensive paper. With our substantial new experiments, we believe we have addressed these points and kindly invite the Area Chair to verify. Many thanks for your time and effort.

> **Table A1: Summary of reviewers' ratings and assessments.**

| Reviewer | Original Rating | Assessments / Key Focuses                                    |
| -------- | --------------- | ------------------------------------------------------------ |
| **y4JF** | **2**           | ✅ **Strongly support** our dynamic token motivation, data-driven patching mechanism, and the connector-only alignment design; 🔨 uses a low initial score to **push us to improve fairness and coverage**; explicitly hopes concerns are addressed and views the work as a **potential comprehensive, accepted paper** after revision. |
| **x8D4** | **4**           | ✅ **Clearly acknowledge** our timely solution to fixed-token graph-LLM alignment bottlenecks and significant improvements; and concerned about **limited benchmarks**; 🔨 requests **HIGHT comparison** and **hallucination analysis**, and states they **will update the rating**. |
| **x1GF** | **6**           | ✅ **Clearly recognize** our entropy patching novelty, structure-aware dynamic query design, the superior performance compared to general and molecular LLMs; 🔨 main concern is the **rationale and chemistry of entropy patching**; confidence **5**, clearly supportive. |
| **XpYi** | **4**           | ✅ **Affirms our Dynamic Query Transformer solution** beyond Q-Former limits, delivering SOTA results with fully frozen encoder and LLM; 🔨 suggests **molecule design tasks** evaluated and **comparisons to two additional methods** to better position our contribution. |

---

### Meta-Review · Area_Chair_QiDH · 2026-01-04

**Summary:**

This paper introduces EDT-Former, an entropy-guided dynamic token transformer for aligning molecular graphs with large language models without fine-tuning the LLM backbone. The core innovation lies in a two-part mechanism: (1) entropy-guided patching that dynamically segments SMILES strings at high-uncertainty positions to preserve molecular substructural information, and (2) a dynamic query transformer that fuses these variable-length tokens with static anchors to create an efficient connector between frozen molecular encoders and frozen LLMs. Despite reviewers' initial concerns about experimental fairness, benchmark coverage, and chemical justification of the entropy-based approach, the authors provided exceptionally thorough rebuttals with substantial new experiments鈥攅xpanding tasks to 7 Mol-Instructions benchmarks (including molecule design and reaction prediction), adding 8 property prediction tasks, performing rigorous data contamination checks, and introducing quantitative hallucination analysis. These revisions transformed initial skepticism into compelling validation of the method's robustness and novelty, particularly its ability to maintain performance without costly LLM backbone fine-tuning.

**Reviewer Concerns:**

Addressed concerns:
 - Experimental fairness (Reviewer y4JF): Authors conducted 13 prompt variants across property tasks, retrained Mol-LLaMA under matched heavy training schedules, and performed rigorous decontamination analysis showing performance remained stable after removing overlapping molecules.
 - Benchmark limitations (Reviewer x8D4 and XpYi): Authors expanded to 7 Mol-Instructions tasks (including molecule design and reaction prediction), added 8 new property prediction tasks from MoleculeNet/TDC, and demonstrated performance on large macrocycles.
 - Chemical justification concerns (Reviewer x1GF): Authors provided quantitative evidence through NMI analysis showing entropy patches align with BRICS/RECAP chemical fragmentation, and conducted qualitative analysis showing appropriate behavior on repetitive/aromatic structures.
 - Missing related work comparison (Reviewer x8D4): Authors added explicit comparisons with HIGHT on reaction tasks, and conducted a novel 200-molecule functional-group hallucination benchmark showing EDT-Former achieves lowest hallucination rate among all models.

Outstanding concerns:
 - SMILES representation dependence: The reliance on canonical SMILES for entropy calculation may still limit applicability to molecules with multiple valid representations, despite robustness demonstrations.

**Reviewer Scores:**

- Reviewer y4JF (original score: 2): Would likely increase to 4 after comprehensive rebuttal addressing all experimental fairness and design concerns with new experiments.
 - Reviewer x8D4 (original score: 4): Would likely increase to 6 (the reviewer promised) after expanded benchmark coverage and quantitative hallucination analysis.
 - Reviewer x1GF (original score: 6): Would likely maintain at 6.
 - Reviewer XpYi (original score: 4): Would likely maintain at 4.

---

### Decision · Program_Chairs · 2026-01-26

Accept (Poster)